# GPI-anchored FGF directs cytoneme-mediated bidirectional contacts to regulate its tissue-specific dispersion

Lijuan Du [1], Alex Sohr [1,2], Yujia Li [1] & Sougata Roy [1]✉

How signaling proteins generate a multitude of information to organize tissue patterns is critical to understanding morphogenesis. In *Drosophila*, FGF produced in wing-disc cells regulates the development of the disc-associated air-sac-primordium (ASP). Here, we show that FGF is Glycosylphosphatidylinositol-anchored to the producing cell surface and that this modification both inhibits free FGF secretion and promotes target-specific cytoneme contacts and contact-dependent FGF release. FGF-source and ASP cells extend cytonemes that present FGF and FGFR on their surfaces and reciprocally recognize each other over distance by contacting through cell-adhesion-molecule (CAM)-like FGF-FGFR binding. Contact-mediated FGF-FGFR interactions induce bidirectional responses in ASP and source cells that, in turn, polarize FGF-sending and FGF-receiving cytonemes toward each other to reinforce signaling contacts. Subsequent un-anchoring of FGFR-bound-FGF from the source membrane dissociates cytoneme contacts and delivers FGF target-specifically to ASP cytonemes for paracrine functions. Thus, GPI-anchored FGF organizes both source and recipient cells and self-regulates its cytoneme-mediated tissue-specific dispersion.

[1] Department of Cell Biology and Molecular Genetics, University of Maryland, College Park, MD 20742, USA. [2] Present address: Division of Cell and Gene Therapy, Center for Biologics Evaluation and Research, Food and Drug Administration, Silver Spring, MD 20993, USA. ✉email: sougata@umd.edu

During development, intercellular communication of morphogens is critical for embryonic cells to determine their positional identity, directionality, and interactions in an organized pattern to sculpt tissue. These conserved families of secreted morphogens/signals, such as fibroblast growth factor (FGF), Hedgehog (Hh), Wingless (Wg)/Wnt, epidermal growth factor (EGF), and decapentaplegic (Dpp—a bone morphogenetic protein (BMP) homolog), act away from their sources and, upon binding to receptors, activate gene regulatory pathways to induce functions in recipient cells[1,2]. Strikingly, each signal and signaling pathway can generate a wide range of cell types and organizations in diverse contexts[3]. Understanding how signals might inform cells of their positional identity, directionality, and interactions and organize these functions in diverse tissue-specific patterns is critical to understanding morphogenesis.

The discrete tissue-specific organization of morphogen signaling is known to be dependent on the ability of signal-receiving cells to selectively sense and respond to a specific signal[3]. In contrast, traditional models predict that the signal presentation from the source via free secretion and extracellular diffusion is a non-selective process. However, recent advances in microscopy revealed that both signal-producing and receiving cells could extend signaling filopodia named cytonemes and selectively deliver or receive signals through cytoneme–cell contact sites[4–9]. Essential roles of cytonemes or cytoneme-like filopodia have been discovered in many vertebrate and invertebrate systems and are implicated in most signaling pathways, including Hh, Dpp, FGF, EGF, Ephrin, and Wnt under various contexts[4–18]. The prevalence and similarities of these signaling filopodia suggest that the polarized target-specific morphogen exchange through filopodial contacts is an evolutionarily conserved signaling mechanism.

These findings bring along a paradox - not only do signals instruct cells and organize discrete cellular patterns, but cells also control the patterns of signal presentation and reception by organizing the distribution of cytonemes and cytoneme contacts[6,9]. This interdependent relationship of signals and signaling cells through cytonemes, however, would require precise spatiotemporal coordination between cytoneme contact formation and signal release. We started the current investigation with the premise that a better understanding of the processes that produce cytoneme contacts and control contact-driven signal release is essential to understanding morphogenesis. We asked: (1) How do cytonemes recognize a specific target cell and form signaling contacts? (2) How are secreted signals controlled for polarized target-specific release, exclusively at the cytoneme contact sites? (3) Do cytoneme contact formation and signal release spatiotemporally coordinate with each other? If so, how?

To address these questions, we focused on the inter-organ dispersion of a *Drosophila* FGF, Branchless (Bnl), during the development of the wing imaginal disc-associated air-sac primordium (ASP)[19,20]. Bnl is expressed in a discrete group of wing disc cells, and it induces morphogenesis of the tubular ASP epithelium that expresses the Bnl receptor, Breathless (FGFR/Btl)[9,19,21]. Epithelial cells at the ASP tip extend polarized Btl-containing cytonemes to contact Bnl-producing wing disc cells and directly take up Bnl in a contact- and receptor-dependent manner[5,9]. The formation of Bnl-specific polarity and contacts of ASP cytonemes are self-sustained by Bnl-signaling feedbacks[9]. Consequently, Bnl reception and signaling via cytonemes can precisely adapt and dynamically coordinate with ASP growth. With increasing distance from the Bnl-source, ASP cells extend gradually fewer polarized Bnl-receiving cytonemes, leading to the emergence of asymmetric Bnl dispersion and signaling patterns within the ASP[9]. However, how ASP cytonemes might recognize the *bnl*-source for signaling contacts, and, on the other hand, how

Bnl producing cells might both inhibit free Bnl secretion and facilitate Bnl release selectively at the cytoneme contact sites are unknown.

Here we report that Bnl is post-translationally modified by the addition of a glycosylphosphatidylinositol (GPI) moiety, which anchors Bnl to the outer leaflet of its source cell membrane. We provide evidence that the GPI anchoring of Bnl enables Bnl source cells to selectively present the signal to Btl-expressing cells through cytonemes, and that the cell adhesion molecule (CAM)-like[22–28] Btl–Bnl interactions coordinate bidirectional matchmaking of cytonemes for contacts. Importantly, although the GPI anchor inhibits free Bnl secretion, it promotes contact-mediated tissue-specific Bnl release for long-range patterning. These findings suggest that while cytonemes are critical for organizing tissue-specific Bnl signaling, the GPI-anchored Bnl programs the spatiotemporal distribution of cytoneme contacts to self-regulate its dispersion.

## Results

**The reciprocal polarity of Bnl delivery and reception.** Bnl is produced in the wing disc and transported target-specifically to the overlaying ASP via Btl-containing ASP cytonemes across a layer of interspersed myoblasts (Fig. 1a, b)[9]. The *bnl*-specific polarity of ASP cytonemes might be determined by the extrinsic patterns of Bnl presentation from the source. Previously, non-permeabilized anti-Bnl immunostaining ($\alpha$Bnl$^{ex}$) designed to detect secreted externalized Bnl (Bnl$^{ex}$)[9,29] showed that the Bnl$^{ex}$ was not randomly dispersed in the source-surrounding extracellular space (Fig. 1a). Instead, Bnl$^{ex}$ was restricted exclusively to the basal surface of Bnl-producing cells and on the ASP cytonemes. Importantly, even within the *bnl*-expressing disc area, Bnl$^{ex}$ puncta were asymmetrically congregated near the contact sites of Btl-containing ASP cytonemes that received Bnl$^{ex}$ (Fig. 1a, c). These results indicated that the Bnl presentation is likely to be spatially polarized.

To examine whether Bnl distribution in source cells is spatially biased toward the ASP, we co-expressed Bnl:GFP with mCherry-CAAX (prenylated mCherry for membrane marking) under *bnl-Gal4*. Strikingly, although *bnl-Gal4*-driven mCherryCAAX equally labeled all source cells, Bnl:GFP was asymmetrically enriched at the ASP-proximal source area (Fig. 1d–d″ and Supplementary Fig. 1a, a′). Bnl:GFP puncta were also displayed on short polarized cytonemes emanating from the ASP-proximal disc cells (Fig. 1d; Supplementary Fig. 1a). To further verify if the Bnl presentation is polarized via cytonemes, we imaged the distribution of endogenous Bnl:GFP$^{endo}$, expressed from a *bnl:gfp$^{endo}$* knock-in allele[9], in the mCherryCAAX-marked *bnl*-source. Bnl:GFP$^{endo}$ puncta represented all Bnl isoforms. Indeed, Bnl:GFP$^{endo}$ puncta were selectively enriched in source cell cytonemes that were polarized toward the ASP (Fig. 1e, e′ and Supplementary Fig. 1b).

To examine the organization of Bnl-presenting source cytonemes, we observed live wing discs that expressed a fluorescent membrane marker (e.g., CD8:GFP or CherryCAAX) either in all of the *bnl*-expressing cells (Fig. 1f, g) or in small clones of cells within the Bnl-expressing area (see Methods; Fig. 1h–h″). Three-dimensional image projections of live discs revealed that each of the Bnl-expressing columnar cells proximal to the ASP extended ~2-4 short (<15 μm) cytonemes perpendicularly from their basal surface (Fig. 1g–h″; Supplementary Fig. 1d, and Supplementary Movie 1). The organization of source cells, therefore, can be described as polarized for Bnl presentation with basal cytonemes extending toward the ASP or ASP cytonemes. This organization is mirrored in the ASP, which is known to exhibit polarized Btl presentation, Bnl reception, and cytoneme orientation toward source cells[9]. Thus, the cellular

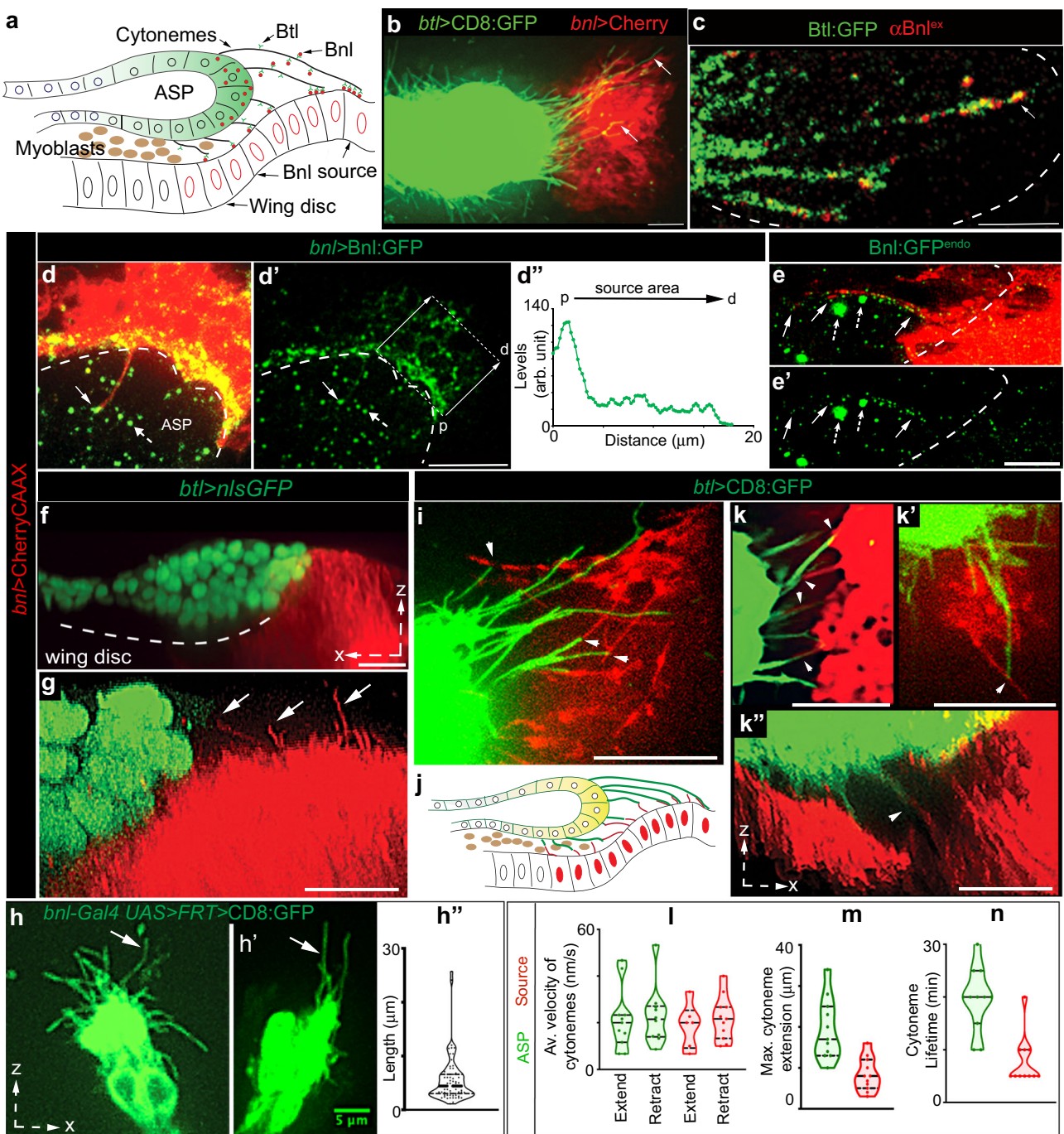

**Fig. 1 The reciprocal polarity of Bnl presentation and reception via cytonemes. a** Drawing depicting the organization of the ASP, wing disc, myoblasts, and Btl-containing ASP cytonemes receiving Bnl from the disc *bnl* source. **b** Spatial organization of the wing disc *bnl*-source (*bnl-LexA,LexO-mCherryCAAX*) and ASP (*btlGal4,UAS-CD8:GFP*) cytonemes (arrow). **c–e'** Polarized Bnl presentation from the source orienting toward the ASP; **c** polarized clustering of externalized Bnl[ex] (red; αBnl[ex]) at the contact sites of the unmarked source (dashed lined area) and Btl:GFP-containing ASP cytonemes (Btl:GFP - *btl:GFP fTRG*, see Supplementary Table 4); **d, d', e, e'** polarized cytoneme-mediated presentation of overexpressed Bnl:GFP (d-d''; *UAS*-mCherryCAAX/+; *bnl-Gal4/ UAS*-Bnl:GFP) and endogenous Bnl:GFP[endo] (**e, e'**; *UAS*-mCherryCAAX/+; *bnl-Gal4/bnl:gfp[endo]*) from the mCherryCAAX-marked *bnl*-source, orienting toward the overlaying ASP (dashed line); dashed arrow, Bnl:GFP puncta in internalized vesicles[9] within the ASP (dashed line); arrow, Bnl:GFP puncta on source cytonemes; **e, e'** airyscan image; **d''** Bnl:GFP intensity plot within the boxed source area in d' along the proximal (p)-to-distal (d) direction (arrows) relative to the ASP, showing selective enrichment of the overexpressed signal toward the ASP. **f–h''** 3D-rendered images showing the ASP-specific polarity of source cytonemes (arrows); **f, g** mCherryCAAX-marked source and nlsGFP-marked ASP (*btl-Gal4,UAS*-nlsGFP/+; *bnl-LexA,LexO*-mCherryCAAX/+); **h, h'** CD8:GFP-expressing mosaic clones within the *bnl* source area (see Methods); **h''** violin plot displaying the source cytoneme length distribution (see Supplementary Fig. 1d). **i–n** Contact-dependent reciprocal guidance of source (red) and ASP (green) cytonemes (*btl-Gal4,UAS*-CD8:GFP/+; *bnl-LexA,LexO*-mCherryCAAX/+); arrowhead, contact site; **j** illustration of **i-k''**; **l-n** violin plots displaying ASP (green) and source (red) cytoneme dynamics as indicated (also see Supplementary Fig. 1e–h and Supplementary Table 1 for statistics). All except **c–e'**, live imaging. Violin plots: black dotted lines - median and 25th and 75th percentiles. Source data are provided as a Source data file. Scale bars, 20 μm; 5 μm (**e, e', h, h'**).

components responsible for Bnl presentation in the disc source and for its reception in the ASP are likely to be reciprocally polarized toward each other.

**Reciprocal guidance of Bnl-sending and -receiving cytonemes.** To examine if Bnl-presenting and -receiving cytonemes could reciprocally guide each other's polarity, we examined live wing discs harboring the CD8:GFP-marked ASP and mCherryCAAX-marked source. Time-lapse imaging of ex vivo cultured discs revealed that ASP and source cytonemes orient toward each other and transiently contact each other's tips, bases, or shafts as they dynamically extend and retract (Fig. 1i–k″ and Supplementary Movie 2). Both cytoneme types had short lifetimes and repeated cycles of contact association-dissociation (Fig. 1l–n; Supplementary Fig. 1e–h, Supplementary Movie 2, and Supplementary Table 1). We also examined the inter-cytoneme interactions during the development of the ASP from the early-to-late L3 larval stages. Despite dynamic morphological changes in the growing ASP and disc, the relative positions of the ASP, *bnl*-source, and the site of inter-cytoneme interactions were maintained throughout the development (Supplementary Fig. 1i–l″). Thus, interacting cells in the ASP and *bnl*-source polarize to face each other and apparently maintain a cytoneme-forming signaling niche at the ASP:source interface.

Based on our previous observations[5,9], Bnl is exchanged at the cytoneme contact sites. However, it was technically challenging to visualize Bnl exchange during dynamic inter-cytoneme interactions. Therefore, we sought to genetically ablate source cytonemes in *bnl:gfp*[endo] larvae and analyze if the level of Bnl:GFP[endo] uptake in the ASP is reduced. An actin modulator formin, Diaphanous (Dia), could influence source cytonemes. Overexpression of Dia:GFP or a constitutively active Dia:GFP[ca] induced source cytonemes (Fig. 2a–e). Asymmetric enrichment of Dia:GFP[ca] puncta in source cytoneme tips suggested localized Dia activity (Fig. 2c). In contrast, *dia* knockdown (*dia-i*) in the mCherryCAAX-marked source (*bnl-Gal4 x UAS-dia-i,UAS-mCherryCAAX*) suppressed cytoneme formation without any visible effects in *bnl* expression (*bnl enhancer*-driven mCherry levels; Fig. 2a–e and Supplementary Fig. 2a). Importantly, the *dia-i* mediated ablation of source cytonemes in *bnl:gfp*[endo] larvae significantly reduced Bnl:GFP[endo] uptake in the ASP. These ASPs were abnormally stunted, suggesting a reduction in Bnl signaling (Fig. 2f–h). Thus, source cytonemes are required to deliver Bnl to the ASP.

Inter-cytoneme Bnl exchange is consistent with reports that Hh and Wg are both sent and received by cytonemes[10,30,31]. However, how do source and ASP cytonemes find and adhere to each other? Dynamic interactions of Bnl-exchanging cytonemes that are convergently polarized toward each other suggested a possibility of contact-dependent reciprocal guidance of source and recipient cytonemes. To test this possibility, we first ablated source cytonemes by *dia-i* expression and analyzed the non-autonomous effects on CD2:GFP-marked ASP cytonemes. The ablation of source cytonemes significantly reduced the long, polarized ASP tip cytonemes (Fig. 2i–k). In contrast, short, randomly oriented ASP cytonemes were unaffected. Thus, Bnl-presenting cytonemes are required for the formation of the polarized Bnl-receiving ASP cytonemes.

We next removed ASP cytonemes by expressing *dia-i* under *btl-Gal4* and recorded non-autonomous effects on mCherryCAAX-marked source cytonemes. The *dia-i* expression had to be controlled with Gal80[ts] to avoid lethality (see "Methods"). Tracheal *dia-i* expression not only reduced ASP cytonemes but also non-autonomously reduced source cytonemes (Fig. 2l–n′). Tracheal expression of a dominant-negative form of Btl (Btl-DN) was known to suppress ASP development without affecting wing disc growth[19].

When both source and Btl-DN-expressing tracheal cells were marked, the complete loss of ASP and ASP cytonemes was found to produce a corresponding loss of *bnl*-source cytonemes (Fig. 2o, p). Thus, Btl-presenting ASP cytonemes are required to produce source cytonemes that polarize toward the ASP. Collectively, these results suggested that the source and recipient cytonemes reciprocally guide each other to form signaling contacts.

**Btl-Bnl binding induces bidirectional contact matchmaking.** The above results also suggested that the inter-cytoneme interactions might recruit and activate a bidirectional signaling mechanism, responses of which could induce ASP cells to extend Btl-containing cytonemes toward the source and activate source cells to extend Bnl-containing cytonemes toward the ASP. We hypothesized that such selective matchmaking between source and ASP cytonemes could be mediated by the binding of surface-displayed Btl and Bnl. In this model, Btl and Bnl are analogous to cell-recognition or cell-adhesion molecules (CAMs), physical interaction of which can produce selective cell-cell adhesion and contact-mediated bidirectional signaling[23,32,33]. CAM-like inter-cellular interactions were known to control cell shapes/polarity and induce contact-dependent bidirectional signaling by modulating local actomyosin complex[22–28]. The initiation of CAM-like interactions might not require Btl to activate the canonical transcriptional outputs[34]. An alternative possibility is that the Bnl-Btl binding activates MAPK signaling and transcription of target genes in the ASP, and these gene products, in turn, non-autonomously act on the wing disc *bnl*-source to induce a response.

Notably, Btl-DN can bind to Bnl via its extracellular ligand-binding domain (Supplementary Fig. 2b) and can heterodimerize with *WT* Btl to inactivate nuclear MAPK signaling due to the lack of its intracellular kinase domain[19,21]. As observed before[13,19], while most wing discs with Btl-DN-expressing trachea (TC) completely suppressed ASP and ASP cytonemes, a few discs produced rudimentary ASP with reduced numbers of cytonemes. This phenotype was likely to be due to the partial dominant-negative effects of Btl:DN. Strikingly, in each of these discs, the appearance of polarized ASP cytonemes correlated with the concomitant appearance of similar numbers of polarized source cytonemes forming direct cytoneme:cytoneme contacts (Fig. 3a–c). These results suggested that the contact-dependent binding of Btl-DN with Bnl could induce the reciprocal cytoneme-forming responses between source and ASP cells.

Expectedly, non-permeabilized αBnl[ex] staining showed that Bnl[ex] was selectively enriched at these inter-cytoneme contact sites (Fig. 3d, d′ and Supplementary Fig. 2c–c‴). Similarly, when we expressed Btl-DN:Cherry under *btl-Gal4*, ASP cytonemes were enriched with Btl-DN:Cherry puncta that colocalized with Bnl[ex] (Fig. 3e, e′ and Supplementary Fig. 2d, d′). This result is consistent with the heterophilic CAM-like activity of surface-bound Btl and Bnl.

To verify if Btl and Bnl can act as CAMs, we performed an in vitro cell culture-based assay using *Drosophila* S2 cells (embryonic hemocyte lineage) that lack endogenous Btl and Bnl expression (modENCODE). When S2 cells ectopically expressed Bnl, αBnl[ex] immunostaining detected Bnl[ex] only on the expressing cell surfaces, like in the wing disc Bnl source (Supplementary Fig. 2e, e′). Moreover, the lack of polarity and cell junctions in S2 cells was suitable for ectopic induction of these properties. As illustrated in Fig. 3f, we mixed and co-incubated Bnl:GFP-expressing cells (S2-Bnl:GFP) with cells that expressed either Btl:Cherry (S2-Btl:Cherry), Btl-DN:Cherry (S2-Btl-DN:Cherry), or a secreted Btl:Cherry (sBtl:Cherry) that lacked its transmembrane and intracellular domains (S2-sBtl:Cherry, see "Methods").

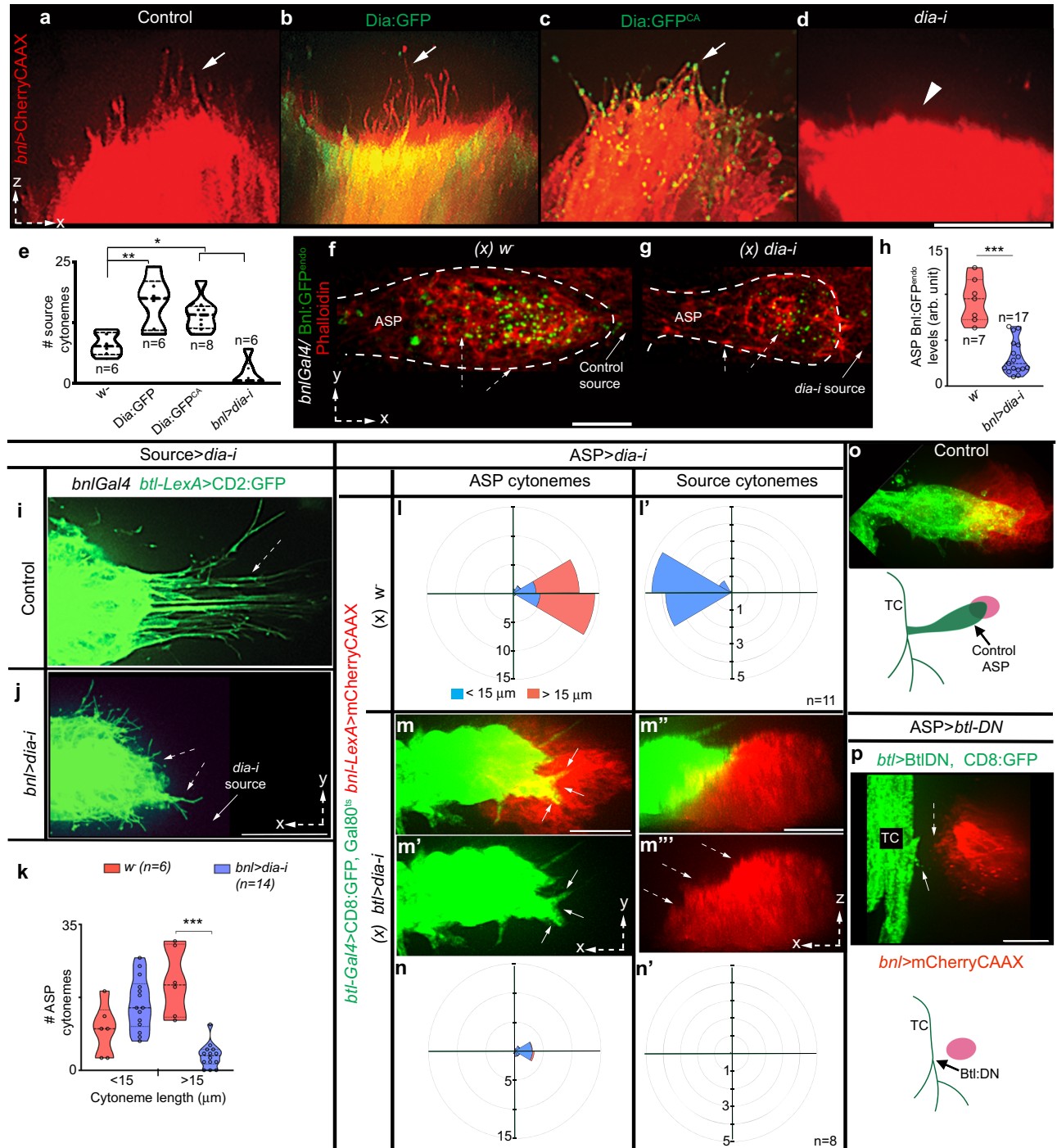

**Fig. 2 Bidirectional matchmaking of Bnl-sending and -receiving cytonemes. a–e** 3D-rendered views of mCherryCAAX-marked *bnl*-source, showing autonomous effects of Dia:GFP, Dia:GFP$^{CA}$, and *diaRNAi* expression on source cytoneme numbers (*UAS-mCherryCAAX;bnl-Gal4* X w⁻ for control, or *UAS-* "X"); arrow, source cytonemes, arrowhead, missing source cytonemes. **e** Violin plots showing numerical values; p values (one-way ANOVA followed by Tukey's honestly significant different (HSD) test)—$p < 0.05$ (*) and $p < 0.01$ (**). **f–h** Levels of Bnl:GFP$^{endo}$ uptake (dashed arrow) in the ASP (dashed line) from wild-type source cells (**f**; control: *bnl:gfp$^{endo}$* X *bnl-Gal4*) and from *dia-i*-expressing source cells (**g**; source cytoneme-depleted condition: *UAS-dia-i,bnl:gfp$^{endo}$* X *bnl-Gal4*), **h** Violin plots showing numerical values, as indicated; p value (***)—$0.06 \times 10^{-5}$ (unpaired two-tailed *t* test). **i–k** Comparison of numbers of CD2:GFP-marked ASP cytonemes (dashed arrow) of various length produced under the control condition and under *dia-i*-induced source cytoneme depleted conditions as indicated. **k** violin plots showing numerical values as indicated; p value (***)—$0.076 \times 10^{-5}$ (unpaired two-tailed *t* test). **e, h, k** Violin plots: black dotted lines show the median as well as 25th and 75th percentiles. **l–n′** Non-autonomous effects of *dia-i*-induced depletion of ASP cytonemes (arrows; **m, m′**) on source cytonemes (dashed arrows; **m″, m‴**); **l, l′, n, n′** R-plots, showing the correlation of ASP and source cytoneme number, length and orientation in control (**l, l′;** w⁻) and *btl > dia-i* condition (**n, n′**); see Supplementary Table 2 for statistical values. **e, h, k, l, l′, n, n′** sample size (*n*)—numbers of independent wing discs/per genotype. **o, p** The *Btl-DN*-induced depletion of ASP cytonemes (arrow) non-autonomously depleted source cytonemes (dashed arrow); genotypes—*btl-Gal4,UAS-CD8GFP/+; bnl-LexA,LexO-*mCherryCAAX/+ (**o**); *btl-Gal4,UAS-*CD8GFP/+; *bnl-LexA,LexO-*mCherryCAAX/UAS-Btl-DN (**p**). All panels (except **f, g**), live imaging. Source data are provided as a Source data file. Scale bars, 20 μm.

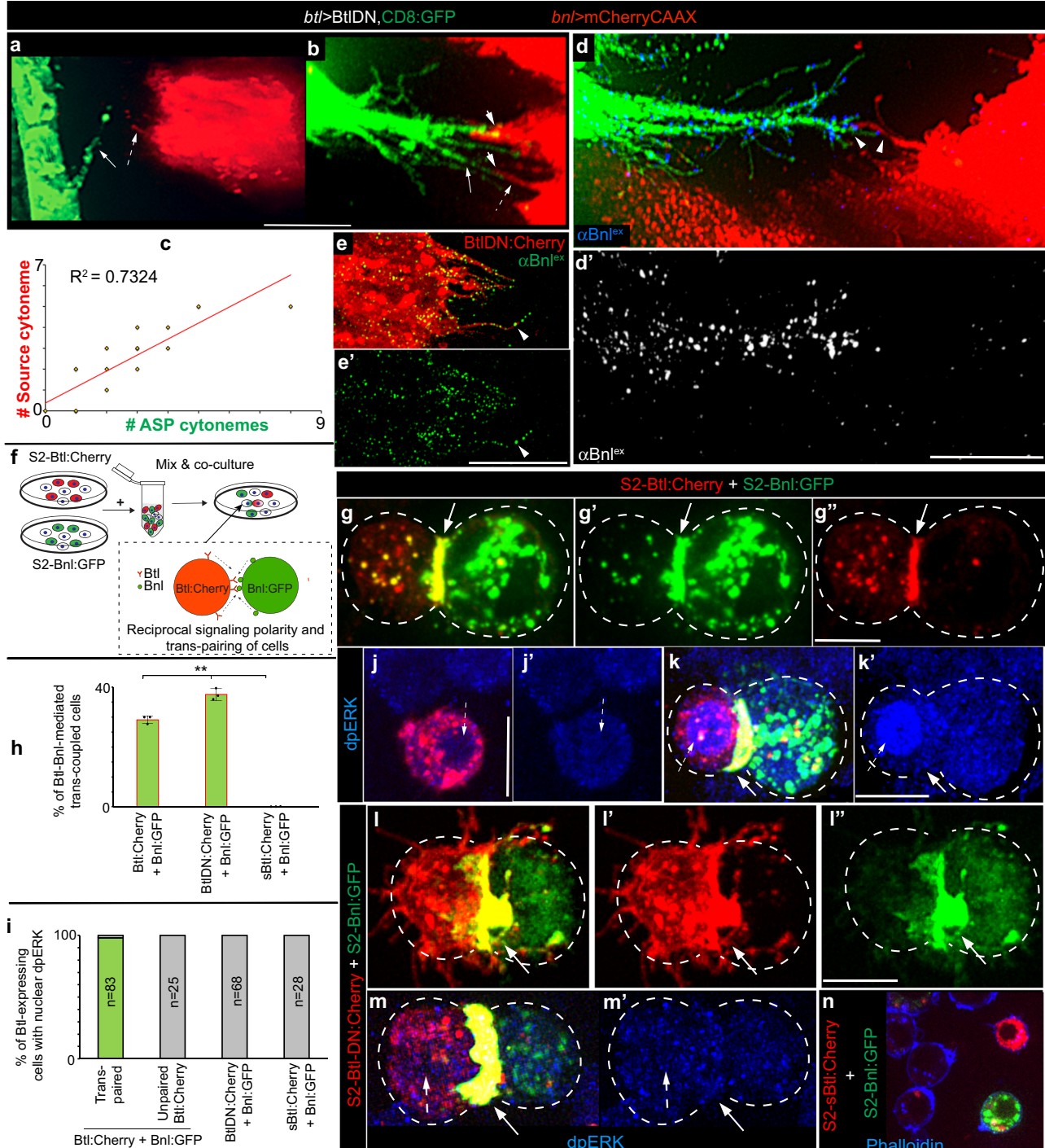

**Fig. 3 CAM-like Btl-Bnl interactions produce signaling contacts. a–c** Correlation of numbers of polarized Btl-DN-expressing ASP cytonemes (arrow) with the polarized *bnl*-source cytoneme (dashed arrow) (*btl-Gal4,UAS*-CD8GFP/+; *bnl-LexA,LexO*-mCherryCAAX/*UAS-Btl-DN*; see "Methods"); arrowheads, contact sites between ASP and source cytonemes; c, graph showing the correlation between source and ASP cytoneme numbers in each disc (*n* = 19). **d, d′** Bnl[ex] (blue, αBnl[ex]) is asymmetrically enriched at the contact sites (arrowheads) between the source and Btl-DN-expressing ASP cytonemes; **d′** only αBnl[ex] channel from **d. e, e′** Btl-DN:Cherry colocalized with Bnl[ex] (green; αBnl[ex]) on tracheal cytonemes; **e′** only αBnl[ex] channel from **e. f** Drawing illustrating the CAM assay using S2 cells and the binding of cell surface Btl:Cherry and Bnl:GFP that could induce reciprocally polarized synaptic co-clustering of receptors and ligands and trans-pairing of the interacting cells. **g–n** Trans-pairing of S2-Bnl:GFP with either S2-Btl:Cherry (**g–g″, k, k′**) or S2-Btl-DN:Cherry (**l–m′**) but not with S2-sBtl:Cherry control (**n**); dashed lines, cell outline; arrow, trans-synaptic receptor-ligand co-clusters; dashed arrow, nucleus; dpERK (blue, αdpERK). **h** Bar graphs comparing the mean frequency (±SD) of receptor-ligand trans-pairing from three independent experiments (see "Methods"); *p* value (**)—p < 0.01 (one-way ANOVA followed by Tukey HSD test); total GFP-positive + mCherry-positive cells analyzed: 1916 (Btl:Cherry + Bnl:GFP), 1868 (BtlDN:Cherry + Bnl:GFP), 1532 (sBtl:Cherry + Bnl:GFP). **i** Graphs comparing the percentage of dpERK-positive Btl-expressing cells that are either trans-paired (for Btl:Cherry and Btl-DN:Cherry) or nearby (for sBtl:Cherry & Unpaired Btl:Cherry) to S2-Bnl:GFP cells; all panels except **a**, **b**: fixed cells/tissues. Source data are provided as a Source data file. Scale bars, 20 μm; 10 μm (**g–n**).

S2-Bnl:GFP and S2-Btl:Cherry cells alone did not show homophilic cell-cell adhesion, but, when co-cultured, S2-Bnl:GFP cells selectively trans-paired with S2-Btl:Cherry by forming trans-synaptic receptor-ligand co-clusters (Fig. 3g–h, Supplementary Fig. 2h, and Supplementary Movie 3). Moreover, the binding of Btl:Cherry and Bnl:GFP induced a reciprocally polarized congregation of the receptors and ligands at the contact interface of the trans-paired cells. We also observed localized enrichment of cortical f-actin (phalloidin-stained) at the synaptic interface (Supplementary Movies 3–5), similar to what was observed in immunological synapses[35]. These results suggest that the Btl-Bnl interactions can induce CAM-like selective reciprocal contacts and signaling polarity between Bnl-exchanging cells.

Almost all (averaging 98%) of S2-Btl:Cherry cells that were trans-paired to S2-Bnl:GFP cells had nuclear-localized dpERK. In the same co-culture experiment, unpaired (non-adhering) S2-Btl:Cherry lacked nuclear dpERK, indicating inactive FGF/MAPK signaling in these cells (Fig. 3i–k′ and Supplementary Fig. 2i, i′). These unpaired S2-Btl:Cherry cells were similar to either control S2-Btl:Cherry cells or non-transfected S2 cells that rarely had nuclear dpERK (average ~2-5% of cells), (Fig. 3j, j′). Similarly, when S2-sBtl:Cherry cells were cocultured with S2-Bnl:GFP, they did not trans-pair with S2-Bnl:GFP and lacked nuclear dpERK (Fig. 3h, i, n and Supplementary Fig. 2g, j, j′). Therefore, we considered that the Btl-Bnl-mediated trans-pairing of S2 cells is a successful in vitro recapitulation of contact-dependent Btl-Bnl signaling between the ASP and Bnl source[9].

Strikingly, S2-Btl-DN:Cherry cells showed strong selective trans-pairing with S2-Bnl:GFP, similar to the S2-Btl:Cherry control. However, the trans-paired S2-Btl-DN:Cherry did not activate nuclear MAPK signaling due to the lack of its intracellular domains (Fig. 3h, i, l–m′ and Supplementary Fig. 2f, h). Therefore, CAM-like physical interactions of the surface-localized Btl-DN (or Btl) and Bnl were sufficient to induce bidirectional contact matchmaking between the Bnl exchanging cells.

**Bnl is tethered to the source cell surface by a GPI anchor**. However, to drive heterophilic CAM-like bidirectional recognition for synapse, Bnl needs to be tightly associated with the source cell membrane. The source surface localization of Bnl was known to be critical for its dispersion and functions[36,37]. Moreover, Bnl is likely to be a membrane-associated protein[38], despite its ability to disperse over long range[9]. How might a secreted protein be associated exclusively on the source cell surface, and be both inhibited and activated for dispersal? A probable mechanism emerged while exploring post-translational Bnl modifications during its intracellular trafficking[36]. We knew that a small N-terminal portion (residue 1-164) upstream of the central 'FGF domain' of Bnl is cleaved off in the source cell Golgi by Furin1 to facilitate polarized trafficking of the remaining C-terminal signaling portion of Bnl to the basal side of the source cell (Fig. 4a; ref. [36]). When cells expressed a Furin-sensor HA$_1$Bnl:GFP$_3$ construct with HA (site 1) and GFP (site 3) flanking the Furin cleavage site, the cleaved HA-tagged portion was retained in the Golgi, and the truncated Bnl:GFP$_3$ fragment was externalized for dispersal[36]. Therefore, we hypothesized that cells expressing a triple-tagged HA$_1$Bnl:GFP$_3$Cherry$_c$ construct with a C-terminal mCherry fusion (Fig. 4a) would externalize a truncated Bnl:GFP$_3$Cherry$_c$ portion marked with both GFP and mCherry.

However, when we expressed HA$_1$Bnl:GFP$_3$Cherry$_c$ (hereafter called Bnl:GFP$_3$Cherry$_c$) in S2 cells, GFP and mCherry tags were separated. Importantly, while the Bnl:GFP$_3$ portion was localized on the cell surface (detected with αGFP$^{ex}$ immunostaining), the C-terminal mCherry remained intracellular (Fig. 4b–b‴). The C-terminal mCherry tag did not alter the predicted topology and

physicochemical properties of Bnl (see Supplementary information). In fact, when Bnl:GFP$_3$Cherry$_c$ was expressed in the wing disc source under bnl-Gal4, the mCherry-tag was retained in source cells, and the Bnl:GFP$_3$ portion was efficiently delivered to the ASP (Fig. 4c). These results indicated an intracellular Bnl:GFP$_3$Cherry$_c$ cleavage, which separated the C-terminal mCherry prior to the secretion of the truncated Bnl:GFP$_3$ portion. Cleavage at multiple locations in the Bnl backbone was consistent with the previously-reported detection of multiple Bnl:GFP bands in Western blots of expressing cell lysates[36].

Bioinformatic analyses revealed that the Bnl C-terminus has a short 20 amino acid hydrophobic tail preceded by a hydrophilic spacer (Fig. 4a, a′). A 15–20 residue long hydrophobic C-terminal tail together with an immediately upstream hydrophilic spacer commonly constitutes the signal sequence (SS) of a pro-GPI-anchored protein (pro-GPI-APs)[25,39–41]. The C-terminal hydrophobic portion of the SS is cleaved off and replaced with a GPI moiety in the endoplasmic reticulum (ER). GPI-APs are trafficked to the cell surface and anchored to the outer leaflet of the plasma membrane by the phosphatidylinositol (PI) portion of the GPI moiety[25,39,40] (Fig. 4d). Because the presence of C-terminal tags does not prevent glypiation of pro-GPI-APs[39], we surmised that GPI-anchoring of Bnl might explain the intracellular cleavage of mCherry from Bnl:GFP$_3$Cherry$_c$ prior to the surface display of its truncated Bnl:GFP$_3$ portion.

We used the phosphoinositide phospholipase C (PI-PLC)-dependent shedding assay to detect Bnl glypiation. Since PI-PLC specifically cleaves the GPI moiety, PI-PLC-dependent shedding of a cell surface protein confirms its GPI anchoring[42]. Using the Gal4/UAS-based expression in S2 cells, we ectopically expressed GFP-GPI[43] (positive control), untagged Bnl (co-transfected with CD8:GFP for detecting transfected cells), Bnl:GFP$_3$, HA$_1$Bnl:GFP$_3$ (henceforth referred as Bnl:GFP), and a palmitoylated cell-surface protein, the constitutively active Drosophila EGF, cSpitz:GFP[44] (negative control). The levels of cell surface proteins were probed by non-permeabilized αGFP$^{ex}$ or αBnl$^{ex}$ (for untagged Bnl) immunostaining, and the ratio of the surface proteins to the total protein per cell was compared between cells and conditions (see Methods). These analyses showed that the PI-PLC treatment specifically removed source-surface Bnl$^{ex}$ and Bnl:GFP$^{ex}$, like control GPI-GFP, but PI-PLC did not remove cSpitz:GFP$^{ex}$ (Fig. 4d, d′, f and Supplementary Fig. 3a–e). Thus, Bnl is a GPI-AP.

An in-silico analysis predicted Bnl-S$^{741}$ as a probable glypiation site (ω-site). To verify if Bnl's C-terminal region acts as a SS, we generated - (i) Bnl:GFPΔC, lacking the C-terminal 40 amino acid residues including the putative ω site; (ii) Bnl:GFPΔC-TM, where the transmembrane domain from the mammalian CD8a was added to the C-terminus of Bnl:GFPΔC; (iii) Bnl:GFP-ω$^m$, Bnl:GFP with mutated ω, ω+1, and ω+2 sites; and (iv) bGFP-GPI, a secreted super-folder GFP (secGFP[9]) fused to Bnl's C-terminal 53 amino acids region (see Methods) (Fig. 4e). Bnl:GFPΔC and Bnl:GFP-ω$^m$ were not localized on the producing cell surface, even without the PI-PLC treatment (Fig. 4d′, g and Supplementary Fig. 3f, h, i). However, when a transmembrane (TM) domain was added to Bnl:GFPΔC (i.e., Bnl:GFPΔC-TM), the protein was surface localized in a PI-PLC-resistant manner (Fig. 4d′–f and Supplementary Fig. 3g, h).

A possibility is that the secreted Bnl binds to GPI-anchored glypicans via the binding sites present within the conserved 'FGF domain' (Fig. 4a)[45]. In this context, PI-PLC treatment of Bnl expressing cells could indirectly remove surface Bnl by acting on glypicans. Indirect PI-PLC-dependent removal of surface Bnl was unlikely, because the addition of Bnl's C-terminal SS to a readily secreted secGFP, which usually was undetectable on the expressing cell surface (Supplementary Fig. 3j, j′), led the

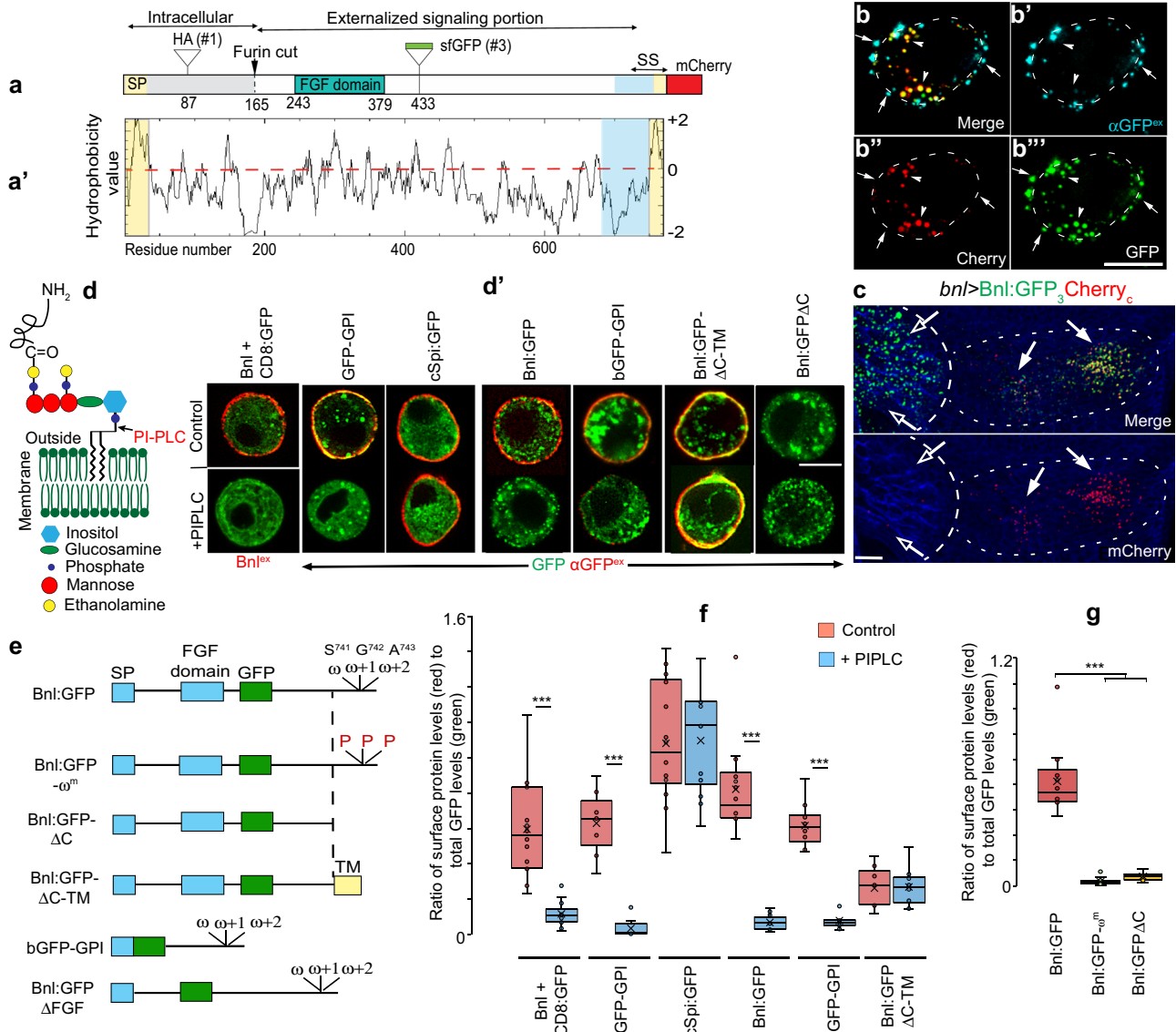

**Fig. 4 Post-translational cleavage and lipidation of Bnl. a, a'** Schematic map of the Bnl protein (**a**) and hydrophobicity plot (**a'**); SP, signal peptide; conserved FGF-domain, HSPG + FGFR binding sites; arrow, furin cleavage site; sites for HA- (site #1), GFP- (site # 3), and mCherry- tags and putative signal sequence (SS; hydrophilic spacer—blue; hydrophobic tail—yellow). **b-b'''** Representative optical sections of S2 cells expressing Bnl:GFP$_3$Cherry$_c$; arrowheads, uncleaved intracellular protein; arrow, cleaved externalized Bnl:GFP$_3$$^{ex}$ portion probed with αGFP$^{ex}$ (blue); dashed line, cell outline; split channels—as indicated. **c** Wing disc expressing Bnl:GFP$_3$Cherry$_c$ under *bnl-Gal4*; arrow, intact protein harboring GFP and mCherry; open arrow, cleaved Bnl:GFP$_3$; long and short dashed line, ASP and *bnl*-source, respectively; blue, αDlg. **d** Illustration of a GPI-AP and the PI-PLC cleavage site. **d'-g** PI-PLC-mediated cell-surface shedding of various constructs expressed from S2 cells. **e** Schematic maps of the Bnl variants used. **d'** red, surface-localized fraction of either GFP-tagged proteins (αGFP$^{ex}$ immunostaining) or Bnl (αBnl$^{ex}$ immunostaining) in expressing cells. **f, g** Box plots comparing the ratio of cell surface (red, αGFP$^{ex}$ or αBnl$^{ex}$ immunofluorescence) to total proteins (GFP fluorescence) in cells with and without PI-PLC treatment; for untagged Bnl, the surface Bnl level was normalized with co-expressed CD8:GFP; box shows the median and 1st and 3rd quartile, and whiskers are minimum and maximum. ***$p < 0.001$ (unpaired two-tailed $t$ test). **f** Number of cells/condition (*n*): Bnl+CD8:GFP control and PI-PLC (14); GFP-GPI—control (9) & PI-PLC (8); cSpi:GFP—control (14) and PI-PLC (14); Bnl:GFP—control (14) & PI-PLC (12); bGFP-GPI—control (13) & PI-PLC (11); Bnl:GFPΔC-TM—control (9) & PI-PLC (10). **g** Number of cells analyzed (*n*): Bnl:GFP (14), Bnl:GFP-ω$^m$ (16), and Bnl:GFPΔC (9). S2 cells: co-transfected with *actin-Gal4* and *UAS-"X"*. Source data are provided as a Source data file. Scale bars: 10 μm.

PI-PLC-sensitive surface localization of the engineered protein (bGFP-GPI) (Fig. 4d'–f). Thus, Bnl's SS is required for glypiation. Secondly, a Bnl:GFP$^{ΔFGF}$ construct, which has the entire Bnl sequence except for the core 'FGF domain' replaced with sfGFP, showed PI-PLC-sensitive surface localization (Fig. 4e and Supplementary Fig. 3k). Thus, PI-PLC can directly cleave the GPI anchor of Bnl.

The GPI-AP signal sequences (including ω-sites) are known to have little sequence conservation, and its extreme C-terminal positioning is not an absolute requirement[41,46]. The Bnl constructs described here were derived from the well-characterized *bnl-PA* isoform[20] that has been used in all previous reports of ectopic Bnl expression. Bnl also has a shorter splice variant (PC) (FlyBase) with altered C-terminal hydrophobicity (Supplementary Fig. 4a–a'''). Therefore, we generated a Bnl:GFP-PC construct (Methods) and expressed it in S2 cells. PI-PLC treatment of S2 cells expressing Bnl:GFP-PC removed the surface-localized Bnl:GFP-PC, indicating its GPI-anchored

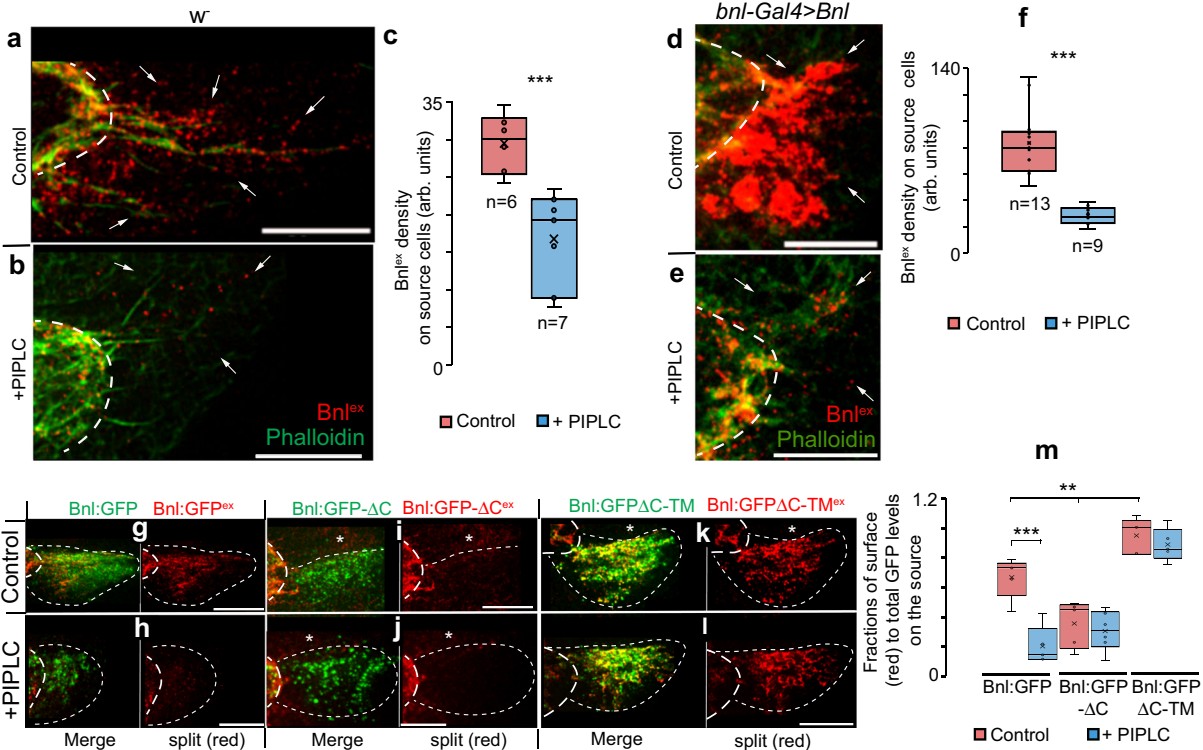

**Fig. 5 Bnl is GPI-anchored to the wing disc source cell surface. a–c** Wing discs ($w^-$ showing surface-localized native Bnl$^{ex}$ (red, αBnl$^{ex}$) levels on the *bnl*-source area (arrows) before (**a**) and after (**b**) PI-PLC treatment; **c** box plots comparing numerical values as indicated, $p = 0.0013$ (**; unpaired two-tailed *t* test). **d–f** Wing discs overexpressing Bnl under *bnl-Gal4* showing surface Bnl$^{ex}$ (red, αBnl$^{ex}$) levels on the *bnl*-source (arrows) before (**d**) and after (**e**) PI-PLC treatment; **f** box plots comparing numerical values as indicated, $p = 0.024 \times 10^{-4}$ (***; unpaired two-tailed *t* test). **g–m** Source surface levels (red, αGFP$^{ex}$) of Bnl:GFP, Bnl:GFPΔC and Bnl:GFPΔC-TM on wing discs when expressed under *bnl-Gal4* before and after PI-PLC; asterisks, non-expressing source-surrounding disc area; dashed line, ASP or *bnl*-source; m, box plots comparing the fraction of surface localized (red, αGFP$^{ex}$) to total protein (probed by GFP) of Bnl:GFP variants on the source before and after PI-PLC treatment; $n = 5$ biologically independent samples for each; $p$ values—$p < 0.05$ (*) and $p < 0.01$ (**) (one-way ANOVA followed by Tukey HSD test). All box plots: box shows the median, 1st quartile, 3rd quartile, and whiskers show minimum and maximum. Source data are provided as a Source data file. Scale bars, 30 μm.

display (Supplementary Fig. 4b–g). Thus, both Bnl-PC and Bnl-PA isoforms are glypiated, but strikingly, with two distinct signal sequences.

Next, to detect Bnl's GPI-anchoring in vivo, we developed a PI-PLC assay on live ex vivo cultured wing discs (see Methods). First, we detected native extracellular Bnl by non-permeabilized immunostaining of ex vivo cultured $w^-$ wing discs with a Bnl antibody that detects all Bnl isoforms (Supplementary Fig. 4a, a′). Bnl$^{ex}$ that normally was asymmetrically enriched on the disc source was significantly reduced with PI-PLC treatment (Fig. 5a–c). When Bnl, Bnl:GFP, Bnl:GFPΔC-TM, Bnl:GFPΔC, and Bnl:GFP-ω$^m$ constructs were expressed under *bnl-Gal4*, PI-PLC treatment significantly reduced Bnl$^{ex}$ and Bnl:GFP$^{ex}$ on the source surface, but not Bnl:GFPΔC-TM$^{ex}$ (Fig. 5d–m). As observed in S2 cells, Bnl:GFPΔC$^{ex}$ and Bnl:GFP$^{ex}$-ω$^m$ puncta were not detected on source cells irrespective of the PI-PLC treatment (Fig. 5i, j, m and Supplementary Fig. 5a, b).

Although Bnl:GFPΔC$^{ex}$ was absent from the source membrane, it was broadly spread through the extracellular disc areas surrounding the source and was also received by the ASP, suggesting that the protein was readily secreted and randomly dispersed from the source (Supplementary Fig. 5a, a′). Externalized Bnl:GFPΔC$^{ex}$ contains the conserved glypican binding FGF domain, yet it was absent on the source surface, indicating that the secreted Bnl:GFPΔC$^{ex}$ was not restricted on the source surface by glypican binding. In contrast to Bnl:GFPΔC$^{ex}$, Bnl:GFP-ω$^m$ showed severely reduced externalization (Supplementary Fig. 5b).

This was consistent with previous reports of ER retention of the uncleaved pro-GPI-APs, in contrast to the normal trafficking of the same protein with deleted SS[47,48]. These results indicated that Bnl is cleaved at its C terminus and added with a GPI moiety, which both facilitated Bnl externalization and inhibited its free secretion.

**GPI-anchored Bnl promotes target-specific cytoneme contacts.** To test if GPI anchoring is required for Bnl's CAM-like activity, we employed the cell culture-based assay. When Btl:Cherry was co-transfected with either Bnl:GFP, Bnl:GFPΔC, or Bnl:GFPΔC-TM in the same cells, almost all (at least 90%) of the cells expressing both ligands and receptors had nuclear dpERK (Supplementary Fig. 5c and Supplementary Table 3a). Thus, all Bnl variants could efficiently activate Btl:Cherry when co-expressed in the same cell. When co-cultured, S2-Bnl:GFP (control) and S2-Btl:Cherry were trans-paired with each other, and the trans-paired S2-Btl:Cherry had nuclear dpERK (Fig. 3). In contrast, co-cultured S2-Bnl:GFPΔC and S2-Btl:Cherry cells were rarely trans-paired (only ~1% frequency of juxtaposition) (Fig. 6a, c, Supplementary Fig. 5d, and Supplementary Table 3b). Even when S2-Btl:Cherry cells were juxtaposed to S2-Bnl:GFPΔC, the contact interface lacked polarized Btl-Bnl co-clusters. Moreover, almost 85% of S2-Btl:Cherry cells that were nearby to the S2-Bnl:GFPΔC source lacked dpERK (Fig. 6a, Supplementary Fig. 5d, and Supplementary Table 3b). A few dpERK-positive S2-Btl:Cherry cells that were found, had unpredictable random locations relative to the S2-Bnl:GFPΔC.

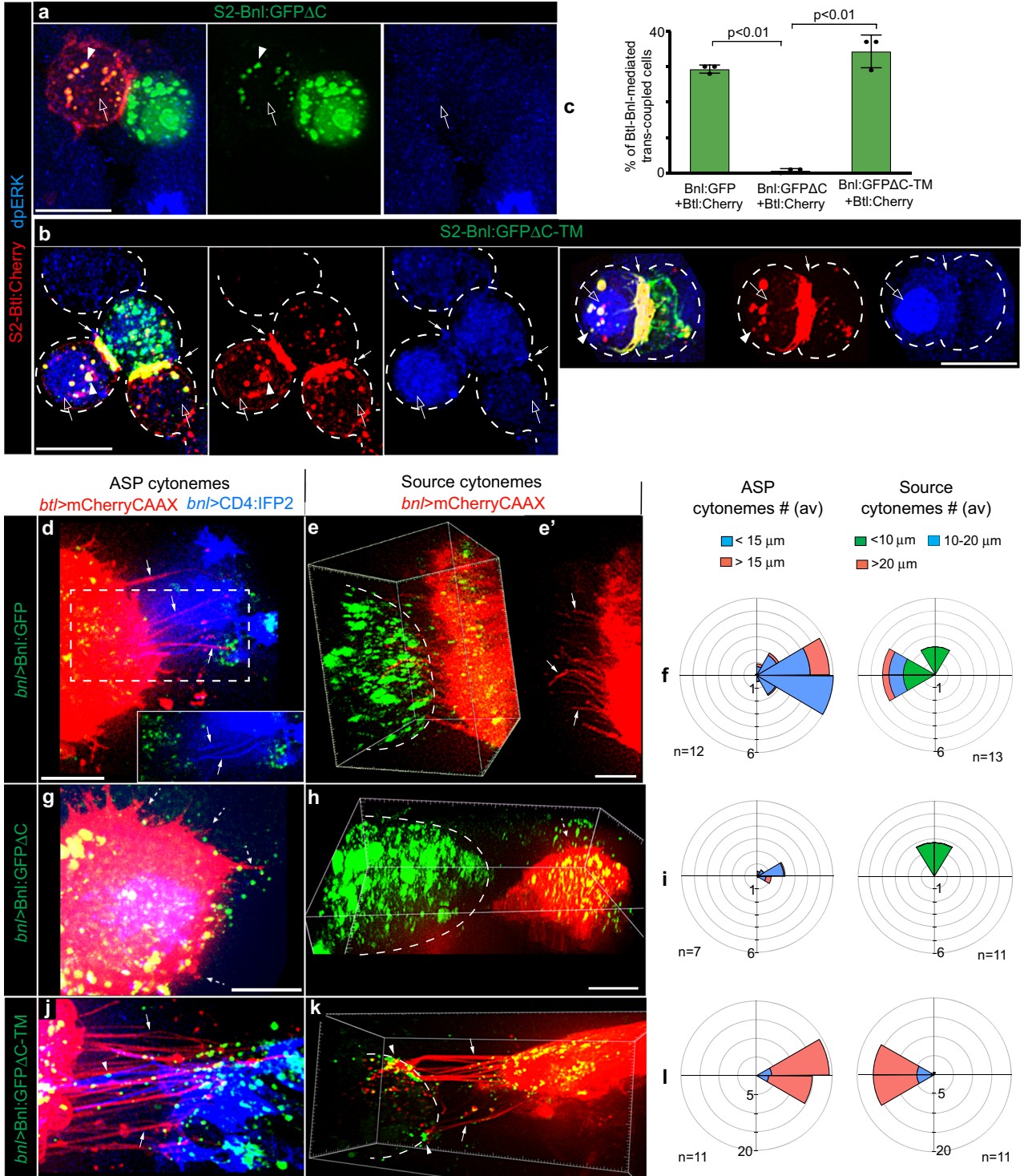

**Fig. 6 GPI-anchored Bnl acts as a CAM. a–c** CAM-like polarized trans-pairing of S2-Btl:Cherry with either S2-Bnl:GFPΔC-TM (**b**, **c**) or S2-Bnl:GFP (**c**; see Fig. 3g, k) but not with S2-Bnl:GFPΔC (**a**, **c**); arrow, polarized receptor-ligand co-clusters at the synaptic site; open arrow, nucleus in source proximal S2-Btl:Cherry (**a**) and trans-paired S2-Btl:Cherry (**b**); arrowhead, Bnl:GFP signal uptake into the juxtaposed or trans-paired S2-Btl:Cherry cell; blue, nuclear dpERK (αdpERK); **c** bar graphs comparing the mean (±SD) frequency of receptor-ligand trans-pairing for GPI-modified and non-GPI modified Bnl:GFP variants from three independent experiments (see Methods); p values were obtained by one-way ANOVA followed by Tukey HSD test; total GFP-positive + Cherry-positive cells analyzed: 1916 (Bnl:GFP + Btl:Cherry), 2664 (Bnl:GFPΔC + Btl:Cherry), 2192 (Bnl:GFPΔC-TM + Btl:Cherry).
**d–l** Comparison of Bnl:GFP (control), Bnl:GFPΔC, or Bnl:GFPΔC-TM signals for induction of reciprocal polarity of ASP and source cytonemes (arrows), when expressed from the disc source; genotypes, as indicated; **d** inset, ROI (dashed box) in green and blue channels; **d**, **g**, **j** extended Z projection; **e**, **e'**, **h**, **k** 3D-rendered views; dashed lines, ASP; **g**, **h** dashed arrows, randomly oriented short cytonemes; **f**, **i**, **l** R-plots comparing numbers, length, and directionality of ASP and source cytonemes as indicated; n = number of discs analyzed (also see Supplementary Fig. 6a–g). Source data are provided as a Source data file. All panels except **a**, **b**, live imaging. Scale bars, 10 μm (**a**, **b**), 20 μm (**d–k**).

In contrast to S2-Bnl:GFPΔC, S2-Bnl:GFPΔC-TM cells selectively trans-adhered to S2-Btl:Cherry as efficiently as S2-Bnl:GFP by forming trans-synaptic receptor-ligand co-clusters (Fig. 6b, c and Supplementary Table 3b). Binding of Bnl:GFPΔC-TM and Btl:Cherry also induced MAPK signaling in the adhering Btl:Cherry-expressing cells, but at a lower frequency than the control S2-Bnl:GFP::S2-Btl:Cherry interactions (Fig. 6b and Supplementary Table 3b). Notably, MAPK signaling was activated only in those trans-paired S2-Btl:Cherry cells that had high numbers of internalized Bnl:GFPΔC-TM puncta (Fig. 6b). It is possible that Btl:Cherry-bound Bnl:GFPΔC-TM could somehow be internalized into adhering recipient cells and the signal internalization was required to activate MAPK signaling. Irrespective of the activation of MAPK pathway, the trans-synaptic binding of Bnl:GFPΔC-TM and Btl:Cherry was sufficient to induce selective cell-cell contacts and reciprocal polarity of Btl and Bnl localization at the contact sites. Thus, membrane tethering of Bnl is required for the CAM-like Btl-Bnl interactions, which, in turn, produce bidirectional cell-cell contacts.

To test if CAM-like Btl::Bnl interactions occur through cytonemes in vivo, we compared how GPI-modified (Bnl:GFP) and non-GPI-modified Bnl:GFP variants affect source and ASP cytonemes. Despite the Bnl:GFP overexpression, both ASP and source cytonemes retained their reciprocal polarity toward each other (Fig. 6d–f and Supplementary Fig. 6g). An increase in extension-retraction rates of ASP cytonemes in this condition suggested an increase in signaling activity in the ASP (Supplementary Movie 6; Supplementary Table 1). In contrast, over-expressed Bnl:GFPΔC significantly suppressed the formation of polarized cytonemes from both source and ASP cells (Fig. 6g–i and Supplementary Fig. 6a, b, g). Short cytonemes, when detectable, lacked any directional bias and Bnl:GFPΔC localization.

Importantly, unlike Bnl:GFPΔC, Bnl:GFPΔC-TM induced both ASP and source cells to extend large numbers of long polarized cytonemes that were adhered to each other (Fig. 6j–l, Supplementary Fig. 6c–h, and Supplementary Movies 7–9). Bnl:GFPΔC-TM puncta populated at multiple inter-cytoneme contact interfaces (Figs. 6j and 7a–a″). To visualize the CAM-like Bnl-Btl binding at the inter-cytoneme contacts, we expressed Bnl:GFPΔC-TM from the CD4:IFP$_2$-marked wing disc source in $btl:cherry^{endo}$ larvae. These larvae expressed endogenous Btl:Cherry$^{endo}$ in the ASP. Btl:Cherry$^{endo}$ puncta on the ASP cytonemes were co-clustered with Bnl:GFPΔC-TM puncta at multiple contact sites along the length of the source and recipient cytonemes (Fig. 7b, c). Bnl:GFPΔC-TM-exchanging cytonemes showed higher stability and longer contact lifetime than WT or Bnl:GFP-exchanging cytonemes (Fig. 7d, e and Supplementary Fig. 6h, Supplementary Movies 10 and 11, and Supplementary Table 1). The increased stability of inter-cytoneme adhesion might account for the higher intensity of bidirectional responses with Bnl:GFPΔC-TM than with Bnl:GFP. These in vivo results showed that the bidirectional responses are produced directly by the CAM-like Btl:Bnl binding.

ASP cells can orient cytonemes toward an ectopic Bnl-expressing clone[19] and Bnl signaling feedbacks are known to promote the source-specific cytoneme polarity[9]. To further verify if the contact-dependent Btl:Bnl binding can also induce cytoneme-polarizing responses in the source, we produced randomly-localized mCherryCAAX-marked wing disc clones that expressed Bnl:GFPΔC-TM (Fig. 7f–j). Clones in the wing disc pouch that occurred far away from the ASP were unable to establish contact with the ASP. These clones had only short, randomly oriented signal-containing cytonemes (Fig. 7f–h). In contrast, ASP-proximal clones extended long polarized cytonemes and established contacts with the ASP (Fig. 7f, i, j).

Similarly, when randomly-localized Btl:GFP-expressing clones were induced in the disc, ectopic Btl-expressing cells and the mCherryCAAX-marked source cells reciprocally polarized cytonemes toward each other (Fig. 7k–m′). These results were consistent with the activation of a retrograde response in the Bnl-source induced by Btl-Bnl binding.

**GPI anchoring promotes ASP-specific Bnl release.** Although Bnl:GFPΔC-TM induced strong bidirectional responses that were manifested in cytoneme polarity and inter-cytoneme contacts, Bnl:GFPΔC-TM-exchanging cytonemes had a significantly longer lifetime than WT or Bnl:GFP-exchanging cytonemes (Fig. 7d, e, Supplementary Fig. 6h, and Supplementary Table 1). Moreover, unlike Bnl:GFP, Bnl:GFPΔC-TM puncta were often abnormally internalized into the ASP with the colocalized source cell membrane, indicating a defect in the release of the TM-tethered signal from the source cell membrane (Fig. 8a, d, e, Supplementary Fig. 7a–c′, and Supplementary Movie 14). When both source and ASP cells were simultaneously marked and imaged in time-lapse, Bnl:GFPΔC-TM-exchanging cytonemes appeared to resist contact dissociation, leading to cytoneme breakage and absorption of the source membrane in the ASP (Supplementary Movie 11). These results suggested that unlike the TM anchor, the GPI anchor can promote Bnl release and cytoneme contact disassembly.

To investigate whether GPI-anchoring of Bnl facilitates its target-specific release, we compared the spatial distribution of GPI-modified (Bnl:GFP) and non-GPI-modified Bnl:GFP constructs expressed from the mCherryCAAX-marked wing disc Bnl source. As observed before[9,36], despite overexpression in the disc bnl-source, Bnl:GFP puncta were exclusively transferred from the disc source to the ASP (Fig. 8a and Supplementary Movie 12). In contrast, two ΔC variants (Bnl:GFPΔC and Bnl:GFPΔC$_{168}$) showed dispersion in the non-specific disc areas surrounding the source (Fig. 8b, c and Supplementary Movies 13 and 14). Importantly, the corresponding Bnl:GFPΔC-TM and Bnl:GFPΔC$_{168}$-TM variants regained the exclusive ASP-specific distribution, but their range of distribution was restricted only to the ASP tip, indicating reduced levels of signal exchange (Fig. 8d, e, Supplementary Fig. 7a–c′, and Supplementary Movies 15 and 16). These results suggested that GPI anchoring might be required for both inhibiting free Bnl secretion/dispersion and facilitating target-specific contact-dependent Bnl release.

To better understand the dual roles of GPI anchoring, we compared the ASP-specific and non-specific spreading of Bnl:GFP variants over time in live ex vivo cultured discs (see Methods). To accurately estimate the levels of signal uptake in the ASP, we took advantage of the dual-tagged Furin-sensors - HA$_1$Bnl:GFP$_3$, HA$_1$Bnl:GFP$_3$ΔC-TM, and HA$_1$Bnl:GFP$_3$ΔC (Fig. 8f–s). As expected, in ex-vivo cultured discs, the N-terminal HA-tagged portions (αHA-probed) of all three constructs were cleaved in the source, and only their truncated Bnl:GFP$_3$ portions were transferred to the ASP (Fig. 8f, g, j, m). However, when Furin inhibitors were added to the culture media, uncleaved signals (αHA-probed) were received by the ASP[36]. Therefore, the fraction of the HA-probed uncleaved signal (HA-probed) relative to the total levels of the signal (i.e., GFP-probed pre-existing Bnl:GFP$_3$ + HA$_1$Bnl:GFP$_3$) accumulated in the ASP during a Furin-inhibited period provided a semi-quantitative estimate of the rate of signal uptake in the ASP (Fig. 8f′).

As observed before[36], the levels of HA$_1$Bnl:GFP$_3$ (control) uptake in the ASP gradually increased with the increasing duration of the culture. In comparison, the levels of HA$_1$Bnl:GFP$_3$ΔC-TM and HA$_1$Bnl:GFP$_3$ΔC in the ASP did not change dramatically, indicating a slow rate of ASP-specific

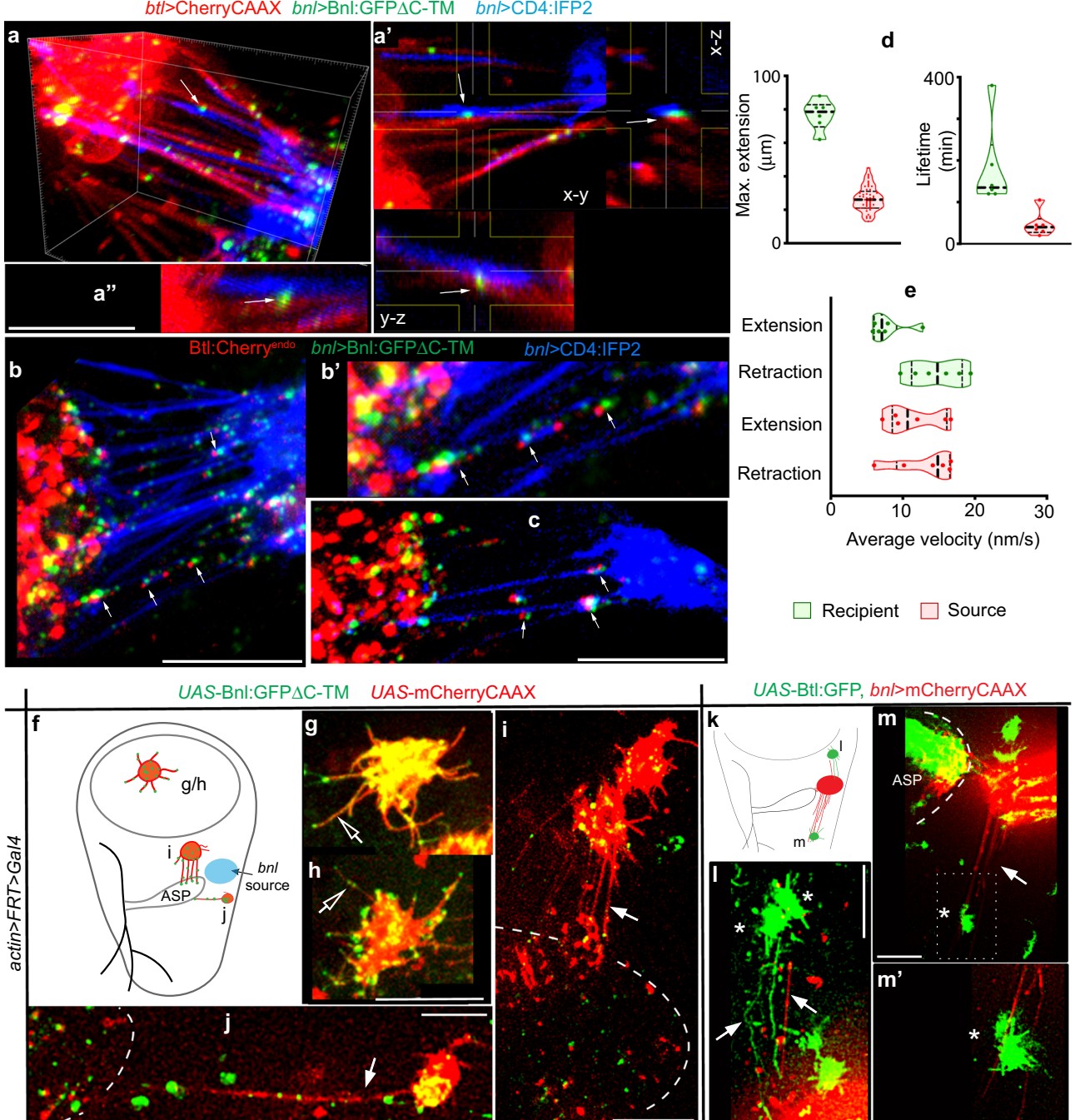

**Fig. 7 CAM-like Btl-Bnl binding induces bidirectional contacts. a–a″** 3D projection (**a**, **a″**) and orthogonal view (a') of wing discs showing cytoneme-mediated adhesion between the ASP (*btl > CherryCAAX*) and the Bnl:GFPΔC-TM-expressing disc *bnl*-source (blue, *bnl > CD4:IFP2*,Bnl:GFPΔC-TM); arrows, Bnl:GFPΔC-TM-localized contact sites. **b**, **c** 3D-projected images of Bnl:GFPΔC-TM-expressing wing disc *bnl*-source (blue, *bnl > CD4:IFP2*) in *btl:cherry^endo* knock-in background (ASP expressed endogenous Btl:Cherry[9]), showing trans-synaptic co-clustering of Btl:Cherry^endo and Bnl:GFPΔC-TM puncta (arrows) at the inter-cytoneme contact sites. **d**, **e** Violin plots showing dynamics of Bnl:GFPΔC-TM-exchanging recipient and source cytonemes (compared to control in Fig. 1l–n; also see Supplementary Table 1 for statistics); in violin plots, black dotted lines show the median and 25th and 75th percentiles. **f–j** Randomly localized wing disc clones (mCherryCAAX-marked) expressing Bnl:GFPΔC-TM, and their cytoneme-dependent interactions (arrow) with the ASP (dashed line); **f** an approximate map of clones in g-j; open arrows, randomly oriented cytonemes. **k–m'** Randomly localized wing disc clones expressing Btl:GFP (*), and their cytoneme-dependent polarized interactions (arrow) with mCherryCAAX-marked *bnl*-source (only the basal-most section of disc columnar cells shown); genotypes: see "Methods"; **m'**, zoomed in ROI of **m**. All panels, live imaging. Source data are provided as a Source data file. Scale bars, 20 μm.

transfer of these variants (Fig. 8g–p). Notably, even after 5 h of culture, HA₁Bnl:GFP₃ dispersed exclusively to the ASP (Fig. 8q). In contrast, within a 5 h of the incubation period, HA₁Bnl:GFP₃ΔC was randomly localized in the source-surrounding disc areas, but was barely received by the ASP from the same disc (Fig. 8j–l, q–s). These results suggested that GPI anchoring is required to both inhibit free Bnl secretion/dispersion and activate its directed contact-dependent release.

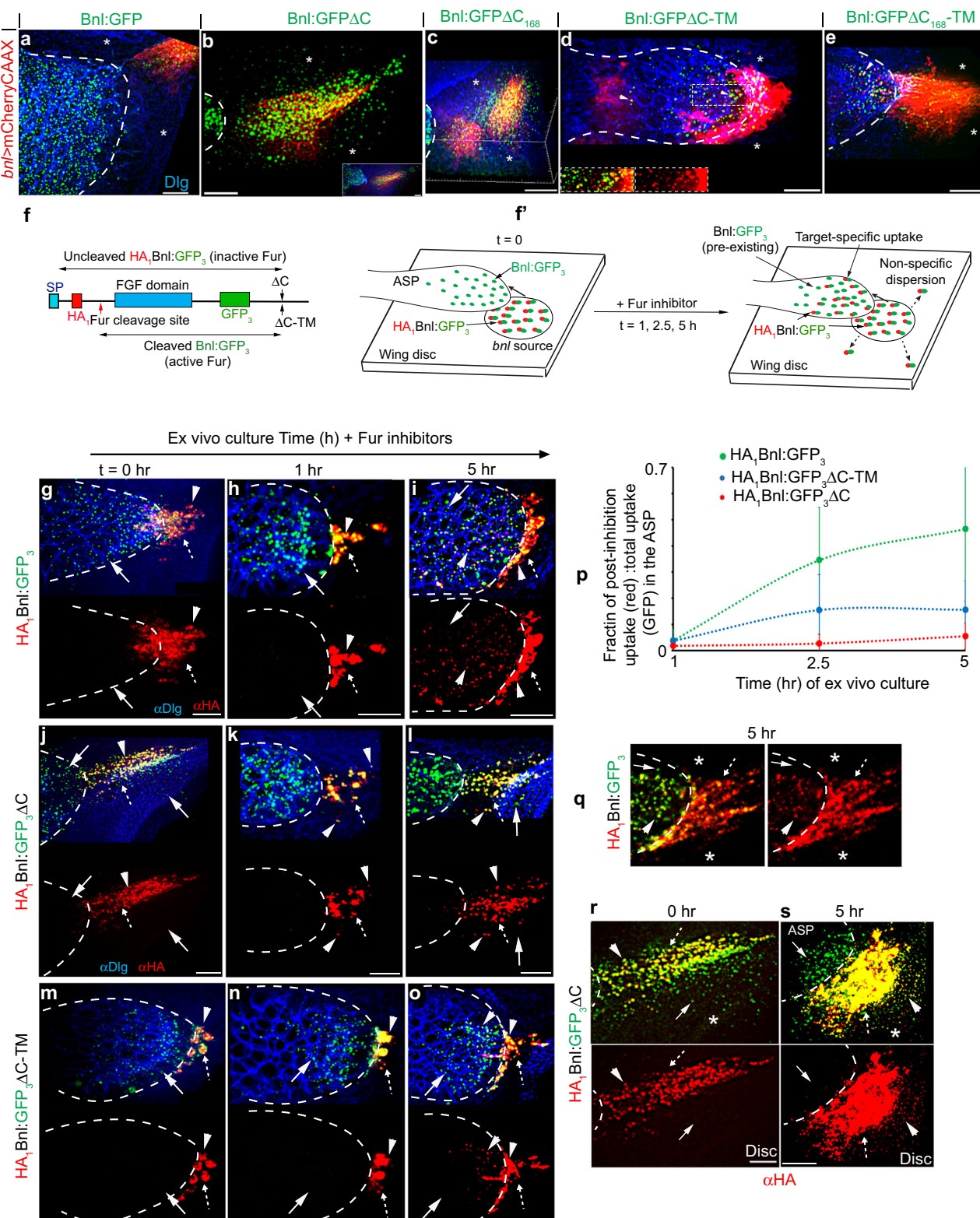

**GPI-anchored Bnl directs context-specific signaling**. To investigate if GPI anchoring and contact-dependent release are important for Bnl signaling, we generated small (2-3 cell) gain-of-function (GOF) clones expressing GPI-modified and non-GPI-modified Bnl:GFP variants directly within the ASP epithelium[9]. To distinguish the ectopic Bnl:GFP signaling from the endogenous Bnl signaling, we analyzed clones within the ASP stalk and transverse connective (TC), which lack Bnl uptake from the

original disc source and MAPK signaling[9] (Supplementary Fig. 7d). In consistence with earlier reports[9], all cells within 3 cell diameter area surrounding a Bnl:GFP GOF clone received Bnl:GFP, and all Bnl:GFP-receiving cells also induced dpERK (Fig. 9a, a', d and Supplementary Fig. 7g). In comparison, Bnl:GFPΔC was received by many cells surrounding its clonal source, but only a few randomly located Bnl:GFPΔC-receiving cells induced dpERK (Fig. 9b, b', d and Supplementary Fig. 7e, g).

**Fig. 8 GPI anchoring promotes target-specific Bnl release. a–e** Distribution patterns of Bnl:GFP variants expressed from the mCherry-marked *bnl* source (*UAS-mCherryCAAX;bnl-Gal4* x *UAS-"X"*); extended Z-projections from the basal disc area and disc ASP interface shown (for 3D projections, see Supplementary Movies 12–16); **b** inset, extended z-stack includes ASP and disc basal area; **d** Inset, split-colors of ROI (box); arrowhead, source cell membrane containing Bnl:GFPΔC-TM puncta in the ASP. **f** Schematic map of a Furin-sensor HA₁Bnl:GFP₃; red arrow, Furin cleavage site; black arrow, ΔC or TM modification sites; double-sided arrows, uncleaved HA₁Bnl:GFP₃ and cleaved Bnl:GFP₃ that were transferred to the ASP in the presence and absence of Fur inhibition, respectively. **f'** Illustration depicting experimental strategy to detect ASP-specific dispersal rate of uncleaved signals in Furin-inhibited media. **g–p** Comparison of the ASP-specific uptake of HA₁Bnl:GFP₃, HA₁Bnl:GFP₃ΔC-TM, and HA₁Bnl:GFP₃ΔC (yellow puncta: red-αHA + green-GFP) from the disc source. **p** Graphs comparing the levels of uptake over time (see "Methods"); for each time point, values represent the mean ± SD from multiple biologically independent samples; number (n) of tissues analyzed per time point, HA₁Bnl:GFP₃: $n = 12$ (1 h), 11 (2.5 h), 10 (5 h); HA₁Bnl:GFP₃ΔC-TM: $n = 16$ (1 h), 18 (2.5 h), 15 (5 h); HA₁Bnl:GFP₃ΔC: $n = 8$ (1 h), 14 (2.5 h), 11 (5 h); $p < 0.01$ for HA₁Bnl:GFP₃ vs HA₁Bnl:GFP₃ΔC-TM or HA₁Bnl:GFP₃ΔC at 2.5 h and 5 h (one-way ANOVA followed by Tukey HSD test). **q–s** Comparison of HA₁Bnl:GFP₃ (**q**) and HA₁Bnl:GFP₃ΔC (**r, s**) for their ability of ASP-specific dispersion over time; all panels: dashed outline, ASP; arrow, cleaved Bnl:GFP₃; arrowhead, uncleaved signal; dashed arrow, source cells; αDlg, cell outlines; asterisk, non-specific disc areas; **g–s** only merged and corresponding red channels were shown. Source data are provided as a Source data file. Scale bars, 20 μm.

Apparently, the normal spatial correlation between signal dispersion and signaling was lost with Bnl:GFPΔC. The coordination between the signal dispersion and signaling was regained with Bnl:GFPΔC-TM, but Bnl:GFPΔC-TM activity was restricted to only a few source-juxtaposed ASP cells (Fig. 9c, d and Supplementary Fig. 7f, g). Similarly, when either Bnl:GFP, Bnl:GFPΔC, or Bnl:GFPΔC-TM was overexpressed from the disc *bnl*-source, unlike Bnl:GFP or Bnl:GFPΔC-TM, a significant number of Bnl:GFPΔC-receiving ASP cells lacked nuclear MAPK signaling (Fig. 9d and Supplementary Fig. 7h–l). These results suggested that GPI anchoring and contact-dependent Bnl release are required for the normal coordination between signal dispersion and interpretation.

Bnl was also known to chemoattract tracheal migration toward its source[49,50]. To assess the morphogenetic potency of Bnl variants, we examined their ability to chemoattract tracheal branches to an ectopic expressing source, such as the larval salivary gland, a non-essential, trachea-free organ, which normally does not express *bnl*[36,50]. *bnl-Gal4* was reported to be non-specifically expressed in the salivary glands, and *bnl-Gal4*-driven Bnl expression in the salivary glands induced tracheal invasion into this trachea-free organ[36]. Therefore, we expressed comparable levels of Bnl:GFP, Bnl:GFPΔC, and Bnl:GFPΔC-TM in salivary glands under *bnl-Gal4* (see Methods and Supplementary Information). Strikingly, both Bnl:GFP and Bnl:GFPΔC-TM induced extensive tracheal invasion and branching into the expressing salivary gland, but Bnl:GFPΔC did not (Fig. 9e–h). The unavailability of Bnl:GFPΔC on its source surface could reduce its ability to guide tracheal invasion into the source. Indeed, although all three Bnl variants were expressed at an equivalent level under *bnl-Gal4*, the level of extra-cellular Bnl:GFPΔC on the salivary gland surface was significantly less than that of Bnl:GFP or Bnl:GFPΔC-TM (Fig. 9i–l). These results suggested that Bnl retention on the source surface is critical for its morphogenetic potency.

## Discussion

This study uncovered an elegant program of reciprocal inter-organ communication that is encoded by the lipid-modification of FGF/Bnl and orchestrated by cytoneme-mediated contact-dependent signaling. We characterized Bnl as a lipid-modified FGF and showed how lipidation enables Bnl to self-regulate its tissue-specific dispersion and interpretation by modulating its cytoneme-mediated signaling. These findings also provide insights into how cytonemes find targets, establish contacts, and exchange signals at their contact sites, and how Bnl might inform cells where they are, what they should do, and when.

We discovered that Bnl is GPI-anchored to the source cell surface, and this modification endows the signal with an ability to

act as a local CAM and a long-range morphogen. As summarized in Fig. 10a, we found that the GPI-anchored Bnl controls cytonemes by directing at least three cellular functions - target selection and contact formation, target-specific signal release, and feedback reinforcement of these events. Bnl source and recipient cells extend cytonemes to present Bnl and Btl on their surfaces and reciprocally recognize and contact each other over distance via heterophilic CAM-like Btl-Bnl binding and bidirectional interactions. These results explain how cytonemes might find directions and recognize a specific signaling target for contacts. The Btl-Bnl-dependent matchmaking of source and recipient cytonemes is reminiscent of CAM-dependent neuronal synaptogenesis[33]. Filopodia-mediated bidirectional matchmaking for synapse has been reported in *Drosophila* neuromuscular junctions[32].

Traditionally, secreted signals are presumed to activate responses unidirectionally, only in recipient cells. In this general model, signals themselves do not physically shape cells/tissues, but control gene transcription required for morphogenesis. However, our results indicate that the lipidated Bnl can directly shape cells/cytonemes by binding to Btl and by inducing a CAM-like contact-dependent bidirectional response in both the source and recipient cells. Btl-Bnl binding can induce at least two interdependent responses in the source. First, it induces polarized Bnl congregation/delivery at the contact sites (Figs. 3f, g, 6b, and 7a–b' and Supplementary Movie 3). Second, it polarizes source cytonemes toward the ASP (Figs. 6d–l and 7f–m' and Supplementary Movies 7–10).

The membrane localization and contact-dependent binding of both Bnl and Btl are required to produce responses in both Bnl-source and -recipient cells (Fig. 10a–c). For instance, BnlΔC, which was not retained on the source cell surface, was incapable of inducing a bidirectional response, even when Btl- and BnlΔC-expressing cells were juxtaposed (Figs. 6a, g–i and 10b). The same BnlΔC, when membrane-tethered to its source via a TM domain (BnlΔC-TM), could induce bidirectional responses via cell-cell contacts (Figs. 6b, c, i–l and 10c). Importantly, membrane-tethered distribution of Btl-DN, which can bind to Bnl but cannot activate MAPK signaling, could efficiently promote bidirectional cytoneme–cytoneme or cell–cell contacts. Unlike Btl or Btl-DN, a freely secreted sBtl (lacked its TM domain), failed to induce reciprocal responses (Fig. 3g–n). All these results indicate that the polarized cytoneme-forming responses in source and recipient cells depend on the CAM-like physical Btl-Bnl interactions.

Thus, conceptually, like other CAMs[22–28], GPI-anchored Bnl can serve as both a ligand and a receptor for Btl and, upon binding to Btl, can transmit information inside-out and outside-in across the source cell membrane. However, the intracellular pathway downstream of the Btl-Bnl binding that instructs the

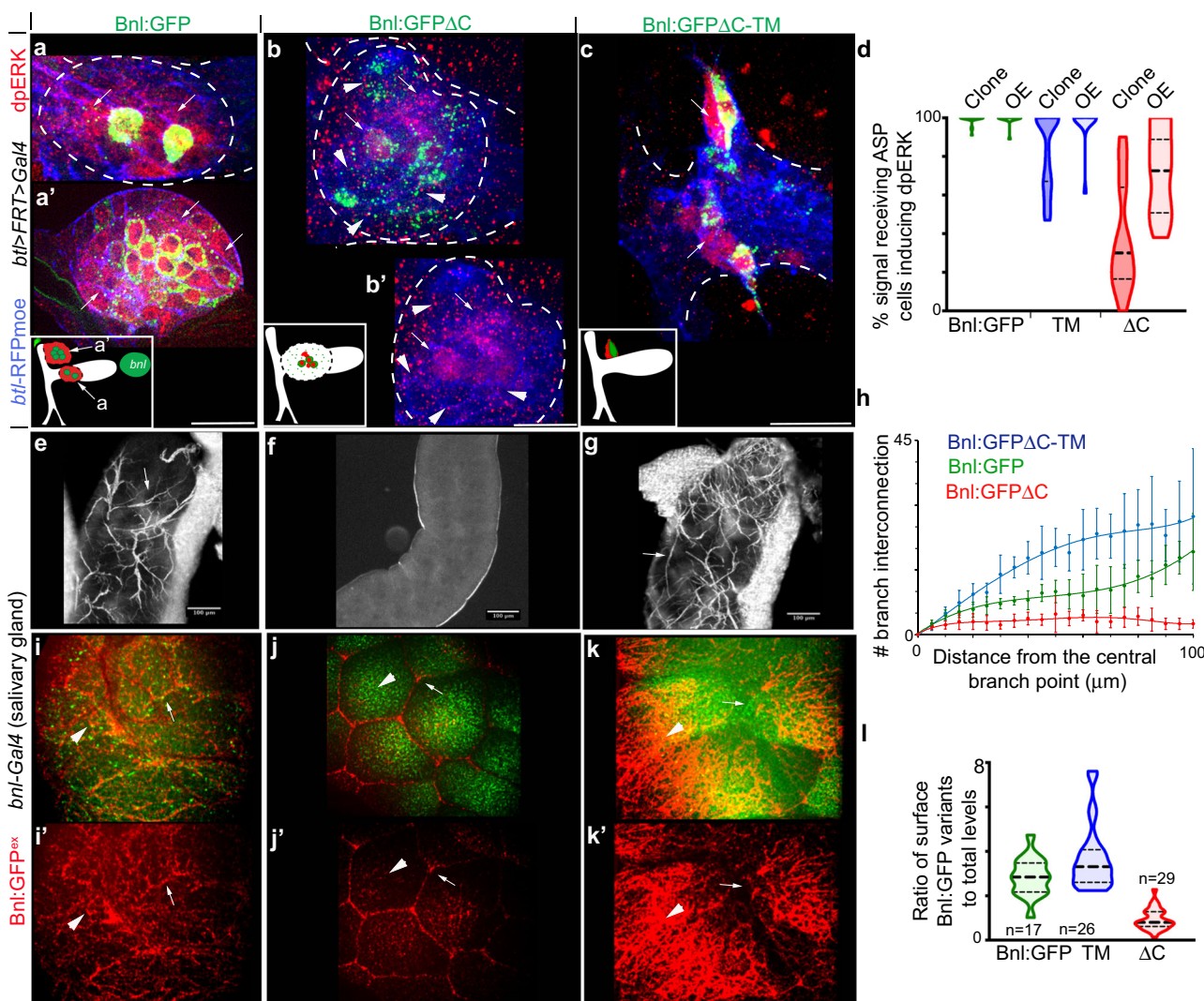

**Fig. 9 GPI-anchored Bnl promotes recipient-specific patterning. a–c** Comparison of non-autonomous signaling (dpERK, red; arrows) patterns of Bnl:GFP (control), Bnl:GFPΔC, and Bnl:GFPΔC-TM, expressed from ectopic GOF clones within the ASP stalk and TC; approximate clone locations and dpERK patterns indicated in inset; arrow and arrowhead, signal-recipient cells with and without nuclear dpERK, respectively; genotype: *hsFlp; btlenh > y* +>Gal4,btlenh-mRFPmoe x UAS-"X". **d** Violin plots comparing the percentage of signal-receiving ASP cells with nuclear dpERK; clone, clonal expression; OE, overexpression (*bnlGa4 x UAS-X*; see Supplementary Fig. 7h–k); for OE: p < 0.01 for ΔC (n = 16) vs either Bnl:GFP (n = 17) or TM (n = 13) (one-way ANOVA followed by Tukey HSD test); for clonal analyses: see Supplementary Fig. 7g for statistics. **e–h** Levels of tracheal branch invasion (arrows) into larval salivary glands ectopically expressing either Bnl:GFP (control), TM, or ΔC under *bnl-Gal4* (*bnl-Gal4 X UAS-X*); **e–g** brightfield images, ×10 magnification. **h** Graphs showing mean frequency (±SD) of terminal branching (see Methods); n, number of tissues analyzed per genotype: 5 (TM), 6 (Bnl:GFP), 4 (ΔC). **i–l** Levels of Bnl:GFP, TM, and ΔC displayed on the basal surface of the expressing salivary glands; arrow, cell junctions. **l** Violin plots comparing the fraction of surface-displayed signals (red/arrowhead, αGFPex) to total protein expressed (GFP level); n, number of salivary glands; p < 0.05: Bnl:GFP vs. TM, p < 0.01: ΔC vs. either Bnl:GFP or TM (one-way ANOVA followed by Tukey HSD test). Violin plots, black dotted lines show the median as well as 25th and 75th percentiles. Source data are provided as a Source data file. Scale bars, 30 μm; 100 μm (**e–g**); 20 μm (**i–k'**).

source cells to polarize cytonemes toward the ASP is unknown. Based on clues from results, we speculate that the contact-dependent Btl-Bnl binding might transmits a mechanochemical cue for local cytoskeletal re-organization required to produce cytonemes (Fig. 10a). Notably, contact-dependent nano-clustering of GPI-APs on the outer cell surface is known to induce local reorganization of cortical actomyosin in the inner membrane leaflet via trans-bilayer lipid interactions[24,27]. Although we detect co-clustering of Bnl and Btl at the contact sites, the possibility of GPI-anchored Bnl transmitting cell-shaping information via trans-bilayer mechanochemical feedbacks[34] needs future investigation.

Bnl is capable of long-range inter-organ dispersal[9,36]. According to the traditional paradigm, free secretion and

dispersion of paracrine signals promotes long-range morphogen-like spreading and signaling and signal retention in the source inhibits these functions. In contrast, we found that GPI anchoring of Bnl promotes long-range target-specific dispersal and morphogenetic potency (Fig. 9a–l). The contact-dependent Bnl release can also promote receptor-mediated endocytosis and activation of MAPK signaling in the recipient cells (Figs. 3k and 6b). These results are consistent with previous reports that the source surface Bnl retention facilitates its recipient-specific long-range dispersion and signaling patterns[9,36]. Although GPI anchor is critical for Bnl release, we do not know how GPI-anchored Bnl is released from the source membrane. We speculate that an enzymatic shedding[51] of Bnl might be activated at the cytoneme contact sites.

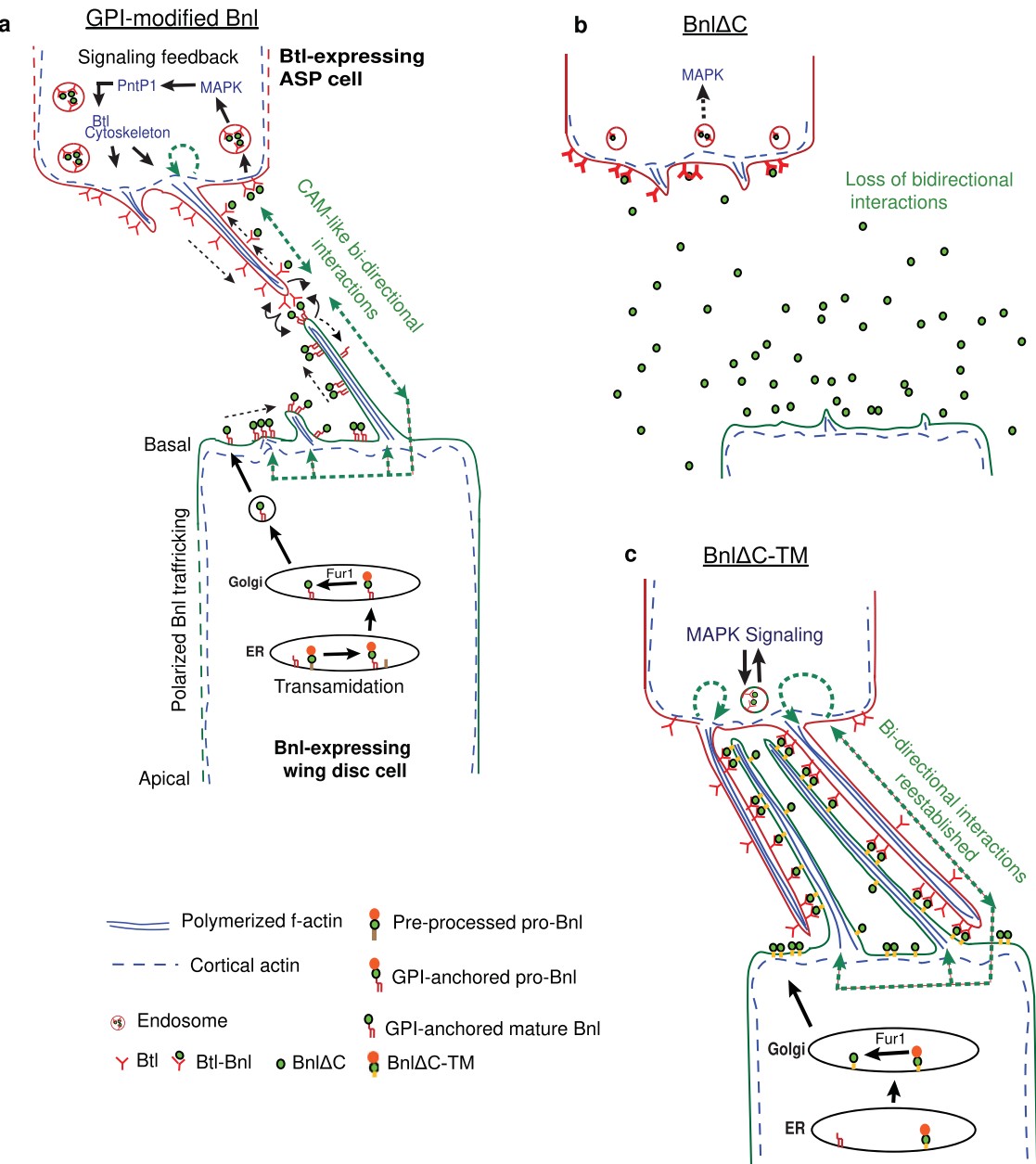

**Fig. 10 Dual roles of GPI-anchored Bnl. a** A model showing that the GPI-anchored Bnl directs both target-specific cytoneme contact formation and contact-dependent Bnl release. Green dashed arrows, a putative CAM-like local signaling mechanism that can promote polarized cytoskeletal reorganization for cytoneme formation. Black arrow (Bnl source), Bnl processing and trafficking[36,67]; black arrow (ASP cells), a MAPK signaling feedback that promotes the formation of Bnl-receiving ASP cytonemes toward the source[9]. **b, c** Schematic models illustrating why freely secreted BnlΔC might lack its CAM-like bidirectional signaling (**b**) and regain the bidirectional activity by the addition of a TM domain (BnlΔC-TM) (**c**).

A consequence of CAM-like Btl-Bnl interactions is that the cause and effect of the signaling process become interdependent. For instance, the same cytoneme contacts that the Btl-Bnl binding helps to form also bring Btl and Bnl molecules together to interact (Fig. 10a). Consequently, not only is the signal exchange cytoneme/contact-dependent, but the cytoneme contacts are also formed signal- or tissue-specifically. Notably, GPI anchoring of Bnl can link Bnl's CAM-like and morphogen-like functions. For instance, a readily secreted non-GPI Bnl that does not act as a cell surface CAM also fails to induce morphogen-like coordinated patterning (Fig. 9a–l). Whereas the same non-GPI Bnl added with a TM tether regains the CAM-like activity. However, the TM-tethered-Bnl induced a scaled-down patterning within a narrow range due to its poor release from the source. Thus, GPI-anchored Bnl provides a balance between two extreme functions - free/random secretion and inhibition of secretion via TM-tethered display.

The dual strategy of inhibition and activation of signal release can encode information for different context-specific outcomes. For instance, CAM-like Btl-Bnl signaling can direct cytoneme pathfinding and tracheal chemotaxis (Figs. 6d–l, 7f–m′, and 9e–h). Simultaneously, contact-dependent Bnl release can produce recipient-specific morphogen-like signaling patterns (Fig. 8a–e)[9,36]. Bnl release also can dissolve inter-cytoneme contacts (Fig. 10a). Dissociation of signaling contacts might be required for context-specific growth and plasticity of tracheal branches and the reciprocal guidance of Bnl-source and recipient cells in the embryo, shown earlier[52].

The membrane association and the dual strategy of inhibition and activation signal release might also be present in other signals. For instance, a transmembrane *Drosophila* FGF, Pyramus (Pyr)[38], is released target-specifically via cytonemes to only those adult muscle progenitors that adhere to the Pyr-expressing wing disc niche[38,53]. Similar to Bnl, Ephrins are GPI-/TM-tethered signals, and their interactions with receptors cause contact-dependent bidirectional signaling[22]. Lipid modifications are critical for the activity of Hh, Wnt, and EGF/Spi[22,44,54,55]. Analogous to Bnl, TM-tethering of Hh, Spi, and Wnt can efficiently induce tissue organization within a narrow range, and removal of lipid-modification and unrestricted spreading of non-lipidated Hh, Spi, and Wnt reduce their morphogenetic potency[44,56–60]. Moreover, all signals, including those that are not known to be lipidated (e.g., BMPs and many FGFs), can interact with membrane-anchored proteoglycans, which can restrict free signal dispersion and induce biphasic signaling activation and inhibition[61–63]. Glypicans also can control cytoneme stability[64–66]. Therefore, our findings showing how GPI anchored Bnl directs source and recipient cells to reciprocally coordinate with each other by cytonemes provide important insights into how other signal retention strategies might control cytoneme-mediated signaling and morphogenesis.

## Methods

**Fly genetics**. All fly lines and their sources are described in Supplementary Table 4. Flies were raised at 25 °C with a 12 h/12 h light/dark cycle, except for tracheal *dia-RNAi* expression. All the experiments were performed in non-crowded situations. The sequence-verified DNA constructs were used to generate transgenic flies by P-element-mediated germline transformation as described in Du et al.[9]. Transgenic injections were performed by Rainbow Transgenic Flies, Inc.

**Mosaic analyses**. (i) To generate ectopic clones in the ASP, *hsFlp; btlenh > y + >Gal4,btlenh-mRFPmoe* females were crossed to males carrying *UAS-Bnl:GFP, UAS-Bnl:GFPΔC, UAS-Bnl:GFPΔC-TM,* or *UAS-Btl-DN.* Flip-out clones were generated by heat shocking early third instar larvae at 37 °C for either 5 or 10 min. Larvae, then were incubated at 25 °C until they reached the mid-late third instar stages and dissected for further analysis.

(ii) To generate CD8:GFP-expressing clones in the *bnl* source, *hs-mFlp;bnlGal4* females were crossed to FlyBow *FB2.0* flies (see Supplementary Table 4) and clones were induced in the progenies by heat-shock. Only CD8:GFP-marked cells were visualized in live tissues.

(iii) Ectopic Bnl:GFP-TM-expressing clones in the wing disc were induced in progenies of *hs-Flp;UAS-bnl:GFP-TM* (females) × *mCherryCAAX;act > CD2 > Gal4* (males) cross.

(iv) Ectopic Btl:GFP-expressing clones in the wing disc were induced in progenies of *hs-Flp;;UAS-Btl:GFP* (females) × *act > CD2 > Gal4;;bnlLexA,LexO-mCherryCAAX* (males) cross.

**Tissue-specific transgene expression**. For the transgene expression in the trachea, *btl-Gal4/UAS* or *btl-LexA/LexO* systems were used. To express transgenes in the wing disc *bnl*-source, *bnl-Gal4/UAS* or *bnl-LexA/LexO* systems were used. Comparable levels of *bnl-Gal4*-driven expression of GPI-modified Bnl:GFP and non-GPI modified Bnl:GFPΔC and Bnl:GFPΔC-TM were determined as described in the Supplementary Information (Supplementary Notes, section C). Although *bnl* is not expressed in the salivary gland, *bnl-Gal4* is non-specifically expressed in the larval salivary gland[37]. Therefore, *bnl-Gal4* was used to ectopically express Bnl:GFP variants in the larval salivary glands. Thus, phenotypic consequences of *bnl-Gal4*-driven expression of Bnl:GFP variants were recorded in two distinct tissue contexts of the same larva: wing disc (for native Bnl source and ASP interactions) and salivary glands (for ectopic source and tracheal invasion into the ectopic source).

**Cytoneme removal from the ASP and *bnl*-source**. To remove source cytonemes, *UAS-dia-RNAi* was expressed under *bnlGal4* and larvae were reared at 25 °C. In the trachea, a high-level *dia-RNAi* expression (at 25 °C) caused larval lethality. Therefore, *tub-Gal80ts; UAS-diaRNAi* males were crossed to *btlGal4,UAS-CD8:GFP; bnlLexA,LexO-mCherryCAAX/TM6* females; the *btl-Gal4*-driven expression of *dia-RNAi* was suppressed by *Gal80ts* at 18 °C until L3 stage and activated by shifting the temperature to 29 °C (that inactivated *Gal80ts*), 24 h prior to harvesting the L3 larvae for imaging.

**Cell lines and cell culture**. Schneider's 2 (S2) cells (S2-DGRC) were cultured and transfected following standard protocols[36]. Cells were transfected either with Lipofectamine 3000 or Mirus TransIT-Insect Transfection Reagent for CAM assays

following the manufacturer's protocol. Transient ectopic expression of various constructs in S2 cells was achieved by co-transfecting *act-Gal4* and *UAS-x* constructs (x = various cDNA or cDNA fusions) and analyzed after 48 h of incubation at 25 °C.

**Immunohistochemistry**. The standard immunostaining and the extracellular immunostaining under live-cell non-permeabilized condition (αGFP^ex for GFP or αBnl^ex for Bnl) were carried out following the standard protocols[9,36]. Supplementary Table 4 lists all antibodies and dilutions used.

**DNA constructs**. All constructs generated and used here are described in Supplementary Table 4.

**Bioinformatic analysis**. DNA sequences were analyzed with SnapGene, Protein sequences were analyzed with MacVector, ProtScale (ExPASy), EMBOSS Pepinfo (www.ebi.ac.uk), and PredGPI (http://gpcr.biocomp.unibo.it/predgpi).

**Flow cytometric analyses**. S2 cells expressing various constructs were immunostained and scanned using a BD CantoII (BD Biosciences) flow cytometer and the data were analyzed using FACSDiva (BD Biosciences). For quantitative assays as shown in Supplementary Figs. 3b–h and 4e–g, the number of cells detected in Q2 (GFP+ cells with αGFP^ex+) was divided by the number of cells in either Q2 or Q4 (total GFP+ cells) to obtain the Y-axis value. These values were obtained from three independent experimental repeats. An example of the gating strategy for FACS analyses is shown in the Supplementary Information (Supplementary Note, section D).

**Ex vivo organ culture and Furin inhibition**. Ex vivo wing disc culture in WM1 media, pharmacological inhibition of Furin in cultured discs, and analyses of ASP-specific uptake of Bnl were carried out following standard protocols described in Sohr et al.[36,67]. In brief, late third instar larval tissues were ex vivo cultured in 2 ml of WM1 medium in the presence or absence of a cocktail of Furin inhibitor I and II (50 μM final concentration each; Calbiochem; 344930 and 344931). Cultured discs were removed from a single pool of culture media after 0, 1, 2.5, and 5 h of incubation at 25 °C, followed by fixation and αHA immunostaining of the tissues. The temporal increase in the levels of GFP-tagged Bnl in the ASP over time was difficult to assess due to the pre-existing Bnl:GFP_3 in the L3 ASP used for culturing. Therefore, Furin-sensors (HA_1Bnl:GFP_3, HA_1Bnl:GFP_3ΔC-TM, and HA_1Bnl:GFP_3ΔC) that were detectable by both αHA immunostaining and GFP were used. The time when tissues were transferred to the Furin-inhibited media was considered as t = 0 for the appearance of intact Furin sensors (HA_1Bnl:GFP_3). For comparative analyses among samples, a semi-quantitative estimate was obtained by measuring the ratio of the uncleaved sensor (αHA immunofluorescence intensity) to the total GFP signal (pre-existing Bnl:GFP_3 + post-inhibition HA_1Bnl:GFP_3) per ASP for t = 1 or 2.5 or 5 h.

**CAM assay using S2 cells**. S2 cells ectopically expressing either Btl variants (*UAS-Btl:Cherry, -BtlDN:Cherry,* or *-sBtl:Cherry*) or ligand variants (*UAS-Bnl:GFP* or *-Bnl:GFPΔC* or *-Bnl:GFPΔC-TM*) (48 h after transfection) were resuspended in 1 ml of fresh M3 media. In all, 200 μl of the receptor-expressing cells was gently mixed with 200 μl of the ligand-expressing cells for 10 min in a sterile tube. The well-mixed cell suspension was plated to the center of a sterile cover slip within a 6-well plate and incubated at 25 °C for 16 h before fixing them with 4% PFA following standard protocols. Coverslips were carefully mounted with cells facing down to 10 μl of the VECTASHIELD on microscopic slides. For comparative analyses, co-culture assays were performed in identical conditions. Cells were analyzed from more than three transfection repeats, with at least 30 random frames/experiment under ×20 and ×40 objectives. Regions with comparable cell density were analyzed. Adjacent cells with the ring-like heterophilic receptor–ligand co-clusters were considered as trans-paired cells and those without the receptor-ligand co-clusters were considered as juxtaposed. Homophilic Btl–Btl or Bnl–Bnl clusters between adjacent cells were rarely observed as indicated in Supplementary Fig. 2h. Cells were imaged in both ×20 and ×40 to thoroughly verify Btl-Bnl trans-pairing in the mixed cell population.

**Autocrine and paracrine Bnl-Btl signaling in S2 cells**. For autonomous MAPK signaling, S2 cells were co-transfected with *act-Gal4, UAS-Btl:Cherry,* and *UAS-X* (X = various Bnl:GFP variants) and prepared on cover-slips as described before. Cover slips with cells were processed with standard fixation and anti-dpERK immunostaining. The percentage of Btl:Cherry expressing cells with nuclear dpERK signals was scored on confocal microscope (×20/×40). For non-autonomous dpERK signaling, cells were prepared following the CAM assay, followed by PFA fixation and anti-dpERK staining. Both trans-paired and unpaired Btl:Cherry variants were recorded to estimate the contact-dependent non-autonomous dpERK signaling.

**PI-PLC treatment of transfected S2 cells and wing imaginal discs**. Transfected S2 cells (1 ml) were harvested (700 g, 5 min) in a 1.5 ml Eppendorf tube. Cells were washed twice in 1×PBS (500 μl each) and incubated either in 500 μl 1×PBS

(control) or in PI-PLC containing 1×PBS solution (1 U/ml PI-PLC) at 20–25 °C for 30 min with gentle rotation. Cells were harvested and prepared for the standard non-permeabilized extracellular staining before imaging or FACS. To reliably compare the levels of surface-localized proteins with and without PI-PLC treatment, the ratio of the surface:total Bnl levels per cell was measured using Fiji (at least three independent repeats). Note that the surface levels of GFP-tagged proteins per cell were measured with αGFP[ex] immuno-fluorescence and the total GFP fluorescence of the same protein measured the total expression in the same cell. For untagged Bnl, CD8:GFP was co-transfected and the Bnl[ex] level was normalized with CD8:GFP in the same cell.

For PI-PLC assay in wing discs, third instar larvae were prepared following ex vivo organ culture method[36] and transferred to 1.5 ml Eppendorf tubes containing 1 ml of either WM1 media (control) or WM1 media with PI-PLC (1 U/ml). Tissues were incubated for 30 min at 20–25 °C with gentle rotation. Then the PI-PLC reaction was stopped by removing the solution and washing the tissues three times with WM1 media. Tissues were then prepared for extracellular staining as described before.

**Live imaging of cytonemes**. Wing imaginal discs were prepared and imaged in WM1 medium as described in Du et al.[9]. Time-lapse imaging of cytonemes was carried out in ex vivo cultured wing discs in Grace's insect culture medium as described in Barbosa and Kornberg[68]. A spinning disc confocal microscope under ×40/×60 magnifications was used to capture ~30–50 μm Z-sections with 0.2 μm step size of wing discs. For Fig. 1e, e′, images were captured using the Zeiss LSM900 confocal with an Airyscan-2 detector in ×60 magnifications. The images were processed and analyzed with Fiji. For 3D-rendering, Andor iQ3 and Imaris software were used.

**Quantitative analyses of cytoneme number, orientation, and dynamics**. Cyonemes were manually counted and plotted by methods described in Du et al.[9]. For ASP cytonemes, cytonemes were recorded across a 100 μm arc centered at the tip (Figs. 2i–k, l, m, m′, n and 6d–l and Supplementary Fig. 6g). Wing disc *bnl* source cytonemes were recorded from the 3D projections across a 100 μm perimeter surface centering at the ASP tip contact as a reference (Figs. 2a–e, l′, m″, m‴, n′ and 6e-l and Supplementary Fig. 6g). For Fig. 2l′, n′, cytonemes were not grouped based on the length as all source cytonemes were <15 μm. For Figs. 1l–n and 7d, e, different parameters of cytoneme dynamics were measured following previous reports (see Supplementary Table 1)[65,69].

**Quantitative analyses of fluorescence intensities in tissues**. For intracellular and extracellular surface Bnl levels, all fluorescent intensity measurements were background corrected. The density of fluorescence intensity (e.g., spatial range and density of signals) was measured from maximum-intensity projections encompassing the wing disc, ASP, or salivary gland sections from a selected region of interest (ROI) using Fiji. For each genotype, at least three samples were used to obtain the average plot profile. Quantitative estimates of levels of Bnl:GFP variants and signaling outcomes are normalized with internal controls to avoid variations among samples. For example, to compare between Bnl variants, we compared the ratio of surface levels of each protein (red, anti-GFP non-permeabilized immunofluorescence) to total expression (total GFP fluorescence) in the same ROI of wing disc source (Fig. 5) and the salivary glands (Fig. 9i–l). Similarly, to assess MAPK signaling patterns of different Bnl variants (Fig. 9a–d), we measured the percentage of signal recipient cells (cells with Bnl:GFP variant puncta) that induced MAPK. The correlated patterns between signal reception and signaling per cell/tissue were then compared between conditions.

**Sholl analysis of tracheal branching in salivary gland**. The extent and frequency of tracheal branching on the larval salivary glands expressing equivalent levels of Bnl:GFP, Bnl:GFPΔC-TM, or Bnl:GFPΔGPI was quantitated using Sholl analysis in Fiji as described in ref. [36]. The analysis created 20 concentric circles in increments of 5-μm radius from the point of origin up to 100 μm and counted the number of times any tracheal branch crossed these circles. These values were averaged across multiple samples and compared between the different Bnl variants expressed in the salivary gland.

**Statistics and reproducibility**. Statistical analyses were performed using VassarStat and GraphPad Prism 8, MS Excel. *p* Values were determined using the unpaired two-tailed *t* test for pair-wise comparisons or the one-way analysis of variance followed by Tukey's honestly significant different test for comparison of multiple groups. $p < 0.05$ is considered significant. All experimental results were analyzed from at least three independent experiments. The sample size (*n*) for each data analysis is indicated in the figures/figure legends and Source data. All cells for each condition showed consistent patterns. Graphs in Fig. 4f, g show intensity analyses from randomly selected cells from a large pool of cells from three experimental repeats. The results were confirmed using FACS analyses of the same cell populations (Supplementary Fig. 3b–h). Rose plots were generated by the R software as described in ref. [9].

**RNA isolation and RT-PCR**. Total RNA was extracted from 20 wing discs of the $w^{1118}$ L3 larvae using TRI reagent (Sigma-Aldrich) followed by Direct-zol RNA

purification kits (Zymo Research). Expression analyses of *bnl PA* and *PC* isoforms are described in Supplementary Information.

**Reporting summary**. Further information on research design is available in the Nature Research Reporting Summary linked to this article.

## Data availability
All data generated and analyzed are included in the manuscript and supporting files. Source data are provided with this paper.

## Code availability
The code for R plots is provided in the Supplementary Information.

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

## Acknowledgements

We thank Dr. T.B. Kornberg and Dr. G.O. Barbosa for sharing the design of the culture chamber for live imaging; Ge Yan for R plot analyses; Dr. N. Andrews, Dr. T.B. Kornberg, Dr. W. Snell, and Dr. S. Ogden lab for comments on the manuscript; the Bloomington Stock Center for *Drosophila* lines; the DSHB for antibodies; and A.E. Beaven for UMD imaging core facility. Funding: NIH grant R35GM124878 to S.R.

## Author contributions

A.S. discovered GPI-anchoring of Bnl and L.D. discovered bidirectional signaling and roles of GPI-anchored Bnl; S.R. supervised the work and designed the project; L.D., A.S., and Y.L. conducted experiments: S.R., L.D., and A.S. wrote the paper.

## Competing interests

The authors declare no competing interests.
