## [Peer Review File · Nature Communications]

GPI-anchored FGF directs cytoneme-mediated bidirectional signaling to self-regulate tissue-specific dispersionREVIEWER COMMENTS

Reviewer #1 (Remarks to the Author):

In this elegant work Du et al. identify for the first time a lipid-modified FGF and characterize how lipidation anchors the signal to the plasma membrane to program cytoneme-mediated events controlling tissue-specific FGF dispersion and signaling. This is a major discovery that changes our view of FGF dispersion during paracrine signaling and it is against signaling models that depend on random propagation of signals from source to recipient cells. The authors also discovered that FGF locates at cytonemes emanating from the source cells and that these cytonemes are oriented towards the receiving ASP tissue. Thus, cytoneme-associated FGF receptors interact with the membrane-anchored ligand in a synapse-like cytoneme-to-cytoneme contact. Furthermore, these cytoneme-cytoneme contacts not only allow the spatial restriction of signaling but also regulate cytoneme establishment and guidance, generating a loop of auto-regulation between donor and receiving tissues. The manuscript is well presented and the experimental evidences are strong and complete, showing spectacular videos and pictures of cytoneme-cytoneme contacts. Therefore, the scientific contribution to the field is undoubtedly valuable, as it provides major advances to our understanding of organized long distance cell-to-cell communication.

Thus, I strongly recommend the publication of the manuscript and just have a few comments/suggestions for the authors to consider.

In general we think the authors should increase the weight of their discovery regarding the GPI anchoring of FGF, all throughout the manuscript. This is an important discovery, that provides solid arguments in favor of cytoneme-mediated signaling, and it should be emphasized. In this context, Western blots of the processed FGF should be presented to show the sizes of the FGF processed proteins. The size of FGF before and after furin cleavage and GPI modification could be analyzed in tissue culture cells.

1. As the author discusses, many FGFs are retained on the cell surface by their interaction with GPI-anchored glypicans, a process critical for signal interpretation (Balasubramanian & Zhang 2016). In addition, glypicans are known to be needed for cytoneme formation in the ASP (Huang and Kornberg, 2016) and for cytoneme guidance and stabilization in the wing disc (González Mendez et al., 2017). More specifically, FGF-receiving ASP cytonemes require Dlp (but not Dally) for navigation (Huang and Kornberg, 2016). Therefore, it could be possible that the reduction of BnlEx after PI-PLC treatment (Fig. 3F-H) could also be mediated by the lack of function of Dlp. It would be interesting to analyze the length of the disc cytonemes and the expression of BnlEx in Dlp mutant conditions in Bnl-expressing cells.

2. Fig. 2 shows that, after DlpRNAi expression in the source cells, randomly oriented short ASP cytonemes are unaffected, while Bnl-sending cytonemes are required for the formation of polarized Bnl-receiving cytonemes from the ASP (Fig. L-K). It also shows that the expression of Btl:DN in the ASP leads to a corresponding non-autonomous loss of polarized source cytonemes (Fig. 2L-O). We suggest the addition of quantifications to better define the cytoneme polarization, what might strengthen the evidence for a cytoneme guidance effect.

3. Additional evidence for cytoneme establishment self-regulation could be presented by looking at the previously published factors (Pointed-P1 and Cut) involved in receiving-cytoneme promotion/inhibition (Du et al., 2018). In this occasion from the perspective of the FGF producing cells.

4. As a minor comment, in line 380-381 the sentence should be changed to "might be sufficient" as no detailed mechanism is described.

5. Fig 4J the intensity of the red channel should be increased or split into the two channels; it is not easy to distinguish the dpERK activation.

6. The interaction between ligand and receptor is shown associated to cytoneme contacts. It would

be interesting to discuss how and where the endocytosis process happens in the receiving cells.

Reviewer #2 (Remarks to the Author):

In the manuscript, Du et al. reported that cytonemes are not only present in the signal receiving cells in the ASP (Bth FGFR expressing) but also are associated with the sending cells that express FGF ligand Bnl. They showed that the distribution of Bnl and effective FGF signaling depends on a C-terminal hydrophobic region that allows the ligands to be anchored to the cell membrane through a GPI moiety. Furthermore, GPI anchorage of Bnl facilitates tissue-specific release at inter-cytoneme contacts and diffusion of its FGF domain-containing fragment into the ASP upon interacting with FGFR on the cytonemes, leading to desirable signaling outcomes during the downstream events. The authors also suggest that Bnl acts as a CAM (cell adhesion molecule?) and signals bidirectionally. No references are included to give this CAM-signaling context; and the angle of bidirectional signal does not seem well supported. On the other hand, the mechanism demonstrated in the manuscript, by which FGF signaling is spatiotemporally regulated at the level of ligand dispersion through its lipidation, is compelling, and potentially provides important insights into how signaling pathways are regulated through membrane-association of ligand. The experiments are well designed and carefully carried out; nevertheless, most of the in vivo experiments rely on ectopic expression instead of mutant allele characterization which is a limitation. It is therefore hard to know if native Bnl function requires the GPI. Lastly, though the main text is logically written and easy to follow, the clarity of figure could be improved; parts are a bit disorganized and several charts lack consistent labelling.

Major concerns:

(1) Bnl is known to have two splice variants expressed in embryos affecting the very C-terminal region of Bnl. One form (being described in the manuscript) is predicted to have a hydrophobic region, while the other is not (see Stepanik et al. *Curr Biol* 2020). Are both forms also expressed in the wing disc? The presence of the shorter variant that does not possess the hydrophobic region might represent yet another layer of how FGF signaling is regulated during development. It's intriguing to think that the two different Bnl isoforms could carry out the combined signaling properties of the other 2 *Drosophila* FGFs to their dedicated FGFR Htl [i.e. TM domain-containing Pyr and non-TM Ths (Stepanik et al., 2020)]. The authors should clarify and expand on that point, perhaps in terms of the coordinated use of GPI-anchored Bnl, post-GPI cleavage Bnl, and "freely-secreted" Bnl in the dynamics of ASP morphogenesis. At minimum, the authors should cite this other study which pointed out the C-terminal hydrophobic domain in Bnl and previously suggested it might be membrane-associated.

(2) The expression level of different mutant forms of Bnl need to be quantified. Modification on the C-terminus of Bnl may affect the stability and other functional cleavage sites of the protein. Therefore, it is particularly important to make sure that all those mutant forms are expressed at comparable levels. Specifically, for Bnl:GFP- ω m and Bnl:GFP Δ C, a western blot should be done to make sure that they are expressed at reasonable levels and to what degree each is secreted into the extracellular space (supernatant).

(3) Provide further evidence that the GPI anchoring is functional in vivo for Bnl. Data presented does strongly suggest that Bnl could be GPI-anchored through the computational identified glypiation site S741. However, the presence of a TM domain is not completely ruled out, and deletion of the 40 aa C-terminus could very well disrupt a potential cleavage site that is important for release of the ligand if TM is functional, even outside of GPI utilization. And indirectly, a protease that is responsible for cleaving off the TM domain could be GPI-anchored, thus PI-PLC treatment may activate the protease and release Bnl from the membrane. That being said, GPI anchorage is still a favorable mechanism that regulates the distribution of Bnl. To make the case more convincing, Bnl:GFP Δ C can be linked to a known glypiation sequence to rescue the mutant

phenotype. Additionally, Crispr bnl mutants bearing mutated glypiation sites (ωm) can (and should) be made to confirm that those sites function endogenously (see point 4 below).

(4) The case for GPI regulating endogenous Bnl activity must be strengthened through further characterization of Bnl- ΔC (C-term hydrophobic domain deletion) and Bnl- $\omega.m$ (GPI predicted residue mutant). Crispr/Cas9 can be used to create mutants (possibly clones) to determine if the mutations affect Bnl endogenous function; there is some concern that ectopic expression experiments for the most part form the basis for conclusions. In addition, Bnl- $\omega.m$ should be assayed in ectopic expression experiments side-by-side with the C-term deletion in order to ascertain whether the GPI linkage is affecting function (rather than some other functionality associated with the C-terminal domain).

(5) Clarify the mechanism by which GPI-anchored Bnl is released upon receptor binding. The results from PI-PLC assay do suggest that Bnl is associated with cell membrane through a GPI anchor, however, it is unclear how it is released in the context of FGF signaling. It will be helpful if the authors can provide more clues by looking at Bnlex distribution in btl mutants or expand on the topic in discussion.

(6) Is the signaling truly bidirectional between Btl and Bnl? Besides the finding that ASP-originated cytonemes can influence cytonemes from the wing disc, what is the additional evidence that supports reverse signaling events in ligand-presenting/ wing disc cells? Is there a proposed nature of the information relayed back to the Bnl-producing cells? The authors also suggest that Bnl acts as a CAM (cell adhesion molecule?) and signals bidirectionally. No references are included to give this CAM-signaling context; and the angle of bidirectional signal does not seem well supported.

(7) Presentation of data/style: (i) Fig3, take out the two black panels in C'; assuming this experiment was not done (why else would the green signal be missing?). (ii) Label L'-Q' on top with α GFPex instead of genotypes. (iii) In general, labelling, styles and formats of the charts in the manuscript are not consistent, different colors are used to represent the same genotype. Would be more friendly to the readers if the authors could improve on the presentation of their statistics. In the zip file for reviewers 12 movies are included; this made analysis of the movies a bit confusing as the files did not contain labels.

Minor suggestions to improve clarity:

Line 70: "signal-exchanging cells ^can^ extend actin-based signaling filopodia"
(thought it is clear that Bnl-Btl signaling uses cytonemes in this context, the same may not be true for all ligand-receptor pairs)

Lines 92-95: sentence is awkward in terms of subject/verb agreement; making "contacts" singular in lines 93 might fix this. It is difficult to interpret if they are saying in lines 94,95 if they are saying that all spatiotemporal cues in morphogenesis are controlled by molecular and cellular processes that control cytonemes, or just mechanisms utilizing cytonemes

Line 123: What is meant by "unbiased interpretation" of tissue-specific dispersion? It may be helpful to more explicitly connect this with the idea following in lines 125-128 that varying fixation/detection conditions can show different pools of Bnl protein. It may be more straightforward to simplify the sentence in lines 122-123 as "...is a simple genetic system for studying dispersion of Bnl produced in the wing disc..."

Lines 260-263:

It should be mentioned that this isolated region of Bnl has already been identified to anchor an FP to the plasma membrane when exogenously expressed in S2 cells (see (Stepanik et al. 2020) above).

Line 277, line 435, elsewhere:

Saying that Bnl's GPI anchor "programs" its dispersion and signaling connotes a scenario where it dictates the entire process, while it might be more precise to say its regulated use is part of a

program/process. Might be better to use words such as "dictates," "enables," or "instructs."

Lines 416-419:

-Using "On one hand..." followed with "On the other hand..." is language that suggests there is a conflict between their observations on cytoneme dynamics and identifying Bnl as the first lipid-modified FGF and effect on cytoneme dynamics. Deleting "On one hand" and replacing "on the other hand..." with "furthermore" would better show the importance of their complementary data.

Line 476:

Delete "is designed to" from "cytoneme-mediated signaling is designed to orchestrate" and use plural "orchestrates" to simplify the sentence, more effectively highlight their finding, and avoid assigning an outside intent to how these processes coordinate.

Reviewer #3 (Remarks to the Author):

Cytonemes are used for intercellular communications through various molecules such as Hh, Dpp, Wg, EGF and FGF. Du et al. revealed that FGF ligand-receptor (Bnl-Btl) signaling between air-sac primordium (ASP) and wing discs involves contact of cytonemes from both tissues. They also found that bidirectional signaling, FGF signal from wing discs and unknown signal from ASP, is essential for formation of both cytonemes. Moreover, they report that *Drosophila* FGF (Bnl) is a GPI-anchored protein. These findings are very interesting and valuable for publication. However, the reviewer suggests some points to improve the manuscript.

Major points

1) Line 179-183: If the expression of Bnl is affected by *dia* knockdown, it is difficult to conclude that source cytonemes are necessary to deliver Bnl. The authors should examine whether the expression of Bnl changes upon *dia* knockdown.

2) Line 200-208: Analysis using dominant negative forms sometimes includes indirect effects. Thus, the authors should use *dai* RNAi instead of Btl:DN. Since Gal4/UAS system is temperature sensitive, the authors may reduce *dai* expression at different levels by doing knockdown experiments at different temperatures.

3) Figure 4D-F: Activation of ERK by endogenous Bnl was reported in the previous report by Du et al (eLife, 2018). Staining with anti-dpERK in Figure 4D-F includes the endogenous activation in addition to ectopic activation by Bnl:GFP, Bnl:GFPdeltaC-TM and Bnl:GFPdeltaC expression. To avoid confusion, the authors should perform these experiments in *bnl* mutant with expressed Bnl:GFP, Bnl:GFPdeltaC-TM and Bnl:GFPdeltaC.

4) Figure 5A-I: There was not a control experiment in which ASP and source cytonemes are analyzed in the wild type. The authors should analyze the length and direction of both cytonemes in the wild type.

5) It is reported that MAPK is efficiently activated in endosomes where a receptor-ligand complex localizes. However, Bnl:GFPdeltaC-TM that cannot be shed in the plasma membrane could activate MAPK in ASP. How do you explain this?

6) Shedding of Bnl in the contact site of cytonemes is essential for the signaling. What is a shedding enzyme responsible for Bnl? This can be discussed.

Minor points

Line137: It would be good to the readers to explain what is mCherryCAAX.

REVIEWER COMMENTS

Reviewer #1 (Remarks to the Author):

In this elegant work Du et al. identify for the first time a lipid-modified FGF and characterize how lipidation anchors the signal to the plasma membrane to program cytoneme-mediated events controlling tissue-specific FGF dispersion and signaling. This is a major discovery that changes our view of FGF dispersion during paracrine signaling and it is against signaling models that depend on random propagation of signals from source to recipient cells. The authors also discovered that FGF localizes at cytonemes emanating from the source cells and that these cytonemes are oriented towards the receiving ASP tissue. Thus, cytoneme-associated FGF receptors interact with the membrane-anchored ligand in a synapse-like cytoneme to cytoneme contact. Furthermore, these cytoneme-cytoneme contacts not only allow the spatial restriction of signaling but also regulate cytoneme establishment and guidance, generating a loop of auto-regulation between donor and receiving tissues. The manuscript is well presented and the experimental evidences are strong and complete, showing spectacular videos and pictures of cytoneme-cytoneme contacts. Therefore, the scientific contribution to the field is undoubtedly valuable, as it provides major advances to our understanding of organized long distance cell-to-cell communication.

Thus, I strongly recommend the publication of the manuscript and just have a few comments/suggestions for the authors to consider.

Ans: We thank the reviewer for the encouragement and strong recommendation for publication. We really appreciate all the valuable suggestions and we tried to address all suggestions, which significantly improved the paper.

MAJOR COMMENTS:

1) In general, we think the authors should increase the weight of their discovery regarding the GPI anchoring of FGF, all throughout the manuscript. This is an important discovery, that provides solid arguments in favor of cytoneme-mediated signaling, and it should be emphasized. In this context, Western blots of the processed FGF should be presented to show the sizes of the FGF processed proteins. The size of FGF before and after furin cleavage and GPI modification could be analyzed in tissue culture cells.

Ans: Thank you for this suggestion. We modified the text to increase the weight of the discovery of the GPI-anchored FGF and its CAM-like bidirectional FGF-FGFR signaling that controls cytoneme contacts/polarity and contact-dependent signal release. We also added additional evidence supporting this model.

Western blot: We appreciate the suggestion of Western blot analyses for detecting the sizes of processed FGF products as this experiment would also allow us to predict any additional Bnl modifications. I am sorry that we did not mention this before, but we had previously published these biochemical analyses in Sohr et al 2019, which provided a clue to multiple Bnl cleavages, in addition to the Furin cleavage reported in Sohr et al. 2019. We now mentioned this in the revised text. Here, I attached the published Western blot picture (anti-GFP) from Sohr et al (2019), analyzing the same Bnl chimeras: Bnl:GFP₃, the Furin sensor HA₁Bnl:GFP₃, and Bnl:GFP₁ (not used in this paper). We used Bnl:GFP, because Bnl antibodies (3 different ones) did not work for the Western blot. These chimeric proteins harboring GFP/Cherry/HA-tags at various locations were originally made to identify an active tagged-protein, led to the accidental discovery of various Bnl processing, including glypiation. Bnl:GFP Western blots (anti-GFP) from cell lysates detected multiple different bands, which is consistent with multiple Bnl cleavages, but it is hard to estimate the MW of the cleaved products. For example, the table below shows the expected and observed MW of Bnl:GFP in cells as it is processed and trafficked from the ER to Golgi to cell surface.

We detected multiple bands between 115-95 kDa, which matched with the expected MW

□GFP Western blot	Bnl protein (kDa)	Expected size of Bnl:GFP ₃ protein (kDa)	Observed size (kDa)
Full length (ER)	84	113	>150
After the signal peptide (SP) cleavage (ER)	80	109	~115?
After the signal sequence (SS) cleavage for GPI (ER) - trafficked to Golgi	66	95	~100?
After the Furin cleavage (Golgi) trafficked to cell surface	63	92	~95?

of proteins after the post-SP cleavage (109 kDa) and post-SS cleavage (95 kDa) (both occur in the ER), and after the pro-domain cleavage by Furin in Golgi (92 kDa). However, it is unclear, whether to expect the correct sized bands for the cleaved products, especially when the full-length protein is detected as >150 kDa, instead of 113 kDa. The mature protein is also expected to have the GPI anchor and additional unknown modifications (e.g., proteolytic shedding or glycosylation), if any. Another example is that the detection of unique smaller bands (~37 and 60 kDa) for N-terminally tagged Bnl:GFP₁ and unique larger bands (>100 kDa) for C-terminally tagged Bnl:GFP₃ and Bnl:HA₁GFP₃ was consistent with a cleavage near tagging-site 1, but where is the actual cleavage site was hard to assess by western blot analyses. Moreover, the intracellular and intercellular trafficking of different Bnl parts could not be visualized using biochemical assays.

2. As the author discuss, many FGFs are retained on the cell surface by their interaction with GPI-anchored glypicans, a process critical for signal interpretation (Balasubramanian & Zhang 2016). In addition, glypicans are known to be needed for cytoneme formation in the ASP (Huang and Kornberg, 2016) and for cytoneme guidance and stabilization in the wing disc (González Mendez et al., 2017). More specifically, FGF-receiving ASP cytonemes require Dlp (but not Dally) for navigation (Huang and Kornberg, 2016). Therefore, it could be possible that the reduction of BnlEx after PI-PLC treatment (Fig. 3F-H) could also be mediated by the lack of function of Dlp. It would be interesting to

analyze the length of the disc cytonemes and the expression of Bnlx in Dlp mutant conditions in Bnl expressing cells.

Ans: Thank you for these important points. We discussed HSPG's role in cytoneme regulation in the discussion of the revised Mss. We are investigating how HSPGs might affect cytoneme-mediated FGF signaling, and we would like to perform the experiments suggested.

A possibility of PIPLC treatment removing Dlp-bound surface Bnl: Although Bnl has the conserved HSPG binding domain (Fig. 4 A, FGF domain), PIPLC-dependent removal of surface-localized Bnl is a direct effect. We conclude this based on the following results in the revised text: a. PIPLC-dependent Bnl shedding was observed in S2 cells that have little glypican expression (based on modENCODE). b. We used two constructs: (i) bGFP-GPI had Bnl's signal peptide (SP) and signal sequence (SS) but lacked Bnl's HSPG binding domain. (ii) Bnl:GFP \square _{FGF} (dummy Bnl) where the central FGF domain containing the HSPG binding site was replaced with an sfGFP sequence. These chimeric proteins were GPI anchored to the surface and were removed by the PIPLC treatment. c. Bnl:GFP \square _C had the glypican binding site, but it was not retained on the source cell surface, even in the absence of PIPLC (Figs. 5I,M; S5A,A'). In contrast, the addition of a TM to the C-terminus of Bnl:GFP \square _C, rescued its source surface localization and target-specific activity, like a GPI-anchored Bnl. We modified the text to highlight these points.

4. Fig. 2 shows that, after DiaRNAi expression in the source cells, randomly oriented short ASP cytonemes are unaffected, while Bnl-sending cytonemes are required for the formation of polarized Bnl-receiving cytonemes from the ASP (Fig. L-K). It also shows that the expression of Btl:DN in the ASP leads to a corresponding non-autonomous loss of polarized source cytonemes (Fig. 2L-O). 2. We suggest the addition of quantifications to better define the cytoneme polarization, what might strengthen the evidence for a cytoneme guidance effect.

Ans: Thank you for this important point! In the revised Figure 2L-N', we added additional quantitative data on *dia-i* expression from the ASP using R-plots (also see Table S2 for the numerical quantifications).

For Btl:DN, we presented a correlation plot between source and ASP cytoneme numbers per wing disc with a goal to highlight the reciprocal relationship between them (Revised Fig. 3C, each dot represents the number of ASP and source cytonemes in a single disc). In Btl:DN condition, the majority of the samples lacked ASP and therefore also lacked source cytonemes. However, a partial Btl:DN effect allowed the appearance of variable-number of ASP/tracheal cytonemes, which in turn also led to the appearance of a similar number of polarized source-cytonemes forming cell-cell contacts. The numbers of cytonemes were variable from disc to disc, which was what we needed to test the reciprocal correlation of cytonemes using a single condition. Moreover, these results also provided a clue that Btl:DN, despite its inability to induce MAPK signaling, might be able to act as a CAM for binding to GPI-Bnl. We further provided evidence of CAM-like activity of Btl:DN in an additional experiment (Fig. 3F-M').

5. Additional evidence for cytoneme establishment self-regulation could be presented by looking at the previously published factors (Pointed-P1 and Cut) involved in receiving-cytoneme

promotion/inhibition (Du et al., 2018). In this occasion from the perspective of the FGF producing cells.

Ans: Thank you for this question and for the appreciation of our previous work on the self-generation of cytoneme-mediated FGF signaling patterns (Du et al., 2018). Mechanisms of scaling spatial patterns of morphogens in coordination with growth is a fascinating topic and the cytoneme-mediated bidirectional signaling could provide one of the simplest mechanisms for this important problem. We would like to address this tissue-level patterning question in the future by making endogenously expressed variants with thorough experimental validations. Considering the goal of this study on testing the role of GPI-Bnl in controlling cytoneme pathfinding and contact-dependent signal release/signaling, we removed emphasis on tissue levels patterning, including signal/signaling gradients with overexpressed proteins.

Please see additional reviewer document.

6. *As a minor comment, in line 380-381 the sentence should be changed to “might be sufficient” as no detailed mechanism is described.*

Ans: Thank you for this correction. The modified sentence is on page 14, the last sentence, and page 15-first sentence.

7. *Fig 4J the intensity of the red channel should be increased or split into the two channels; it is not easy to distinguish the dpERK activation.*

Ans: Thank you for this correction. In the revised manuscript, this data is presented in Fig. 9B,B'. We presented an image with better resolution and also showed split channels highlighting dpERK staining.

8. *The interaction between ligand and receptor is shown associated to cytoneme contacts. It would be interesting to discuss how and where the endocytosis process happens in the receiving cells.*

Ans: This is an important point considering the nuclear MAPK signaling, the robustness of which requires endocytosis. We discussed this point in the 6th paragraph of the Discussion.

Reviewer #2 (Remarks to the Author):

In the manuscript, Du et al. reported that cytonemes are not only present in the signal receiving cells in the ASP (Bth FGFR expressing) but also are associated with the sending cells that express FGF ligand Bnl. They showed that the distribution of Bnl and effective FGF signaling depends on a C-terminal hydrophobic region that allows the ligands to be anchored to the cell membrane through a GPI moiety. Furthermore, GPI anchorage of Bnl facilitates tissue-specific release at inter-cytoneme contacts and diffusion of its FGF domain-containing fragment into the ASP upon interacting with FGFR on the cytonemes, leading to desirable signaling outcomes during the downstream events. The authors also suggest that Bnl acts as a CAM (cell adhesion molecule?) and signals bidirectionally. No references are included to give this CAM-signaling context; and the angle of bidirectional signal does not seem well supported. On the other hand, the mechanism demonstrated in the manuscript, by which FGF signaling is spatiotemporally regulated at the level of ligand dispersion through its lipidation, is compelling, and potentially provides important insights into how

signaling pathways are regulated through membrane-association of ligand. The experiments are well designed and carefully carried out; nevertheless, most of the *in vivo* experiments rely on ectopic expression instead of mutant allele characterization which is a limitation. It is therefore hard to know if native Bnl function requires the GPI. Lastly, though the main text is logically written and easy to follow, the clarity of figure could be improved; parts are a bit disorganized and several charts lack consistent labelling.

Ans: We thank the reviewer for all critical comments and suggestions! We tried to address all the comments to the best of our ability. Before addressing the major concern, let me first briefly clarify several critical points, raised above. First, we are sorry that the previous version of this paper did not clarify many aspects and had kept the references within a certain limit. We missed citing many important and relevant papers. Stepanik et al was published when our paper was in preparation/circulation. In the revised version, we added additional references for GPI, CAM activity, and the discovery of the TM-tethered Pyr. We added additional experimental data showing CAM-like trans-coupling of Btl and Bnl and bidirectional signaling, from both *in vivo* and *in vitro*. We also added supplemental data showing Bnl isoforms and their surface presentation by GPI anchor. We have re-written the Introduction to clarify the goal and Results/Discussion to address all the comments.

Major concerns:

(1) Bnl is known to have two splice variants expressed in embryos affecting the very C-terminal region of Bnl. One form (being described in the manuscript) is predicted to have a hydrophobic region, while the other is not (see Stepanik et al. *Curr Biol* 2020). Are both forms also expressed in the wing disc? The presence of the shorter variant that does not possess the hydrophobic region might represent yet another layer of how FGF signaling is regulated during development. It's intriguing to think that the two different Bnl isoforms could carry out the combined signaling properties of the other 2 *Drosophila* FGFs to their dedicated FGFR Htl [i.e. TM domain-containing Pyr and non-TM Ths (Stepanik et al., 2020)]. The authors should clarify and expand on that point, perhaps in terms of the coordinated use of GPI-anchored Bnl, post-GPI cleavage Bnl, and "freely-secreted" Bnl in the dynamics of ASP morphogenesis. At minimum, the authors should cite this other study which pointed out the C-terminal hydrophobic domain in Bnl and previously suggested it might be membrane-associated.

Ans: (a) bnl isoforms: Endogenous native Bnl (probed by either Bnl antibody that binds to both isoforms or the Bnl:GFP^{endo} knock in that marks both isoforms) was shown to be transmitted from the disc to the ASP through cytonemes and cytonemes are essential for the inter-organ Bnl movement and signaling (Du et al *Elife* 2018; Roy et al, *Science* 2014). We started the current investigation with goals to understand the processes that produce cytoneme contacts and control contact-driven Bnl release. Although the probable differential expression and roles of Bnl splice variants are intriguing to think about, we did not intend to study these aspects. Of course there will be many variations and not all cytoneme-dependent signals are GPI-anchored. However, in the context of GPI anchoring, we do agree that the C-terminal differences between Bnl-PA and PC isoforms might be important. As suggested, at least we should clarify if the PC variant is (i) expressed on the source surface like the PA variant, and (ii) whether/how PC variant is retained on the source surface for signaling.

(i) Native Bnl is GPI-anchored: We now clearly indicated in the text that we used a Bnl antibody that detects both Bnl isoforms and this reagent helped us to show that the native Bnl exchanged

between the source and ASP is GPI-anchored (Figs. 5A-C; 3D-E', S4A,A'). Similarly, endogenous Bnl:GFP^{endo} (Figs. 1E,E; S1B; 2F-H; S4A), which constituted of both isoforms, had similar source surface distribution as observed for native Bnl and were shown to move through ASP cytonemes (Du et al 2018).

Bnl is essential for morphogenesis in many systems, and are expressed in all essential organs that require tracheal supply for oxygen. In all these prior studies, Bnl's role was established by expressing the original *bnl-PA* cDNA construct. Bnl-PC, although was predicted in Flybase, has not been verified for function in vivo. Because of these reasons, we did not first examine the Bnl-PC isoform. Based on the Flybase modENCODE data, PA and PC are expressed in the larval discs. We also performed isoform-specific RT-PCR from total wing-disc RNA (see Supplementary data) to find that both PA and PC isoforms are expressed in the wing disc.

(ii) We compared secondary structures and biophysical properties of two isoform using various methods (eg. ProtScale analyses (EXPASY) with methods such as Eisenberg; Abraham & Leo; Kyte & Doolittle for hydrophathy). These analyses consistently showed that the PC variant has a hydrophobic tail but is shorter than the PA form. However, the hydrophilic spacer immediately upstream is conserved in both PA and PC (please see Fig. S4A-A"). We generated a UAS-Bnl-PC:GFP and found that the protein has PIPLC-sensitive cell surface distribution, like a GPI-AP (Fig. S4B-G). Based on these analyses, we concluded that both PA and PC are GPI-APs. We did not further pursue Bnl-PC. An important lesson for us was that the absence of a TM-like hydrophobic domain as detected by sequence analysis tools should not always be the basis to predict "free secretion" of a protein. Bnl signaling is paracrine and Bnl-PC lacked a TM, but there is no direct experimental evidence for "free secretion" of Bnl-PC in any published document. Instead, it is GPI-anchored, which is consistent with cytoneme-dependent Bnl uptake and signaling as observed before (Du et al, 2018).

b) Stepanik et al. Sorry again for the failure. We now cited the significant discovery of a TM-tethered FGF, Pyramus, and probable roles of TMD in regulating Pyr signaling.

c) Discussion on the similarities between TM-Pyr & free secreted Ths with GPI-Bnl-PA and free-secreted Bnl-PC. Since both Bnl-PA & PC are GPI-anchored, and since the interorgan transport and signaling of endogenous Bnl require cytonemes (Du et al 2018, Roy et al 2014), we did not speculate functions such as free release and dispersion. However, we do agree with the reviewer that it would be interesting to know why two different isoforms of Bnl are needed, especially if both are functionally very similar to each other and GPI-anchored. We do not know if all GPI-anchors are the same in relation to the lipid and carbohydrate architectures. A speculation could be that different SS can add different types of GPI moiety, but there are no prior study in this regard.

(2) The expression level of different mutant forms of Bnl need to be quantified. Modification on the C-terminus of Bnl may affect the stability and other functional cleavage sites of the protein. Therefore, it is particularly important to make sure that all those mutant forms are expressed at comparable levels. Specifically, for Bnl:GFP- ω m and Bnl:GFP Δ C, a western blot should be done to make sure that they are expressed at reasonable levels and to what degree each is secreted into the extracellular space (supernatant).

Ans: a) Comparable levels of expression: Sorry that we did not clarify the quantitative methods before. We now added experimental details, including expression levels in Methods and main texts. Briefly:

For S2 cells: We used identical conditions/expression. However, expression from transient transfections in cultured cells can be variable despite of the identical conditions used. *Therefore and very importantly, we normalized all quantitative values from S2 cells with internal controls to avoid variations from cell to cell.* For example, to understand PIPLC-dependent removal of surface protein, we compared the ratio of the levels of cell surface GFP-tagged protein (probed by non-permeabilized anti-GFP immunofluorescence levels) to the total protein levels (total GFP fluorescence levels) expressed in the same cell. The ratios were compared between cells/conditions detected using microscopy (for protein localization) and FACS (~5000 cells/rxn, for high cell number counts). Note that this quantitative approach used is rigorous - we probed the same X:GFP protein twice, once with non-permeabilized immunofluorescence against GFP (for surface fraction) and once with GFP fluorescence (for total expression), and the ratio of the two were compared between samples. Only for untagged Bnl, we normalized its surface expression relative to the co-expressed CD8:GFP.

For detecting the GPI-anchored cell surface display, we used a direct approach of visualizing the cell surface protein using non-permeabilized immunostaining. The presence of surface Bnl variants in the absence of PIPLC and removal of surface protein in the presence of PIPLC determined whether the protein is GPI-anchored or not. On the other hand, if the Bnl variants are absent from the surface without the PIPLC treatment, the protein is predicted to be non-GPI-modified. The reasons for the absence of a protein on cell surface even without the PIPLC treatment could be many - could be due to its readily/free secretion (e.g., secreted GFP (Fig. S3J,J'), Bnl \square C (Figs. 4D',G)). It could also be due to ER retention/defects in externalization (e.g., omega mutant, Figs. 4G; S3I). In both cases, theoretically, the protein is not GPI modified. Note that we did not investigate why a protein is not present on S2 cell surfaces and we used *in vivo* analyses to address this question as explained below (revised Fig. S5A-B).

For transgenic flies: We used Bnl:GFP, Bnl:GFP \square C and Bnl:GFP \square C-TM lines that showed comparable expression levels. We added a Table in the Supplementary data showing the lines used and their expression levels (methods in Sohr et al 2019). Again, all our quantitative estimates of surface protein levels and signaling outcomes are normalized with internal controls to avoid variations among samples. For example, to compare between Bnl variants, we compared the ratio of surface protein (red, anti-GFP non-permeabilized immunofluorescence) to total expression (total GFP fluorescence) in the same disc source area (Fig. 5G-M). Similarly, to assess MAPK signaling patterns of different Bnl variants, we measured the number of signal recipient cells (cells with Bnl:GFP variant puncta) that induced MAPK. The correlated patterns between signal reception and signaling per cell/tissue were then compared between conditions (Fig. 9A-D, we now clarified these detection methods in the text).

b) Externalization of Bnl:GFP- ω^m and Bnl:GFP Δ C. We presented data on Bnl:GFP Δ C and Bnl:GFP- ω^m externalization *in vivo* (Fig. S5A-B). Figures 8 & S5 showed that Bnl:GFP Δ C spreads out randomly through the extracellular plane of the wing disc tissue and ASPs also received Bnl:GFP Δ C. Thus, Bnl:GFP Δ C is freely released and randomly dispersed in comparison to the same protein with a TM-tether (Bnl:GFP Δ C-TM) or a GPI-tether (Bnl:GFP).

In contrast, the omega mutant was not externalized or received by the ASP in our experiment (Fig. S5B). It was known that the mutation in the omega site of a pro-GPI-AP leads to the ER retention of the signal (we added the reference in the text). In comparison, either a C-terminal deletion of the signal sequence or replacement of the signal sequence with a TM was known to have no detectable effects in protein stability or trafficking (we cited reference). Thus, all our findings are consistent with the existing literature.

c) Stability of C-terminal mutants: In Fig. 4B-C, we showed experiments that Bnl:GFP₃CherryC (C-terminal mCherry-tag) molecules are cleaved intracellularly, and while the mCherry tag is retained inside the cell (not membrane localized), the processed Bnl:GFP₃ portion is externalized and delivered to the ASP. Similarly, Bnl:GFP₃C and Bnl:GFP₃C-TM were trafficked to the surface of the wing disc source and delivered to the ASP. Based on these data, it is unreasonable to think that modulation of Bnl's C-terminus would affect its stability. Secondly, we provided a number of evidence that Bnl's SS is for GPI anchoring, which facilitates Bnl's externalization and polarized inter-organ dispersal/signaling via cytonemes. To avoid any confusion, we tried our best to explain this pathway in the text. We also added analyses for secondary structural domains of all the fusion proteins in the new Supplemental data to show that the C terminal FP tagging or deletions of SS or addition of TM in Bnl has least probable effects in protein domain.

The deletion of the extreme C-terminal signal sequence and addition of a TM replacing the deleted portion is a standard practice to characterize GPI-APs and was experimentally shown to have little effects in protein intracellular stability or trafficking in published literatures. Few common examples: similar analyses were used to characterize *Drosophila* GPI-APs like acetylcholinesterase, Dally/Dlp, and MMPs. In addition, we showed GPI-anchoring of bGFP-GPI - a secreted GFP added with Bnl's C-terminal SS portion, indicating the function of the SS (Figs. 4, S3J,J').

*(3) Provide further evidence that the GPI anchoring is functional in vivo for Bnl. Data presented does strongly suggest that Bnl could be GPI-anchored through the computational identified glypiation site S741. However, the presence of a TM domain is not completely ruled out, and deletion of the 40 aa C-terminus could very well disrupt a potential cleavage site that is important for release of the ligand if TM is functional, even outside of GPI utilization. And indirectly, a protease that is responsible for cleaving off the TM domain could be GPI-anchored, thus PI-PLC treatment may activate the protease and release Bnl from the membrane. That being said, GPI anchorage is still a favorable mechanism that regulates the distribution of Bnl. To make the case more convincing, Bnl:GFP Δ C can be linked to a known glypiation sequence to rescue the mutant phenotype. Additionally, Crispr bnl mutants bearing mutated glypiation sites (*w^m*) can (and should) be made to confirm that those sites function endogenously (see point 4 below).*

Ans: We appreciate multiple different points and alternative hypotheses at various levels of functions. This helped us to clearly present our findings and re-write the manuscript.

a) About in vivo evidence of GPI anchor: We did provide evidence that GPI-anchoring is functional for Bnl in vivo (Figure 5). In fact, we, for the first time, developed a unique protocol to perform the PI-PLC assay in live ex-vivo cultured organs to directly visualize PIPLC-sensitive cell surface distribution of Bnl (native form, all isoforms). Similarly, we also compared the surface display of various Bnl:GFP constructs in the wing disc source (Fig. 5G-M). Note that the

molecular and cellular machinery of GPI-anchoring is conserved in all cells. Since the discovery of PIPLC in 1976, a vast number of GPI-APs have been discovered in diverse organisms for a variety of functions, including CAMs, ectoenzymes, and signaling proteins. Based on our literature survey, most (if not all) GPI-APs were characterized using cell culture-based biochemical/cellular assays. To emphasize the in vivo work, we now separated the in-vivo data in the revised Figure 5. We also showed a surface display of Bnl constructs in the ectopic tissue source (Fig9. I-L).

b) Stability of proteins with C-terminal modulations: We addressed this point in previous comments (#2). Deletion of the SS of GPI-APs was known to facilitate protein secretion. We now added additional evidence that the Bnl:GFP□□ is secreted and dispersed. With a combination of in vitro and in vivo experiments, we also showed that the protein is active. We did not pursue omega mutant due to its ER retention (please see the previous comment).

c) Ruling out TM-tether; PI-PLC activating another protease that removed a TM tethered Bnl: We want to remove any concern on our experimental data, if possible. In Biology, unpredictable things are possible, but we provided much experimental evidence of GPI-anchoring of Bnl, starting from a native Bnl to different Bnl variants and engineered constructs harboring Bnl's SS and SP showing that Bnl is a GPI-AP in vitro and in vivo (Figs. 4, 5; S3,S4). Therefore, it is unreasonable to think of a TM-tethered Bnl. We tried to remove any confusion that might have been caused by our unclear/brief description of experiments.

In addition to the textual changes, I want to explain a few basic concepts in relation to our experiments:

(i) As illustrated in Figure 4, the C-terminal hydrophobic region constitutes only a portion of the entire signal sequence (SS) for GPI anchorage. The C-terminal hydrophobic part of the SS is cleaved off in the ER and is not known to be externalized (Fig. 10). On the other hand, uncleaved pro-GPI-APs (i.e. omega mutants, without GPI addition) are retained in the ER, and C-terminal deletion of SS and replacement of SS with a TMD do not affect signal trafficking. We cited these previous findings.

(ii) Finally, most proteases are activated by proteolytic removal of their pro-domains by Furin-like enzymes, and PI-PLC is a lipase that specifically acts on the phosphatidylinositol of GPI moiety. Moreover, activation of any GPI-anchored ectoenzymes has not been shown to be dependent on their PIPLC-dependent shedding.

() I should mention that the TM-prediction analyses using bioinformatic tools often can detect that many signals such as Hh or EGF/Spitz contain TM-like hydrophobic domains, similar to Pyr/Bnl. These TMDs in Hh/Spitz (EGF)/Ephrin-A (and Bnl) are known to be for ER/Golgi localization, intracellular cleavage, and lipid addition, which, in turn, controls intracellular and intercellular signal trafficking. So, sequence analyses of a protein are indicative of a function, but we have to be careful of predicting the functions as none of these TM-like domains are for cell membrane/surface localization with a TM.

From Bnl, we learned that most of the regulatory functions are encoded in the unconserved and unstructured regions, which cannot be predicted by bioinformatics as there are no consensus for these regions. Our identification of GPI anchor was based on an accidental

experimental observation stemming from fusing FP tags at various positions of Bnl during my post-doc, which was then extended in my lab very thoroughly (Fig. 4B,C). We relied on bioinformatic predictions for the omega site. Although we mutated the predicted omega sites and observed desirable outcomes of ER-retention, we softened the conclusive sentences. SS of *Drosophila* GPI-APs needs to be thoroughly studied for consensus and variations. Because, the GPI-AP signal sequences (including ω -sites) are known to have little sequence conservation, and its extreme C-terminal positioning is not an absolute requirement. We tried many different prediction analyses for Bnl's omega site, but only one GPI-predictor could predict the GPI anchorage of Bnl. At the same time, the same tool is unable to predict a known *Drosophila* GPI-AP, glypican (*dlp*). So, bioinformation prediction should be carefully assessed, especially for predicting GPI- or TM- anchoring.

(iv) There are subtle differences between the TM domain and the hydrophobic SS for GPI addition. Based on accepted models, TM domains are usually flanked by charged residues carboxyl-terminal to the hydrophobic domain that acts as stop-transfer signals preventing full translocation through the translocon pore into the ER. The transmembrane domains are then partitioned into the lipid bilayer by lateral gating of the subunits of the translocon pore. In contrast, signal sequences of GPI-APs are fully trans located into the ER lumen before being used as a substrate for glypiation. Experiments have shown that the hydrophobic domain of a GPI-AP SS cannot function as a transmembrane domain even when flanked by charged residues. It is fascinating to imagine that so many outcomes can be programmed just at the levels of a protein sequence. Finally, it is hard to distinguish between TM-tethered and GPI-tethered proteins based on their membrane/cells surface localization unless we do the PIPLC assay.

Based on the thoroughly analyzed experiments, rigorous quantitation with internal controls, we are very confident to conclude that Bnl is a GPI-AP (both in vitro and in vivo) and this modification is necessary for the bidirectional Btl-Bnl signaling via cytonemes. We also added additional data for CAM activity, as suggested by the reviewer. This new modification has significantly strengthened our model.

d) "to make the case more convincing, Bnl:GFP Δ C can be linked to a known glypiation sequence to rescue the mutant phenotype." We did show the rescue experiments by adding the TM domain from the mammalian CD8a protein to *Bnl:GFP Δ C*. Comparative analyses of *Bnl:GFP Δ C*, *Bnl:GFP Δ C-TM*, and *Bnl:GFP* (GPI modified) showed that *Bnl:GFP Δ C-TM* can rescue many of the functional attributes of *Bnl:GFP Δ C* at various levels. For instance, we showed that the TM variant could rescue activity at various levels: source surface localization, cytoneme induction and reciprocity, target-specific signal distribution, and signaling both in vitro and in vivo.

Based on my literature survey, the replacement of Δ C with a TM-tether is the standard way to show functional rescues of Δ C. Because: (i) GPI-anchors provide a balance (or biphasic control) between two extreme states - the complete free secretion of the protein (e.g., AC variants) and absolutely no secretion like the TM-protein (e.g., AC-TM variants). GPI-AP functions can be best-interpreted relative to the two extreme states. (ii) More specifically, since the functional property of the protein is assigned by the covalently attached GPI-moiety (a glycolipid), no matter what SS we add to the protein, it will still be added with the same/similar GPI-moiety produced in the ER of the expressing cell (see Fig. 10). So, replacing Bnl's SS with another SS will theoretically produce the same GPI-Bnl that we wanted to modulate for functions. So, it won't be able to test the functionality of the GPI anchoring. We added more

clarification on GPI anchoring in Fig. 10 to explain. Notably, we and others had used a GFP-GPI protein (from S Eaton's lab) as a control GPI-AP in *Drosophila* cells, and it has a mouse protein SP and SS added to the GFP (Fig. 4D'-F). We do not find any visible difference between this GFP-GPI and our bGFP-GPI that has Bnl's SP and SS. It is interesting that the SS can be widely variable but still can be used for glypiation in cells across species/taxa.

However, if a concern is about Bnl's SS functionality, I would like to emphasize that we extracted the N-terminal SP and C-terminal SS of Bnl and added them to the N and C terminal of a GFP, respectively. This chimeric GFP (b-GFP-GPI) was converted into a GPI-AP (Fig. 4D'-F). The same protein without the C-terminal SS (named secGFP) was not retained on the cell surface (Fig. S3J,J'). Thus, Bnl's SS acts as a glypiation signal. We added an explanation.

(4) The case for GPI regulating endogenous Bnl activity must be strengthened through further characterization of Bnl-ΔC (C-term hydrophobic domain deletion) and Bnl-ω.m (GPI predicted residue mutant). Crispr/Cas9 can be used to create mutants (possibly clones) to determine if the mutations affect Bnl endogenous function; there is some concern that ectopic expression experiments for the most part form the basis for conclusions. In addition, Bnl-ω.m should be assayed in ectopic expression experiments side-by-side with the C-term deletion in order to ascertain whether the GPI linkage is affecting function (rather than some other functionality associated with the C-terminal domain).

Ans: a) Bnl-ω.m: As addressed in comment #2, we presented in vivo data on both Bnl:GFPΔC and Bnl:GFP-ω.m (Figure S5). In these experiments, the extracellular GFP-tagged signals were detected with non-permeabilized anti-GFP staining. Results showed that the Bnl:GFPΔC spreads out through the extracellular plane of the wing disc. Extracellular Bnl:GFPΔC was also on the ASP and was internalized (green only puncta are) in the ASP, indicating signal uptake. Du et al 2018, showed that internalized Bnl puncta in the ASP colocalized with receptors in endosomes. In comparison, Bnl:GFP-ω.m was poorly localized in the extracellular surface and barely received by the ASP. Our data is consistent with previous reports that the mutation in the omega site of a GPI-AP leads to its ER retention. In contrast, the deletion of the C-terminal SS of the same protein leads to normal trafficking leading to free secretion. We did not pursue omega mutation as it is not externalized.

(b) Functional significance with CRISPR mutation. We appreciate the suggestion as we do plan to understand the link between the asymmetric Bnl presentation/reception to asymmetric tissue patterning. However, generating CRISPR mutations and establishing stocks (especially for unmarked mutations) is time-consuming and requires thorough genomic and phenotypic analyses at various levels to avoid the diverse range of issues including off target mutations that emerge during the process. Based on our previous experience with Furin-uncleavable mutant Bnl:GFP^{endo} (Sohr et al, 2019) and its subtle steady state phenotypes in the tissue/organismal level (eg. lethality, reproduction etc), I do not see a major advantage of CRISPR mutants to test subtle regulatory functions in our model. We hope to achieve CRISPR mutagenesis in the future and examine defects in tissue-level organization after a thorough, careful investigation. This decision, however, DOES NOT alter the quality and conclusion of the data presented here that we thoroughly analyzed for the last 5-6 years. I can explain why the quality of the data on the role of GPI anchor in cytoneme-mediated signaling is not affected by the lack of an overall mutant:

i) Functionality of GPI anchoring can be assessed at various levels of organizations. As discussed before, the immediate function of the GPI-moiety is to provide a subtle balance between two extreme functions - free/random secretion (paracrine functions) and no-secretion (TM-like membrane tethering). Both extremes are critical for Bnl functions. Therefore, the working model we tested in this study is specifically focused on the ability of GPI and non-GPI modified Bnl to induce CAM-like bidirectional Btl-Bnl's interactions (which require membrane tether), and contact-dependent Bnl exchange and signaling (which require signal release). In this context, we performed in vitro and in vivo experiments that can easily and convincingly be interpreted for CAM activity, signal release, and signaling activation. Considering the sequence of events as described in the model, we modified the texts to first describe experiments on Bnl's CAM-like activity (Figs. 3,6,7) followed by its release (Fig. 8) and signaling (Fig. 9).

Based on our previous experience, the subtle regulatory functions at the molecular and cellular levels are challenging to probe and interpret from the loss-of-function genetic mutant organisms, especially because of the complex and unknown compensatory mechanisms at various levels (genome, cell, tissue, organism levels) that might be required to make an intact mutant fly. For instance, the only example of mutant analyses of GPI-APs in the intact organism was known to me is for the *Drosophila* acetylcholinesterase (*ace*; AChE). Rescue experiments of *ace* null by TM-AChE (large SS deletion and replacement by TM) or GPI-AChE minigenes showed that TM-AChE could in general substitute for GPI-AChE minigene for viability and normal behaviors, but the adult flies are short-lived, less reproductive, and had subtle changes in behaviors. The defect in a cholinergic neuron might be compensated at the cell/tissue levels by different neuronal connections. Such mutant analyses, without a thorough understanding of the entire system, would not immediately provide a picture of the essential functions of GPI anchoring AChE in spatiotemporally regulating both the enzymatic (cholinergic) and non-enzymatic roles of the AChE protein at the cholinergic neuronal synapses.

In this context, it is important to highlight what we did and how those analyses helped us to interpret functions:

ii) GPI's Role in cytoneme polarity and matchmaking: Bidirectional cytoneme-mediated contact matchmaking and signaling response were identified in normal/WT and various mutant condition, without signal overexpression (Figs. 1, 2, 3). We also showed Btl-Bnl-dependent bidirectional matchmaking at the molecular and cellular levels using in vivo clonal analyses and in vitro cells. These observations had set up standard phenotypes for us that we used to assess the activity of Bnl with or without a GPI tether. Clearly, the loss and gain of membrane tether of Bnl correlated with the loss and gain of its CAM-like bidirectional cytonemal matchmaking and exclusive tissue-specific release. To reliably compare the effects on cytoneme polarity, numbers, and length under the influence of GPI- and non-GPI-modified Bnl, we counted cytoneme numbers per area/perimeter for each sample and compared the normalized values between conditions. So, sample-to-sample or condition-condition variations are eliminated. We also added additional data on bidirectional CAM-activity by using clonal analyses (in vivo) and S2 cell CAM assay (thanks to the reviewer's comments). (Figs. 6,7).

) GPI's role in the signal release: Based on our previous experience with endogenous Bnl:GFP and Furin-uncleavable Bnl:GFP mutant (see Du et al, 2018, Sohr et al 2019, also an example is shown in the current paper - Figs. 1E,E'), it is extremely difficult to visualize/quantitate the levels of endogenous molecules in the source cells, due to either the low expression levels or the

rapid signal turnover. Detection of rapid nanoscale organization of GPI-APs requires specialized microscopic light sources and analyses. Therefore, making an endogenous mutant with a lot of effort is not expected to dramatically change our ability to interpret signal release/dispersion. *(Please also see the reviewer's/editors only document.)*

Therefore, the simplest way to observe the release and dispersion of a signal would be to express the protein from the FP-marked point-source in the wing disc and see whether the signal moved away from the source to the neighboring region of the disc or to the ASP (Fig. 8A-E). We did this experiment and compared the control profile of Bnl:GFP (which is GPI-modified) with Bnl:GFPAC and Bnl:GFPAC-TM, under the same condition and levels of expression. Clearly, in comparison to control Bnl:GFP, Bnl:GFPAC moved non-specifically through the extracellular space, and the addition of a TM domain to AC rescued target-specificity in dispersion. We also showed that the rescue of target-specificity correlated with the rescue of contact-dependent cytoneme mediated signaling. Secondly, for the quantitative estimate of signal dispersion over time. We followed a previously standardized technique and unique features of Bnl protein (Sohr et al, JCB, 2019, Bio-protocol, 2019). Note that the quantity of signals received by the ASP from $t=0$ to $t=x$ is normalized with internal control as described in Fig. 8F-P). This allowed us to accurately predict a relative change over time without any sample-to-sample variation. All results presented in Figure 8 are sufficient to suggest that membrane anchoring ensures target-specific signal exchange by inhibiting random dispersal. This Figure was initially presented in the supplement, but it is an important result.

iv) Signaling: Previous studies comparing endogenous and ectopic expression of Bnl:GFP or mutants from original and clonal sources (Du et al, 2018, Sohr et al 2019) had set up two standard phenotypes: (a) an exclusive target-specific movement of signal from the disc to the ASP (not in the non-specific disc area), irrespective of the mode of Bnl:GFP expression; and (b) precise spatial correlation of signal reception and signaling in growing ASPs. Therefore, in this paper, we expressed signals from a point source and asked questions: (a) whether ASP cells that received the signal (i.e., that contain GFP-tagged signal) from the source also induced signaling (anti-dpERK). For accurate interpretation, we used mosaic analyses (Fig. 9A-D).

The genetic mosaic technique represents by far the most powerful and reliable tool for deciphering the inductive function of a signaling protein expressed from small clones without adversely affecting the overall health or viability of the animal. This is how Dpp/Hh producing clones of cells in the recipient field of wing disc cells were shown to induce ectopic mirror-image duplication of the wing. Similarly, Bnl-expressing clones were shown to attract ectopic tracheal branch migration toward the Bnl source (Mark Krasnow lab), which connivingly established Bnl's chemotactic activity. We also have shown that a small 1-2 cell size clone of Bnl:GFP in the ASP epithelium can reorganize ASP cells surrounding the clonal signal source and guide them to form an ectopic branch budding out of the existing ASP (Du et al. 2018). In the same paper, we had endogenous Bnl:GFP^{endo}, but without the gain-of-function clonal analyses (GOF) of Bnl:GFP, Bnl's organizer-like inductive functions could not have been directly shown.

Using the clonal techniques, we compared the inductive power of Bnl:GFP, Bnl:GFPAC, and Bnl:GFPAC-TM. To accurately interpret our data, we analyzed Bnl GOF clones only in those areas of Btl-expressing ASP/tracheal epithelium (ASP stalk and TC) that normally do not receive Bnl and do not activate nuclear MAPK signaling (Du et al., 2018) (Fig. S7D). Importantly, we did not simply compare the dpERK levels induced by different signals. *We*

measured the percentage of signal recipient cells (cells with Bnl:GFP puncta) that induced MAPK in each tissue sample. We compared this correlation of spatial patterns of signal reception and signaling on a cell-to-cell basis, irrespective of signals/conditions (Fig. 9A-D). This provided a clear measurement of Bnl function solely on the basis of the altered anchoring domain.

Similarly, the induction of tracheal migration into an ectopic Bnl source is a very powerful technique to assess the signaling efficacy of Bnl (Sohr et al. ;2019; see modified text, Fig. 9E-L). In this experiment, we used larval salivary glands because it is a 100% trachea-free and *bnl*- and *btl*- expression free non-essential organ. So, when we see tracheal growth into the organ, it is purely due to the ectopically expressed Bnl in this organ (Sohr et al.;2019). We also showed that the levels of surface-tethered Bnl:GFP variants on the salivary glands were positively correlated with the extent of tracheal growth into the organ. Note here, the levels of surface proteins were normalized with the levels of total protein expression in the gland.

Finally, in all these analyses, a major advantage was the uniqueness of Btl-Bnl signaling. Btl is the only receptor for Bnl and Btl and Bnl are expressed in different tissues/organs. While the wing disc expresses Bnl, it lacks Btl expression. Similarly, while the ASP expresses Btl, it lacks Bnl expression. So, the changes in the ASP due to the change in the disc Bnl expression is from the direct effect of Bnl signaling, not from a secondary effect due to the defect in the wing disc or other tissue development.

*(5) Clarify the mechanism by which GPI-anchored Bnl is released upon receptor binding. The results from PI-PLC assay do suggest that Bnl is associated with cell membrane through a GPI anchor, however, it is unclear how it is released in the context of FGF signaling. It will be helpful if the authors can provide more clues by looking at Bnl distribution in *btl* mutants or expand on the topic in discussion.*

Ans: Thanks for the suggestion. We discussed and speculated a mechanism. We provided a Bnl^{ex} distribution in Btl:DN condition in the trachea (please see Fig. 3D-E' and Fig. S2C-C'''). Polarized clustering of Bnl^{ex} is observed when cytoneme contacts are formed.

(6) Is the signaling truly bidirectional between Btl and Bnl? Besides the finding that ASP- originated cytonemes can influence cytonemes from the wing disc, what is the additional evidence that supports reverse signaling events in ligand-presenting/ wing disc cells? Is there a proposed nature of the information relayed back to the Bnl-producing cells? The authors also suggest that Bnl acts as a CAM (cell adhesion molecule?) and signals bidirectionally. No references are included to give this CAM-signaling context; and the angle of bidirectional signal does not seem well supported.

Ans: We thank the reviewer for this challenging question. We performed additional experiments to prove Bnl shows CAM-like bidirectional signaling activity. We added additional discussion to clarify. Here are the results:

a) All data panels in Figure 3, Figure 6, and Figure 7 and related texts in the revised manuscript show both in vivo and in vitro evidence supporting CAM-like synaptic Btl::Bnl trans-pairing, inducing selective adhesion of source and recipient cells. We also showed that such cell-cell adhesion could selectively induce MAPK signaling in the trans-paired recipient cells.

b) Figure 7F-M': We provided additional in vivo evidence of contact-dependent Btl-Bnl bidirectional signaling using clonal analyses. Ectopic Bnl-TM clones could polarize toward the ASP to form signaling contacts and maintain/reinforce the ASP-directed polarity, likely by contact-dependent reverse signaling. Similarly, ectopic Btl clones and wing disc Bnl sources polarize toward each other to matchmake via cytoneme contacts. I would like to add here that imaging cytonemes (<200 nm diameter thick) in live growing tissues is one of the most challenging experiments. However, thanks to the simplicity of Btl-Bnl signaling and evolving technology, we could do these experiments.

(7) Presentation of data/style: (i) Fig3, take out the two black panels in C'; assuming this experiment was not done (why else would the green signal be missing?). (ii) Label L'-Q' on top with α GFPex instead of genotypes. (iii) In general, labelling, styles and formats of the charts in the manuscript are not consistent, different colors are used to represent the same genotype. Would be more friendly to the readers if the authors could improve on the presentation of their statistics. In the zip file for reviewers 12 movies are included; this made analysis of the movies a bit confusing as the files did not contain labels.

Ans: We are terribly sorry for these unintentional issues with the movie files. We thank the reviewer for all the above corrections. We corrected all related figures and graphs and tried our best to be consistent. We will be careful to upload individual movie

files. **Minor suggestions to improve clarity:**

1) Line 70: "signal-exchanging cells ^{can} extend actin-based signaling filopodia" (though it is clear that Bnl-Btl signaling uses cytonemes in this context, the same may not be true for all ligand-receptor pairs)

Ans: Thank you for the suggestion, we modified the text.

2) Lines 92-95: sentence is awkward in terms of subject/verb agreement; making "contacts" singular in lines 93 might fix this. It is difficult to interpret if they are saying in lines 94,95 if they are saying that all spatiotemporal cues in morphogenesis are controlled by molecular and cellular processes that control cytonemes, or just mechanisms utilizing cytonemes

Ans: Thank you for the suggestion, we modified the text.

3) Line 123: What is meant by "unbiased interpretation" of tissue-specific dispersion? It may be helpful to more explicitly connect this with the idea following in lines 125-128 that varying fixation/detection conditions can show different pools of Bnl protein. It may be more straightforward to simplify the sentence in lines 122-123 as "...is a simple genetic system for studying dispersion of Bnl produced in the wing disc..."

Ans: Thank you! We modified.

4) Lines 260-263:

It should be mentioned that this isolated region of Bnl has already been identified to anchor an FP to the plasma membrane when exogenously expressed in S2 cells (see (Stepanik et al. 2020) above).

Ans: We cited Stepanik et al for the discovery of TM tethered Pyr.

This part is for editors/reviewers:

Dr. Stathopoulos's pioneering work on Pyr and This is a role model for many of us to be inspired to work on FGF. While it is worth discussing the major finding on Pyr by Stepanik et al, I do not find it appropriate and worthwhile to discuss several minor issues based on a single inconclusive result on Bnl presented in the supplement of Stepanik et al. The supplementary experiment shown in Figure S6 of Stepanik et al., is inconclusive to me, because - it is not a simple FP-tagged Bnl that is tested, it is a functional Pyr-Bnl chimera, and chimeric signaling proteins can have complex behaviors introduced by both. So, theoretically, the interpretation on Bnl localization was not completely free from the unknown regulatory regions of Pyr or vice versa. For instance, a possibility could be that an N-terminal large size FP tags (~25 kDa) juxtaposed between the SP and active 'FGF domain' of Pyr might affect externalization of the chimeric protein. The chimeric FP-Pyr-Bnl protein, when expressed and visualized in transfected cells, the mCherry fluorescence was diffused everywhere in the cytoplasm (Fig. S6 of the paper). It is hard to predict if there is any actual trans-membrane localization. An experiment with soluble mCherry expressed alone in S2 cells might show similar fluorescence pattern. A non-permeabilized immunostaining (similar to the beautiful paper by Tulin et al (2010)) for Pyr or mCherry would have ruled out the possibility of inhibition of externalization but was not shown. It is possible that the Stepanik et al had ruled out such possibilities, but it was not apparent in the published text.

These issues are non-essential for discussion, especially if we think about the beautiful thorough work and major discovery that Stepanik et al made on finding the first TM-tethered FGF. I hope the reviewer will understand our arguments on this minor issue.

5) Line 277, line 435, elsewhere:

Saying that Bnl's GPI anchor "programs" its dispersion and signaling connotes a scenario where it dictates the entire process, while it might be more precise to say its regulated use is part of a program/process. Might be better to use words such as "dictates," "enables," or "instructs."

Ans: Thank you for this correction!

6) Lines 416-419:-Using "On one hand..." followed with "On the other hand..." is language that suggests there is a conflict between their observations on cytoneme dynamics and identifying Bnl as the first lipid-modified FGF and effect on cytoneme dynamics. Deleting "On one hand" and replacing "on the other hand..." with "furthermore" would better show the importance of their complementary data.

Ans: Thank you for this correction!

7) Line 476: Delete "is designed to" from "cytoneme-mediated signaling is designed to orchestrate" and use plural "orchestrates" to simplify the sentence, more effectively highlight their finding, and avoid assigning an outside intent to how these processes coordinate.

Ans: Thank you for this correction!

Reviewer #3 (Remarks to the Author):

Cytonemes are used for intercellular communications through various molecules such as Hh, Dpp,

Wg, EGF and FGF. Du et al. revealed that FGF ligand-receptor (Bnl-Btl) signaling between air-sac primordium (ASP) and wing discs involves contact of cytonemes from both tissues. They also found that bidirectional signaling, FGF signal from wing discs and unknown signal from ASP, is essential for formation of both cytonemes. Moreover, they report that Drosophila FGF (Bnl) is a GPI-anchored protein. These findings are very interesting and valuable for publication. However, the reviewer suggests some points to improve the manuscript.

Ans: We thank the reviewer for the encouragement and recommendation for publication. Before addressing the major comments, I would like to thank the reviewer again for the description: "They also found that bidirectional signaling, FGF signal from wing discs and unknown signal from ASP, is essential for the formation of both cytonemes." This was helpful for us to think of a better way to show the probable origin of bidirectionality, which was not well described in the previous version. Based on the results in pre-revised paper, we had predicted that the contact-dependent FGF-FGFR binding could directly induce a polarized reverse signaling response from the source cells, analogous to the Notch-Delta and Ephrin-Eph signaling.

In the revised version, we also tested the model that the signal from the ASP is the FGFR, and the GPI-anchored FGF acts as both a receptor and a ligand of FGFR to transduce signaling inside-out and outside-in across the source cell membrane. Although the molecular mechanisms that produce the robust reverse signaling output are unknown, our data in the revised manuscript clearly shows that the bidirectional contact-dependent FGF-FGFR interactions can initiate the reciprocal FGF-FGFR signaling (please see Figures 3, 6, 7).

Major points

1) Line 179-183: *If the expression of Bnl is affected by dia knockdown, it is difficult to conclude that source cytonemes are necessary to deliver Bnl. The authors should examine whether the expression of Bnl changes upon dia knockdown.*

Ans: Thank you for this valuable point! We now added the quantitative data on *bnl* expression in Figure S2A. This and previous experimental results presented in the Figure 2A-D, suggested that the wing disc *bnl* source *dia-i* expression has little effect on *bnl* expression levels. The genotype for this experiment was *UAS-mCherryCAAX/+; bnlGal4 /UAS-dia-i*. We analyzed - (i) Mean mCherry expression intensity in the *bnl* source and (ii) the size of the *bnl*-expressing mCherry-marked area in the control and *bnl-Gal4 > dia-i* conditions in the wing discs as described in Figure S2A. Therefore, *dia-i* expression in the source cells in *bnl:gfp^{endo}* condition (the experiment explained in line 179-183) is not expected to affect the expression of the Bnl:GFP^{endo} protein from the source. Yet, we observed a reduced inter-cellular uptake in the ASP, suggesting a defect in trafficking.

In this regard, we would like to point out that we could not directly assess the levels of Bnl:GFP^{endo} in the source *dia-i* expression analyses due to poor detectability. Theoretically, the intensity of Bnl:GFP^{endo} in the source would show its expression levels. However, as presented before in Du et al, 2018, the intensity of Bnl:GFP^{endo} on the source is very low and is challenging to detect. We need a *bnl:gfp^{endo}* homozygous condition for detectable expression (genotype was: *bnl-Gal4/UAS-dia-i, bnl:gfp^{endo}*). In addition, we need to use Airyscan nanoscopy (a combination of <200nm XY resolution and low light ultra-sensitivity) for detection of source/recipient surface Bnl:GFP^{endo}. We predict that the poor detectability of GPI-anchored Bnl:GFP on the source surface is because of its GPI-anchor. GPI-APs are known to be highly dynamic on the cell surface, unless cross-linked to form membrane nano raft-like organization. In contrast,

endocytosis in the ASP accumulates many Bnl:GFP molecules in small well separated vesicles inside ASP cells and these vesicles can be efficiently detected under light microscopy (Du et al, 2018). So, this condition was perfectly suited to measure signal uptake and dispersion patterns in the ASP.

2) *Line200-208: Analysis using dominant negative forms sometimes includes indirect effects. Thus, the authors should use dai RNAi instead of Btl:DN. Since Gal4/UAS system is temperature sensitive, the authors may reduce dai expression at different levels by doing knockdown experiments at different temperature.*

Ans: Thank you for this important point. We added new experimental results with *btl-Gal4* driven *dia-i* expression (Fig. 2M-N'). We modulated *dia-i* expression with Gal80^{ts}, as a high-level of *dia-i* expression in the trachea was larval lethal. These results showed that the reduction of polarized ASP cytonemes could also non-autonomously reduce source cytonemes.

However, we also described Btl:DN results in Figure 3, due to its fascinating ability to bind to Bnl and induce a contact-dependent retrograde signaling response in the source via cytonemes, like a CAM (Please see Figures 3; S2C-F, and related text). This provides the first indication that Btl-Bnl binding via cell-cell contact induces a bidirectional signaling response.

A major advantage of using Btl-Bnl signaling for signal dispersion is that we only have a single receptor and a ligand, and Btl and Bnl are expressed in two different tissues. Previous studies (Sato et al 2005) suggested that the Btl:DN expression in the trachea has only autonomous effects in the ASP cytoneme/ASP growth.

3) *Figure 4D-F: Activation of ERK by endogenous Bnl was reported in the previous report by Du et al (eLife, 2018). Staining with anti-dpERK in Figure 4D-F includes the endogenous activation in addition to ectopic activation by Bnl:GFP, Bnl:GFPdeltaC-TM and Bnl:GFPdeltaC expression. To avoid confusion, the authors should perform these experiments in bnl mutant with expressed Bnl:GFP, Bnl:GFPdeltaC-TM and Bnl:GFPdeltaC.*

Ans: Thank you for this important suggestion.

We agree that the description of graded patterns of signal dispersion might be confusing, although the experiments were done over a *bnl* null (*bnl-Gal4*). So, we removed any emphasis on the signal gradient and also removed original Figure 4D-F. For functionality, we focused on the probable requirement of GPI moiety in signal release and signaling. To ensure clarity in a sequence of events, we first presented experiments on the roles of GPI-anchored Bnl in organizing cytoneme contacts (Figs. 6,7) and then the experiments on the requirement of GPI on signal release in Fig 8, and then on signaling in Fig 9.

We tried to perform the experiment suggested - expressing Bnl variants in *bnl* null background. This would test for the ability of Bnl variants to rescue the lethality of the *bnl* null. If lethality was rescued, then we would be able to compare the dispersion/signaling patterns between the GPI- and non-GPI modified variants. Similar experiments were carried out for ace, encoding acetylcholinesterase found in synapse (<https://pubmed.ncbi.nlm.nih.gov/8730103/>). However, despite of several months of efforts, we failed to get the desired genotype (*bnlG4/bnl-lacZ* (or *bnl-LexA*), UAS-Bnl:GFP variants (note *bnl-GAL4*, *bnl-LacZ*, and *bnl-LexA* are *bnl* null alleles). We did not have favorable second chromosome insertions for Bnl:GFP□C and Bnl:GFP (control) (see Supplemental table on all transgenics), and we could not get recombination and linkage of UAS-x and *bnl* null alleles together in the same chromosome, which would have

allowed us to do the proposed experiments. So, we presented the old Fig. 4D-F work in the supplement (Fig. S7H-K) to provide additional support to the clonal analyses (Figs. 9A-D; S7D-G).

However, to avoid any confusion on signaling outcome, we used Bnl GOF clonal analyses (Fig. 9A-D), which provided more clear interpretation of the spatial signaling patterns than the pre-revised Fig. 4D-F (i.e., *bnlGa4* driven overexpression). Genetic mosaics provide a reliable tool for deciphering cell/tissue behavior surrounding a marked clone expressing a signaling protein. This is how a clone of Bnl was shown to attract a tracheal branch toward the source and clones of Dpp/Hh in the recipient field of cells were shown to induce change in the surrounding cells of the clones. We also have shown that a small clone of Bnl:GFP in the ASP epithelium can reorganize the fates of ASP cells surrounding the clonal source (Du et al. 2018).

Therefore, we focused on signal expression from a point source and probed for simple, measurable features. To accurately interpret our data, we analyzed Bnl GOF clones only in those areas of Btl-expressing ASP/tracheal epithelium (ASP stalk and TC) that normally do not receive Bnl and do not activate nuclear MAPK signaling (Du et al., 2018) (Fig. S7D). Importantly, we did not simply compare the dpERK levels between the conditions. We measured the number of signal recipient cells (cells with Bnl:GFP puncta) that induced MAPK per sample. We compared this correlation of signal reception and signaling on a cell-to-cell basis, irrespective of conditions (Fig. 9A-D).

In addition to MAPK patterns, Bnl signaling was assessed by its chemotactic activity inducing tracheal migration (Fig. 9E-L). To assess this phenotype properly, we used salivary glands because it is 100% trachea- and *bnl*- free organ. So, when we see tracheal growth into the organ, it is due to the ectopically expressed Bnl in this organ (Sohr et al (2019)). We showed that the level of surface-tethered Bnl:GFP variants on the salivary glands was positively correlated with the induction of tracheal growth to the organ. Thus, source-tethered signals act as better chemoattractant than the freely released \square C mutants, even though \square C can activate receptors for MAPK signaling.

I hope that the clonal analyses and salivary gland expression are sufficient to suggest that GPI-anchor is required for the spatial coordination between Bnl dispersion and signaling. These results are also consistent with the new S2 cell data presented in Fig. 6.

Please also see responses to the editor and comments #3 and #4 of reviewer 2, and additional documents for reviewers/editor.

4) Figure 5A-I: There was not a control experiment in which ASP and source cytonemes are analyzed in the wild type. The authors should analyze the length and direction of both cytonemes in the wild type.

This is an important point. In Figure 2L-N', we added the required R-plots to compare reciprocal source-recipient cytoneme distribution between the control and *btl>dia-i* mutant condition. We modified the text accordingly. Notably, by comparing Fig. 2L,L' to Fig. 6F, it is intriguing to see that ASP and source cytoneme numbers were slightly reduced, and ASP cytonemes were relatively shorter and more dynamic with Bnl:GFP overexpression in *bnl* source. This is due to the growth of the ASP, which brought the ASP and source relatively close to each other. However, the important point we wanted to highlight was reciprocal polarity and change in the source and ASP cytonemes. And this reciprocal polarity and length of cytonemes were lost with Bnl:GFP \square C and gained with Bnl:GFP \square C-TM. Thus, we assessed the gain and loss of the normal

bidirectionality of cytoneme-mediated signaling by solely comparing overexpressed Bnl:GFP and its non-GPI variants.

In addition, in Figure 1H", we presented the analysis of source cytoneme length distribution. Individual source cells were marked using clonal expression of CD8:GFP within the *bnl*-expressing disc area using Flybow technique as described in Methods. Since cytonemes were recorded only from the narrow basal sides of each disc columnar epithelial cell, and since all the cytonemes were oriented vertically only toward the overlaying ASP, we did not use R plot for these experiments.

Notably, we already published similar R-plots for the ASP cytonemes, showing variable numbers, lengths, and polarity of *bnl*-receiving cytonemes depending on the position of recipient cells relative to the *bnl*-source (Du et al., 2018).

5) *It is reported that MAPK is efficiently activated in endosomes where a receptor-ligand complex localizes. However, Bnl:GFPdeltaC-TM that cannot be shed in the plasma membrane could activate MAPK in ASP. How do you explain this?*

Ans: Thanks for this intriguing question! We added additional results using S2 cells (Figs. 3K,K'; 6A,B; S5C,D) showing that Bnl:GFP \square C-TM (TM) can activate the MAPK pathway in both cell-autonomous and non-autonomous (contact-dependent, juxtacrine) manners (Fig. 6B,C). We also detected internalized receptor-bound TM puncta within the recipient cells, and recipient cells with relatively higher levels of the internalized puncta often had the dpERK. Although, the efficiency of Bnl:GFP \square C-TM internalization is relatively lower than the control Bnl:GFP.

It is possible that Bnl:GFP \square C-TM is shed at a lower rate from the surface by a proteolytic enzyme. In *in-vivo* experiments, we often found that the ASP can internalize the Bnl-TM-colocalized source cell membrane (Figs. 8D,E; S7A-C'). Movie S11 shows that strong cytoneme:cytoneme adhesion led to cytoneme breakage and absorption of the source membrane within the ASP. On the other hand, although Bnl:GFP \square C without a TM is often found in the endocytic vesicles within the ASP or S2 cells, they often did not induce nuclear dpERK. We suspect a polarized signal delivery/reception via contact might have some regulatory effects on intracellular trafficking of MAPK signaling components.

However, it is important to note that the requirement for receptor endocytosis is not universal as several receptors have been shown to activate MAPK without internalization. While receptor dimerization and activation of the MAPK pathway begin in the inner cell membrane, receptor-mediated endocytosis is likely to regulate the robustness of the process.

6) *Shedding of Bnl in the contact site of cytonemes is essential for the signaling. What is a shedding enzyme responsible for Bnl? This can be discussed.*

Ans: We added a discussion: Although the Bnl release is critical, we do not know how Bnl is released from its GPI-tether, specifically at the cytoneme contact sites. We speculate an enzymatic shedding that might cleave either the GPI moiety or the protein. We hope to identify and characterize such processes in the future.

Minor points

Line137: It would be good to the readers to explain what is mCherryCAAX.

Ans: Thank you for this suggestion! We now introduced mCherryCAAX as a "prenylated mCherry for membrane marking".

REVIEWERS' COMMENTS

Reviewer #1 (Remarks to the Author):

GPI-1 anchored FGF directs cytoneme-mediated bidirectional signaling to self-regulate tissue-specific dispersion_Revised

We consider that the manuscript by Du et al., describing for the first time the GPI anchoring of the morphogen FGF to plasma membrane for cytoneme localization, has been revised properly. The key discoveries on cytoneme-mediated target selection for FGF signal release and feedback regulations of these processes are now better presented and the experimental evidence improved. The authors demonstrate that the anchored Bnl-GPI is acting as a local CAM promoting cytoneme guided-establishment that facilitates cytoneme contact for the long range signaling between two different tissues. The new data also clearly show that bidirectional FGF-FGFR contact-dependent interactions can initiate reciprocal FGF-FGFR signaling, although the molecular mechanisms are still unknown.

The authors have experimentally addressed some of our suggestions, including the R-plots quantifications. Although the experiments we suggested regarding how HSPGs might affect cytoneme mediated FGF signaling have still not been performed, comments have been made about the role of glypicans on cytoneme stability. The authors also clarified and boosted the molecular proofs for GPI addition to FGF as requested.

In addition, the authors have also included more experimental evidence using cultured cells, verifying the CAM properties of Btl and Bnl and the signaling capability of all the Bnl:GFP variants used. Four additional figures have been included together with additional panels, providing further proof of the GPI anchoring of FGF and its functional implications, following as well other reviewers requests. Finally, the authors also have improved the schemes to better explain conclusions.

Thus, we agree with the manuscript's publication, once some small details are tackled:

1. Line 126-127 Results: requires rewriting as part of the sentence has been repeated.
2. Panels N and M in Fig. 1 are not mentioned in the main text.
3. The labelling in panels L-N in Figure 1 is not clear. The placing of the colour labels specifying source or ASP cytonemes for the three graphs is confusing. We suggest referring to these colours in the figure legend.
4. In general, the manuscript could be revised for fluency as reading through it can result repetitive at points, specially through the results section.

Reviewer #2 (Remarks to the Author):

The revised manuscript by Du et al is much improved. In particular, the presentation was easier to follow and the reviewer appreciates the effort of the authors to add additional supporting evidence - in particular data for clones presented in Figure 7F-M.

In the opinion of this reviewer, I believe that the paper would have been stronger if a mutant was generated to examine the role of GPI linkage for endogenous Bnl. The authors state that it's difficult to assay mutants due to compounding effects; nevertheless, clones are possible and it's alternatively possible that little change might result; for instance, that GPI linkage is not necessary for function but, perhaps, increases robustness of association. However, this would take months to accomplish so it is understandable if the authors want to publish this initial study as is.

However, I don't necessarily agree that the authors have identified bidirectional "signaling" and this discussion warrants revision. Yes - they can reorient ASP or Bnl source protrusions with clones (see Fig. 7F-M). This points to physical interaction of cytonemes. MAPK signaling is activated in the forward direction, but no "signaling" pathway has been identified that is activated in the Bnl source cell. While the title and the abstract are appropriate; the last model figure with label "bidirectional signaling" is an overstatement/confusing. They have identified cytoneme-cytoneme interactions but they have no evidence for reverse signaling in the Bnl expressing cells; therefore, to avoid confusion, they should not refer to what they have identified as bidirectional signaling. Perhaps bidirectional cytoneme-cytoneme associations?

The Bnl hydrophobic C-terminal domain was shown to promote cell membrane localization in a recent publication, the authors choose not to acknowledge this. It's unfortunate as it would support the view that the current analysis is that much more impactful, as GPI linkage was not entertained by this other group. Du et al. having identified the second Drosophila FGF to be membrane-associated only makes it more interesting to understand the diverse mechanisms by which FGF presentation is regulated.

I commend the authors on a beautiful study that makes an important contribution to the field. Despite the lack of endogenous mutants to support a pivotal role for GPI linkage, the authors have done many other experiments that support a model for cytoneme-cytoneme supported cell-cell interactions between Bnl and Bth-expressing cells. They should however temper the conclusion that they have demonstrated bidirectional 'signaling'.

Minor comments:

i) What are arrows in Fig 1E that are large green spots not associated with cytonemes? Please explain in the figure legend.

ii) Figure 1L-N. Show datapoints as in other graphs - what is "n"?

iii) Figure 2M-M''' - not described in the figure legend.

iv) Figure 5G-L. Can you use different dashed lines to signify ASP versus source cells.

v) Figure 6B. Is only one Btl-Cherry cell positive for dpERK? Why?

vi) Figure 6F,I,L. How many discs were analyzed? What is "n"?

vii) Figure 8A-E. In general, there is a concern that the source of GFPdeltaC is smaller than BnlGFP. In the supplement, can the authors show images of the source in order to be sure that levels are comparable. Also, where is the ASP in Figure 8B? Why is there no blue signal?

viii) The result in Fig 9F is interesting. However, the authors should show a western blot to ensure that the levels of 3 molecules compared in Fig 9 are equivalent (i.e. Bnl, Bnl.delta.TM, and Bnl.delta.C)

ix) As discussed above, cell-cell adhesion does not prove there is reverse signaling.

Reviewer #3 (Remarks to the Author):

The authors answered my comments and improved the manuscript.

REVIEWERS' COMMENTS

Reviewer #1 (Remarks to the Author):

GPI-1 anchored FGF directs cytoneme-mediated bidirectional signaling to self-regulate tissue-specific dispersion_Revised

We consider that the manuscript by Du et al., describing for the first time the GPI anchoring of the morphogen FGF to plasma membrane for cytoneme localization, has been revised properly. The key discoveries on cytoneme-mediated target selection for FGF signal release and feedback regulations of these processes are now better presented and the experimental evidence improved. The authors demonstrate that the anchored Bnl-GPI is acting as a local CAM promoting cytoneme guided-establishment that facilitates cytoneme contact for the long range signaling between two different tissues. The new data also clearly show that bidirectional FGF-FGFR contact-dependent interactions can initiate reciprocal FGF-FGFR signaling, although the molecular mechanisms are still unknown.

Ans: Thank you for encouraging comments.

The authors have experimentally addressed some of our suggestions, including the R-plots quantifications. Although the experiments we suggested regarding how HSPGs might affect cytoneme mediated FGF signaling have still not been performed, comments have been made about the role of glypicans on cytoneme stability. The authors also clarified and boosted the molecular proofs for GPI addition to FGF as requested.

In addition, the authors have also included more experimental evidence using cultured cells, verifying the CAM properties of Btl and Bnl and the signaling capability of all the Bnl:GFP variants used. Four additional figures have been included together with additional panels, providing further proof of the GPI anchoring of FGF and its functional implications, following as well other reviewers requests. Finally, the authors also have improved the schemes to better explain conclusions.

Ans: Thank you for all comments and suggestions that helped to improve this paper.

Thus, we agree with the manuscript's publication, once some small details are tackled:

1. Line 126-127 Results: requires rewriting as part of the sentence has been repeated.

Ans: Thank you! the sentence is corrected as follows on page 5:

" Bnl is produced in the wing disc and transported target-specifically to the overlaying ASP via Btl-containing ASP cytonemes across a layer of interspersed myoblasts (Fig. 1a,b) ⁹."

2. Panels N and M in Fig. 1 are not mentioned in the main text.

Thank you! We corrected as follows:

"Both cytoneme types had short lifetimes and repeated cycles of contact association-dissociation (Fig. 1I-n; Supplementary Figure 1e-h, Movie 2, Table 1).

3. The labelling in panels L-N in Figure 1 is not clear. The placing of the colour labels specifying source or ASP cytonemes for the three graphs is confusing. We suggest referring to these colours in the figure legend.

Ans: We slightly modified the Figure panels to address this issue. We also corrected the legend for Figure 1I-n as follows (page 38):

" **i-n** Contact-dependent reciprocal guidance of source (red) and ASP (green) cytonemes (*btl-Gal4,UAS-CD8:GFP/+; bnl-LexA, LexO-mCherryCAAX/+*)."

4. In general, the manuscript could be revised for fluency as reading through it can result repetitive at points, specially through the results section.

Ans: Thank you for this suggestion! We revised the text and tried to remove repetitive sentences.

Reviewer #2 (Remarks to the Author):

1. Comment: The revised manuscript by Du et al is much improved. In particular, the presentation was easier to follow and the reviewer appreciates the effort of the authors to add additional supporting evidence - in particular data for clones presented in Figure 7F-M.

Ans: Thank you for all comments, questions, and suggestions that helped us to strengthen our finding on the CAM-like bidirectional signaling of a GPI-anchored FGF and its role in polarized patterns of signaling responses in both source and ASP cells via cytonemes.

2. Comment: In the opinion of this reviewer, I believe that the paper would have been stronger if a mutant was generated to examine the role of GPI linkage for endogenous Bnl. The authors state that it's difficult to assay mutants due to compounding effects; nevertheless, clones are possible and it's alternatively possible that little change might result; for instance, that GPI linkage is not necessary for function but, perhaps, increases robustness of association. However, this would take months to accomplish so it is understandable if the authors want to publish this initial study as is.

Ans: Thank you! We provided strong evidence from multiple different ways that Bnl is GPI anchored, and GPI-anchoring enables Bnl to act as a CAM to induce CAM-like FGF-FGFR bidirectional signaling for cytoneme contacts prior to its contact-dependent release for tissue-specific dispersal.

3. Comment: However, I don't necessarily agree that the authors have identified bidirectional "signaling" and this discussion warrants revision. Yes - they can reorient ASP or Bnl source protrusions with clones (see Fig. 7F-M). This points to physical interaction of cytonemes. MAPK signaling is activated in the forward direction, but no "signaling" pathway has been identified that is activated in the Bnl source cell. While the title and the abstract are appropriate; the last model figure with label "bidirectional signaling" is an overstatement/confusing. They have identified cytoneme-cytoneme interactions but they have no evidence for reverse signaling in the Bnl expressing cells; therefore, to avoid confusion, they should not refer to what they have identified as bidirectional signaling. Perhaps bidirectional cytoneme-cytoneme associations?

Ans: Thank you for the opinion, but unfortunately, this is in contrast to other reviewers' comments. For instance, the reviewer#1 indicated that " The new data also clearly show that bidirectional FGF-FGFR contact-dependent interactions can initiate reciprocal FGF-FGFR signaling, although the molecular mechanisms are still unknown."

However, this provided us with an opportunity to explain and strengthen the paper, where needed. For instance, We agree that the reverse signaling pathway is not identified; we did not claim to do so. However, the reviewer agreed that we showed "bidirectional cytoneme-cytoneme association". Therefore, I would also hope that the reviewer would appreciate so many of our experimental data, which provided strong evidence (see Figures 2, 3, 6, 7, 10a) that this bidirectional/reciprocal cytoneme-cytoneme-forming cell-polarizing response is caused by **CAM-like contact-dependent Btl-Bnl signaling responses**.

Experimental evidence: Btl-DN can bind to Bnl via its extracellular ligand-binding domain (Supplementary Figure 2b) and can heterodimerize with *WT* Btl, but cannot activate nuclear MAPK signaling (transcriptional response) due to the lack of its intracellular kinase domain. Strikingly, Btl-DN-containing ASP cytonemes when contacted the Bnl source or Bnl, it induced polarized cytoneme-formation (Fig.3a-c). In cultured S2 cells, Btl-DN expressing and Bnl expressing cells could reciprocally recognize and adhere to each other by CAM-like receptor-ligand binding across the contact sites. These results suggested that the polarized cytoneme-forming responses in the ASP/recipient cells induced by the Btl:DN-Bnl binding is due to a CAM-like signaling for which transcriptional signaling output is not required. CAM-CAM interactions induces local cytoskeletal organization in the membrane cortex that might be sufficient to produce cytoneme polarity.

2. Similarly, we showed that a membrane-tethered form of Bnl can induce the polarized cytoneme-forming response in the source by binding to Btl. This response also polarizes the Bnl presentation and cytoskeletal organization. Therefore, Btl-Bnl binding via cell-cell contacts induced a signaling response in the source.

3. With many different experiments we showed that the the response in the source is dependent on Btl-Bnl interactions and is produced when Bnl is tethered to the source

cell membrane. Bnl Δ C is unable to induce a reciprocal signaling response due to the lack of its source membrane tether. However, when we tethered the same Bnl Δ C to its source membrane with a TM domain, it efficiently could regain its bidirectional signaling. Similarly, a soluble Btl/FGFR that lacked a TM domain did not induce bidirectional responses, suggesting that the Btl-Bnl interactions via contacts induces CAM-like bidirectional responses. Supportive evidence presented: in vivo - Fig. 1,2,3a-d'; 6d-l, 7a-k; Suppl Figs. 1a-h; 2a-d'; 6a-h' in vitro - Figs. 3f-n; 6a-c; Suppl Figs. 2f-j'; 5c,d. (Please also read the explanation in the minor comment # ix).

4. Transcriptional output is a critical step toward achieving the steady-state fates, but not all signaling outputs are nuclear/transcriptional. For instance, non-canonical signaling responses are cytoplasmic, transient, and fast. CAM-CAM receptor-ligand binding induce responses in cells by modulating the local distribution of CAMs and many subcellular regulators (e.g., actomyosin complex, small GTPases, polarity proteins) in the membrane cortex. We cited these papers.

Therefore, throughout the text, we ensured that we describe the bidirectional Btl-Bnl signaling process as a "CAM-like" or "contact-dependent" signaling response. In addition to clarifying this point in the text, we slightly modified the Fig 10. Moreover, we clearly defined the green dashed arrows as proposed mechanism and black arrows that are experimentally validated. In addition, as per requests, we tried to tone down the claim for reverse signaling. Instead, we tried to clearly write these sections with proper discussion of the experimental results. I hope these changes would be helpful.

4. Comment:

The Bnl hydrophobic C-terminal domain was shown to promote cell membrane localization in a recent publication, the authors choose not to acknowledge this. It's unfortunate as it would support the view that the current analysis is that much more impactful, as GPI linkage was not entertained by this other group. Du et al. having identified the second Drosophila FGF to be membrane-associated only makes it more interesting to understand the diverse mechanisms by which FGF presentation is regulated.

Ans: We cited all relevant papers as requested. Also, we discussed the conceptual contribution of the discoveries of TM-Pyramus and GPI-anchored Bnl in the last paragraph of the discussion:

Reference:

"However, to drive heterophilic CAM-like bidirectional recognition for synapse, Bnl needs to be tightly associated to the source cell membrane. The source surface localization of Bnl was known to be critical for its functions^{36,37}. Moreover, Bnl is likely to be a membrane-associated protein³⁸, despite of its ability to disperse over long range⁹. How might a secreted protein be associated exclusively on the source cell surface to act as a CAM and is it both inhibited and activated for dispersal?

5. Comment: I commend the authors on a beautiful study that makes an important contribution to the field. Despite the lack of endogenous mutants to support a pivotal role for GPI linkage, the authors have done many other experiments that support a model for cytoneme-cytoneme supported cell-cell interactions between Bnl and Bth-expressing cells. They should however temper the conclusion that they have demonstrated bidirectional 'signaling'.

Ans: Thank you for the comments. Please see answers to comment #3.

Minor comments:

i) What are arrows in Fig 1E that are large green spots not associated with cytonemes? Please explain in the figure legend.

Ans: Thank you for pointing this out. These are endocytosed Rab7-positive vesicles containing receptor-bound Bnl:GFP in ASP cells (shown before by Du et al, 2018).

Corrected legend (page 37): "dashed arrow, Bnl:GFP puncta in internalized vesicles⁹ within the ASP (dashed line); arrow, Bnl:GFP puncta on source cytonemes;"

ii) Figure 1L-N. Show datapoints as in other graphs - what is "n"?

Ans: Thank you! We added data points. n numbers are provided in the Supplementary Figure 1g,h and Supplementary Table 1.

iii) Figure 2M-M'" – not described in the figure legend.

Ans: Thank you! We corrected this.

iv) Figure 5G-L. Can you use different dashed lines to signify ASP versus source cells.

Ans: Thank you! We corrected this.

v) Figure 6B. Is only one Btl-Cherry cell positive for dpERK? Why?

Ans: Thank you for the question. We now clarified this in the text.

This is due to the inefficient release of the Bnl:GFP Δ C-TM at the contact sites. We showed that TM-tethered signals are poorly released at the cell-cell cytoneme contact sites. Apparently, the signal release and receptor-mediated endocytosis are critical for the MAPK signaling activation. In our experiments, nuclear MAPK signaling localization correlated with the increased number of cell-internalized Bnl puncta in the recipient cells.

"In contrast to S2-Bnl:GFP Δ C, S2-Bnl:GFP Δ C-TM cells selectively trans-adhered to S2-Btl:Cherry as efficiently as S2-Bnl:GFP by forming polarized trans-synaptic receptor-ligand co-clusters (Fig.6b,c; Supplementary Table 3b). Polarized trans-pairing of Bnl:GFP Δ C-TM and Btl:Cherry also induced MAPK

signaling in the adhering Btl:Cherry-expressing cells, but at a lower frequency than the control S2-Bnl:GFP::S2-Btl:Cherry interactions (Fig.6b; Supplementary Table 3b). Notably, MAPK signaling was activated only in those trans-paired S2-Btl:Cherry cells that had high numbers of internalized Bnl:GFP Δ C-TM puncta (Fig.6b). It is possible that despite being TM-tethered on the source, Bnl:GFP Δ C-TM could somehow be released and internalized into some of the adhering recipient cells through the cell-cell contact sites. Irrespective of the activation of MAPK signaling, the trans-synaptic binding of Bnl:GFP Δ C-TM and Btl:Cherry was sufficient to induce reciprocal polarity of signal delivery and reception. This is consistent with the CAM-like activity of membrane-tethered Bnl."

vi) Figure 6F,I,L. How many discs were analyzed? What is "n"?

Ans: Thank you for checking this. We now provided the n values in the figure panels. We also mentioned that the n is provided in Supplementary Figure 6) as follows:

" f,i,l, R-plots comparing numbers, length, and directionality of ASP and source cytonemes as indicated; n = number of discs analyzed (also see Supplementary Figure 6a-g)."

vii) Figure 8A-E. In general, there is a concern that the source of GFPdeltaC is smaller than BnlGFP. In the supplement, can the authors show images of the source in order to be sure that levels are comparable. Also, where is the ASP in Figure 8B? Why is there no blue signal?

Thank you very much! There was an unintended mistake in the scale bars of Figure 8b,c. This occurred during the compilation of the multi-panel Figure. We corrected the scale bars.

Note that we showed two different deletions (long and short) of Δ GPI and its corresponding Δ GPI-TM. According to our measurements, irrespective of the constructs, there are little significant differences in the size of the *bni* source or the expression levels of the signals (note that the green puncta in the source area show expression and the mCherry level show *bni* enhancer activity). The steady-state levels of GFP to mCherry in the source are equivalent for all constructs. We added an example of measurements we performed to verify equivalent levels of expression in Supplementary Information>Supplementary Notes>Section C.

However, the reviewer might also be referring to the variable positioning of the source cells relative to the ASP tip obtained with Bnl:GFP, Δ GPI, and Δ GPI-TM expression. This is expected. According to our model of CAM-like bidirectional communications can inform source and recipient cells to reciprocally coordinate in space. This is consistent with our previous findings (Du et al. 2017), showing that the *bni*-source and tracheal cells in the embryo can reciprocally guide each other and co-migrate in synchrony. Consequently, a loss of cytoneme-cytoneme signaling contacts due to the loss of the GPI moiety of Bnl:GFP Δ GPI mutant leads to the loss of its ability to induce such reciprocal coordination between the ASP and source. On the other hand, the same Δ GPI mutant, when added with a TM domain, it induced the source cells to strongly polarize and adhere to the ASP (please also see Suppl. Fig. 7a-c'). GPI-anchoring provides a balance between these two extreme phenotypes.

About 8b: As indicated in the Figure and legends, blue is the Dlg staining, marking cell-cell junctions. This was not shown in the panel to clearly highlight green puncta in the black background. Now, we added an inset with the extended Z-stack of the same sample, showing the ASP and the Dlg stain (blue).

In Figure 8A-E, we planned to show the distribution of Bnl variants into the non-expressing wing disc areas. This is why we showed only extended projections of the 60-80 Z-sections encompassing the wing disc cells and the disc-ASP interface. We had imaged and examined additional 50-60 Z sections encompassing the disc-overlying ASPs. We showed these 3D projections in Supplementary Movies 12-14. We now added two additional movies (Supplementary Movies 12-16) to show 3D projections of all the tissues samples that were shown in Figure 8A-E panels.

viii) The result in Fig 9F is interesting. However, the authors should show a western blot to ensure that the levels of 3 molecules compared in Fig 9 are equivalent (i.e. Bnl, Bnl.delta.TM, and Bnl.delta.C)

Ans: These Figure panels were presented in pre-revised versions along with the graphs (Fig. 9I) indicating the levels and distributions of the Bnl variants in these tissues. However, we added additional modifications in the revised text to clarify any confusion. In the text, we now mentioned that we used the *bnl-Gal4* driver (that is ectopically expressed in the salivary glands) to drive the expression of the proteins at comparable levels. Figure 9I is now described in more detail. Note that the GFP fluorescence levels in each panel represent the total protein expressed in these tissues. We also detected the same proteins on the cell surface by using an anti-GFP antibody under non-permeabilized conditions. The ratio of extracellular to total proteins was compared between genotypes to produce the graph in Fig. 9I.

However, based on the reviewer's request, we now provided an additional example of these measurements in Supplementary Information>Supplementary Notes>Section C. The *bnl-Gal4*-driven equivalent expression levels were also shown in Figure 5.

In the previous version of the response to reviewers' comments (see the response to comment# 1, reviewer #1), we showed published data on Bnl:GFP Western Blot and discussed the difficulty in detecting/quantitating the total protein based on the multiple Bnl bands (due to multiple cleavage and post-translational processing). A lack of good antibodies against the entire Bnl protein causes difficulty in the biochemical detection of the protein.

ix) As discussed above, cell-cell adhesion does not prove there is reverse signaling.

Ans: Cell-cell reciprocal affinity, adhesion, sorting, and repulsion are the results of an active bidirectional matchmaking process involving CAMs. At the molecular levels, homophilic or heterophilic CAM-CAM interactions between two cells are known to induce polarized recruitment of the CAMs and actomyosin complex to the contact sites.

Polarized localization of CAMs to the contact sites, in turn, is required to self-sustain cell-cell contacts. Consequently, reciprocal polarity and trans-adherence between CAM-expressing cells provide a reliable phenotype to suggest CAM-like bidirectional interactions. I hope I could explain the basis for the interpretation of the bidirectional Btl-Bnl signaling by cell-cell trans-pairing in our experiments. The additional explanation of our experiments and results are provided in the main text and comment #3 (reviewer #2) in the current document.

Reviewer #3 (Remarks to the Author):

The authors answered my comments and improved the manuscript.

Thank you!